# HIERARCHICAL ONE-CLASS DATA DESCRIPTION VIA PROBABILISTIC GRANULAR-BALL COMPUTING

## ABSTRACT

One-class data description aims to model the distribution of target data by constructing a compact representation of the target class. This approach is widely applied in tasks like anomaly detection, where the objective is to differentiate the target data from outliers. Traditional methods typically rely on single-sphere or pre-defined multi-sphere representations. However, these simplistic assumptions often fail to capture the anisotropic structures and intricate patterns present in real-world data, limiting their effectiveness in representing distributions across multiple scales. To address these limitations, we propose Probabilistic Granular-ball Computing (PGBC), a hierarchical framework for one-class data description. PGBC uses ellipsoidal granular-balls to align with the anisotropic geometry of data and recursively refines them through statistical splitting, achieving precise and adaptive data representation. Additionally, PGBC approximates a hierarchical Gaussian mixture model by aggregating data description scores via granular-ball distribution entropy at each layer. This enables PGBC to capture data patterns at multiple levels of granularity, modeling both global structures and fine local variations. Extensive experiments on benchmark datasets demonstrate that PGBC consistently outperforms related strong baselines, offering superior accuracy for hierarchical one-class data description while maintaining a low false positive rate.

## 1 INTRODUCTION

One-class classification has become increasingly critical in real-world scenarios where acquiring representative anomalous samples are impractical, unpredictable, or even hazardous such as in mechanical failure detection (Pang et al., 2021), arrhythmia diagnosis from ECG signals (Kavya et al., 2024), or cybersecurity intrusion monitoring (Patcha & Park, 2007). In these domains, anomalies are not only rare but also highly diverse in form and origin, making it infeasible to comprehensively define them through labeled datasets. To address these challenges, one-class data description focuses solely on modeling the intrinsic structure of normal data, without requiring labeled anomalies (Pimentel et al., 2014; Ruff et al., 2018). Unlike conventional supervised approaches that rely on both normal and abnormal examples, one-class methods construct a reference model of normality and identify inputs that deviate from this reference as potential anomalies (Schölkopf et al., 2001; Tax & Duin, 2004). By isolating the learning process from the variability and unpredictability of anomalous events, one-class data description offers a robust and versatile framework, particularly well-suited for safety-critical or data-scarce applications where systems must autonomously detect novel or unexpected behaviors (Ruff et al., 2021). Nevertheless, anomaly detection in practice remains highly challenging, as real-world data distributions often demonstrate complex and anisotropic structures, where local density varies significantly with direction. This complexity underscores the need for models that can achieve high detection accuracy while also ensuring low false positive rates.

Traditional one-class data description methods often rely on spherical boundaries. For instance, *Support Vector Data Description (SVDD)* (Tax & Duin, 2004) encloses data in a hypersphere in feature space, and *DeepSVDD* (Ruff et al., 2018) learns deep latent representations to tighten the sphere around normal samples. Extensions such as *MCDD* (Lee et al., 2020) and *THOC* (Shen et al., 2020) employ multiple spheres to enhance flexibility, yet they require the number of spheres to be predefined and still enforce isotropic boundaries.

*Granular-ball computing* (Xia et al., 2019; 2020; Xie et al., 2025) offers an adaptive alternative by automatically generating multiscale hyperspheres without predefining their number. While more flexible, existing granular-ball methods (e.g., GBDO, GBMOD) also assume isotropic shapes, which struggle with elongated clusters and lead to redundant overlapping spheres when modeling anisotropic geometries. As illustrated in Figure 1, isotropic sphere-based granular-ball approaches approximate clusters using equal-radius contours to define simplified boundaries. In such representations, two points that are equidistant from a centroid are always assigned the same anomaly score, thereby ignoring that real data often concentrates heavily along a principal component direction. Crucially, this limitation cannot be resolved by simply aggregating geometric distances (e.g., average distance to centroids), as iso-probability contours in anisotropic distributions often diverge from iso-distance contours. This limitation often leads to redundant components for elongated clusters and further increases the number of false positives by misclassifying points aligned with the dominant geometry.

This motivates us to adopt the probabilistic ellipsoidal granular-ball depicted in Figure 1(b) for constructing a one-class data description. Specifically, this approach allows the granular-ball to flexibly adjust its shape by stretching along the principal component directions of the data. As a result, two points with the same geometric distance from the centroid may lie on different $\sigma$-level contours of the ellipsoid, leading to significantly different anomaly scores. This alignment with the intrinsic data geometry both reduces redundancy and suppresses false alarms, thereby providing a more expressive and reliable framework for anomaly detection. The formal definition and computational details of granular-balls are provided in Appendix A.

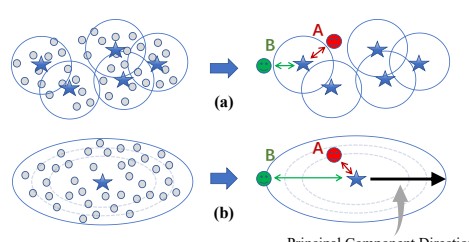

Figure 1: (a) Traditional granular-ball representation uses isotropic spheres, which struggle to fit complex or anisotropic regions without excessive splitting. (b) The proposed probabilistic granular-balls form ellipsoidal regions aligned with the principal component direction, offering adaptive shape and orientation for more efficient coverage.

In addition, AutoEncoder (Sakurada & Yairi, 2014) and Deep Autoencoding Gaussian Mixture Model (DAGMM) (Zong et al., 2018) are also widely used to enhance expressiveness by learning compact representations. However, this often comes at the expense of intensive training requirements and high sensitivity to hyperparameters. Meanwhile, Hierarchical Gaussian Mixture Normalizing Flow Modeling (HGAD) (Yao et al., 2024) presents a hierarchical probabilistic approach that captures anisotropic structures through component-wise covariances and flows. However, HGAD's reliance on a predefined structural design limits its ability to adapt to the unknown complexity of the data, as both the number of mixture components and the hierarchical levels are fixed beforehand, rather than being dynamically learned from the data's intrinsic geometry.

In this paper, we propose *Probabilistic Granular-ball Computing* (PGBC) for hierarchical one-class data description. PGBC models data with ellipsoidal granular-balls that adaptively align with principal components, overcoming the geometric rigidity of spherical granular-balls. By combining the geometric flexibility of ellipsoids with the statistical rigor of Gaussian components, PGBC provides an expressive and statistically grounded representation of data distributions. Moreover, each granular-ball is refined iteratively using statistical criteria such as the Bayesian Information Criterion (BIC) and log-likelihood improvement, allowing the model to adjust its complexity automatically without deep architectures or fixed mixture sizes. For anomaly detection, PGBC organizes granular-balls into a hierarchical structure resembling a Gaussian mixture model, where anomaly scores are aggregated across layers using entropy-based weights, enabling detection at multiple levels of granularity. Overall, this design combines geometric adaptivity with principled statistical refinement, providing both flexibility and robustness for one-class data description.

Our main contributions are summarized as follows:

i) We introduce probabilistic granular-ball computing (PGBC), a hierarchical one-class data description framework that adaptively captures local data distributions by iteratively splitting and refining ellipsoidal granular-balls, enabling both anisotropic and probabilistic modeling of the data.

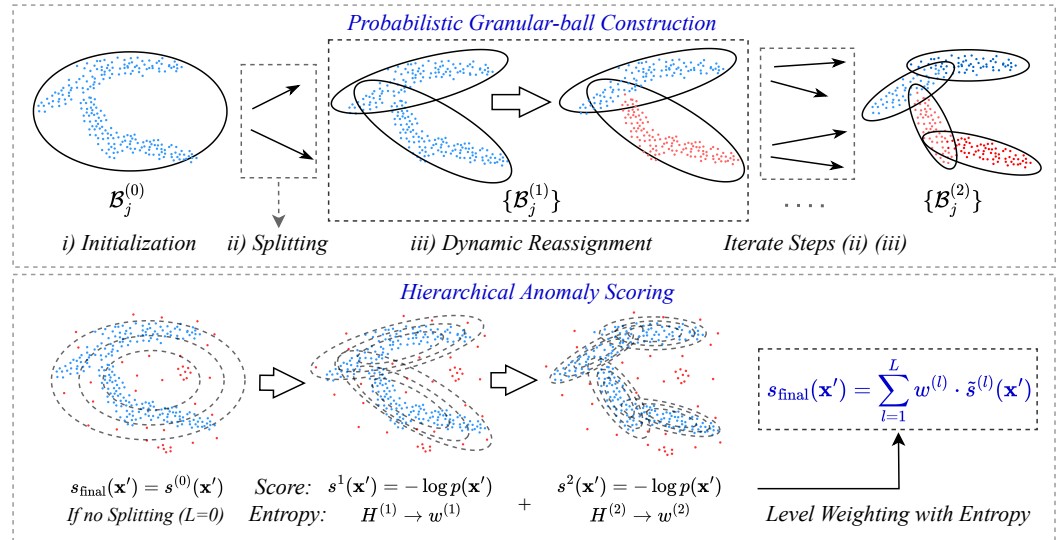

Figure 2: Framework of probabilistic granular-ball computing (PGBC). For data with a single principal component (no splitting), the model simplifies to a single global Gaussian.

ii) To complement this, we propose a systematic anomaly scoring mechanism that aggregates likelihoods across hierarchical levels by entropy-based weighting, effectively amplifying consistent abnormality signals while suppressing spurious noise, thereby reducing the false positive rate.

iii) Extensive experiments demonstrate the superiority of PGBC, consistently outperforming recent state-of-the-art baselines in both tabular and time series anomaly detection tasks.

## 2 METHODOLOGY

**Problem definition and notations.** In the context of one-class data description, we are given a set of $N$ training samples $\mathcal{X} = \{\mathbf{x}_1, \mathbf{x}_2, \ldots, \mathbf{x}_N\}$, where each $\mathbf{x}_i \in \mathbb{R}^d$ is a $d$-dimensional feature vector representing normal data. The goal is to learn a compact representation of the normal data distribution $p(\mathbf{x})$ based solely on $\mathcal{X}$. This description serves as the foundation for distinguishing normal samples from anomalies. For a given test sample $\mathbf{x}'$, its anomaly score is determined by quantifying how much it deviates from the learned description of the normal data. Samples that significantly deviate are identified as anomalies, while those that align closely with the one-class description are classified as normal.

**Overview.** In this section, we formally present the proposed framework of Probabilistic Granular-Ball Computing (PGBC). As depicted in Figure 2, the PGBC pipeline consists of two main phases: (i) *Probabilistic Granular-Ball Construction* (top) and (ii) *Hierarchical Anomaly Scoring* (bottom).

The first phase, *Probabilistic Granular-Ball Construction*, adaptively builds a hierarchical one-class data description through three core steps: (i) *Initialization*, (ii) *Recursive Splitting Strategy*, and (iii) *Dynamic Reassignment*. Steps (ii) and (iii) are alternated iteratively to progressively refine the description of the underlying data distribution. At each iteration, a probabilistic model is fitted to the data encapsulated within a granular-ball, which is then divided into smaller components. Each granular-ball models a local Gaussian distribution, and this recursive process systematically constructs a data-driven mixture tree. The resulting hierarchy captures both global and local structures of the normal data, providing a comprehensive and adaptive probabilistic one-class description.

The second phase, *Hierarchical Anomaly Scoring*, leverages the constructed hierarchy to compute anomaly scores for test samples of the data. Instead of relying solely on the leaf-level granular-balls, PGBC aggregates scores across multiple levels of the hierarchy. This multilevel aggregation combines both global and local perspectives, enabling robust anomaly detection by capturing coarse-grained and fine-grained patterns in the data. By integrating information from different levels, PGBC

ensures that anomalies are ultimately and effectively identified, regardless of whether they deviate from global trends or local structures.

## 2.1 Probabilistic Granular-Ball Construction

To capture the anisotropic and locally-varying structures of data, we extend classical granular-balls into a data-driven hierarchical structure composed of ellipsoidal probabilistic components. Each region is represented by a Gaussian distribution, with its mean and covariance matrix adaptively estimated to align with the local data geometry. This approach systematically captures structural variability and enables flexible density estimation across multiple levels of granularity. Formally, the definition of a probabilistic granular-ball is provided in Definition 1.

**Definition 1** (Probabilistic Granular-Ball). *A probabilistic granular-ball $\mathcal{B}$ is a Gaussian approximation in $\mathbb{R}^d$ defined by the following parameters: i) A mean vector $\mu \in \mathbb{R}^d$, representing the center of the data distribution within the region. ii) A covariance matrix $\Sigma \in \mathbb{R}^{d \times d}$, encoding the shape, orientation, and dependencies within the data distribution of the region.*

The PGBC framework constructs a hierarchical density representation of the data by sequentially executing three main steps: i) *Initialization*, ii) *Recursive Splitting*, and iii) *Dynamic Reassignment*. Steps 2 and 3 are performed alternately to iteratively refine and model underlying data distribution.

**Step 1: Initialization.** The construction process begins with an initial probabilistic granular-ball $\mathcal{B}^{(0)}$ that encapsulates the entire dataset $\mathcal{X}$. The parameters of $\mathcal{B}^{(0)}$ are computed as follows:

$$\mu^{(0)} = \frac{1}{N} \sum_{i=1}^{N} \mathbf{x}_i, \quad \Sigma^{(0)} = \frac{1}{N-1} \sum_{i=1}^{N} (\mathbf{x}_i - \mu^{(0)})(\mathbf{x}_i - \mu^{(0)})^\top, \tag{1}$$

where $\mu^{(0)}$ is the empirical mean and $\Sigma^{(0)}$ is the covariance matrix. In practice, $\epsilon > 0$ (set to $10^{-6}$ in our implementation) is a small regularization term added to $\Sigma^{(0)}$ to ensure numerical stability. At this step, $\mathcal{B}^{(0)}$ serves as the root of the tree hierarchy, capturing the global structure of the data.

**Step 2: Recursive Probabilistic Granular-ball Splitting.** To refine the density representation, each granular-ball $\mathcal{B}^{(l)}$ (parent ball) is recursively split into smaller components if the data within it exhibits sufficient structural variability. **This recursive process systematically decomposes the complex global data distribution into simpler, statistically validated ellipsoidal components.** The splitting strategy is governed by a dual-criterion rule, ensuring that the decision to split is both statistically sound and adaptive to the data structure.

**Splitting rule.** A granular-ball $\mathcal{B}^{(l)}$ is split into two child granular-balls $\mathcal{B}_1^{(l+1)}$ and $\mathcal{B}_2^{(l+1)}$ if and only if the following two criteria are satisfied:

$$\mathrm{BIC}(M_2) < \mathrm{BIC}(M_1), \quad \text{and} \quad \Delta \log \mathcal{L} > 0. \tag{2}$$

Here, $M_1$ represents the model of $\mathcal{B}^{(l)}$ as a single Gaussian model, while $M_2$ represents $\mathcal{B}^{(l)}$ as a two-component Gaussian Mixture Model (GMM). The splitting rule is designed to ensure that splitting occurs only when the two-component model offers a statistically significant improvement over the single Gaussian model.

The two criteria used in the splitting rule are defined as follows:

*i) Bayesian Information Criterion (BIC).* The Bayesian Information Criterion (Schwarz, 1978) evaluates the trade-off between model complexity and data fit. It is computed as:

$$\mathrm{BIC}(M) = -2 \log \mathcal{L}(M) + k \log N, \tag{3}$$

where $\mathcal{L}(M)$ is the likelihood of model $M$, $k$ is the number of free parameters in the model, and $N$ is the number of data points within $\mathcal{B}^{(l)}$. Crucially, **the BIC term acts as a statistical regularizer**, penalizing excessive complexity to ensure that splitting is driven by significant structural gains rather than local noise. A lower BIC indicates a better balance between model simplicity and accuracy. Splitting is preferred if the two-component model $M_2$ achieves a lower BIC than the single Gaussian model $M_1$.

*ii) Log-likelihood gain (LLG).* The log-likelihood gain (Fisher, 1922; Wilks, 1938) measures the improvement in data fit when replacing the single Gaussian model ($M_1$) with the two-component GMM ($M_2$). It is computed as:

$$\Delta \log \mathcal{L} = \sum_{i=1}^{N} \left[ \log p_{M_2}(\mathbf{x}_i) - \log p_{M_1}(\mathbf{x}_i) \right], \tag{4}$$

where $p_{M_1}(\mathbf{x}_i)$ and $p_{M_2}(\mathbf{x}_i)$ are the likelihoods of $\mathbf{x}_i$ under models $M_1$ and $M_2$, respectively. A positive $\Delta \log \mathcal{L}$ indicates that the two-component GMM ($M_2$) provides a better fit to the data.

If the splitting rule is satisfied, the parent granular-ball $\mathcal{B}^{(l)}$ is divided into two child granular-balls $\mathcal{B}_1^{(l+1)}$ and $\mathcal{B}_2^{(l+1)}$, each modeled as Gaussian components. After splitting, these child granular-balls inherit local data properties and are treated as new candidates for further splitting. This recursive process continues until no granular-ball satisfies the splitting criteria, resulting in a hierarchical representation of the data. However, splitting alone may leave some data points assigned to suboptimal granular-balls. To address this, a *dynamic reassignment step* is performed after each split.

**Step 3: Dynamic Reassignment.** To ensure consistency and improve local fit, each data point $\mathbf{x} \in \mathcal{X}$ is reassigned to the granular-ball that maximizes its log-likelihood:

$$\mathcal{B}^*(\mathbf{x}) = \arg \max_{\mathcal{B}_j} \log p_{\mathcal{B}_j}(\mathbf{x}), \tag{5}$$

where $p_{\mathcal{B}_j}$ is the Gaussian density parameterized by $(\mu_j, \Sigma_j)$ for granular-ball $\mathcal{B}_j$. Dynamic reassignment ensures that the hierarchical structure adapts to the evolving density distribution. By reallocating data points to the granular-balls that best represent their local characteristics, the framework maintains an accurate and adaptive representation of the data. After each round of splitting, dynamic reassignment is performed to refine the data distribution within the granular-balls.

This alternating process of splitting and reassignment continues iteratively until the hierarchical structure stabilizes. The resulting tree hierarchy encapsulates the global structure at the root, progressively refines intermediate levels, and captures fine-grained local patterns at the leaves. This hierarchical organization enables the framework to effectively balance global and local density estimation, providing both coarse and fine-grained insights into the data distribution.

The whole construction procedure is summarized in Algorithm 1 in Appendix B.

## 2.2 HIERARCHICAL ANOMALY SCORING

After constructing the hierarchical structure of probabilistic granular-balls, the PGBC framework represents the normal data distribution in a coarse-to-fine manner. Each level $l \in \{1, \dots, L\}$ in the hierarchy contains a set of granular-balls $\{\mathcal{B}_j^{(l)}\}_{j=1}^{K^{(l)}}$. Specifically, the hierarchical scoring aggregates information from levels $l = 1$ to $L$. The root level ($l = 0$) serves as a fallback representation: in the degenerate case where no splitting occurs ($L = 0$), the anomaly score is derived exclusively from the single global Gaussian at $l = 0$.

For a test sample $\mathbf{x}'$, an anomaly score is calculated by combining information from all levels of the hierarchy. At each level $l$, the anomaly score is based on the negative log-likelihood of the sample under the Gaussian components defined by the granular-balls at that level:

$$s^{(l)}(\mathbf{x}') = -\log \left( \sum_{j=1}^{K^{(l)}} \pi_j^{(l)} \cdot \mathcal{N}(\mathbf{x}' \mid \mu_j^{(l)}, \Sigma_j^{(l)}) \right), \tag{6}$$

where $\mathcal{N}(\mathbf{x}' \mid \mu_j^{(l)}, \Sigma_j^{(l)})$ is the Gaussian density defined by the $j$-th granular-ball, and $\pi_j^{(l)}$ represents the normalized weight of the $j$-th granular-ball. The weight $\pi_j^{(l)}$ reflects the relative importance of the granular-ball in the overall probabilistic distribution and is computed as:

$$\pi_j^{(l)} = \frac{n_j}{\sum_{k=1}^{K^{(l)}} n_k}, \tag{7}$$

where $n_j$ is the number of data points covered by granular-ball $\mathcal{B}_j^{(l)}$. This level-wise score $s^{(l)}(\mathbf{x}')$ evaluates how well the sample aligns with the normal data distribution at level $l$, with higher scores indicating greater deviation.

**Level Weighting with Entropy.**    To fully utilize the hierarchical structure, anomaly scores are aggregated across all levels. First, the scores at each level are normalized using min-max scaling to ensure consistency: $\tilde{s}^{(l)}(\mathbf{x}') = \frac{s^{(l)}(\mathbf{x}')-\min(s^{(l)})}{\max(s^{(l)})-\min(s^{(l)})}$, where $\min(s^{(l)})$ and $\max(s^{(l)})$ are computed from the training data. The final anomaly score is then calculated as a weighted sum of the normalized scores across all levels:

$$s_{\text{final}}(\mathbf{x}') = \sum_{l=1}^{L} w^{(l)} \cdot \tilde{s}^{(l)}(\mathbf{x}'), \tag{8}$$

where $w^{(l)}$ is the weight assigned to level $l$, reflecting its importance in the anomaly scoring process. Here, the weights $w^{(l)}$ are determined using an entropy-based scheme. The entropy of level $l$ quantifies its granularity and confidence in representing the data distribution: $H^{(l)} = -\sum_{j=1}^{K^{(l)}} \frac{n_j}{N} \log\left(\frac{n_j}{N}\right)$, where $n_j$ is the number of samples covered by granular-ball $\mathcal{B}_j^{(l)}$, and $N$ is the total number of samples. Higher entropy indicates a finer-grained partition of the data at that level. The weight for each level is then computed as: $w^{(l)} = \frac{H^{(l)}}{\sum_{l'=1}^{L} H^{(l')}}$. Since the root layer $(l=0)$ encompasses the entire dataset, it yields zero entropy and is inherently excluded from the weighted sum when $L \geq 1$. This allows the scoring mechanism to focus on the refined structural information provided by subsequent levels. Consequently, the root node $\mathcal{B}^{(0)}$ contributes to the final score only in the degenerate case $(L=0)$, as established earlier. By combining information across levels, the PGBC framework produces a hierarchical anomaly score that effectively captures deviations in both global and local patterns. The detailed pseudocode for the scoring process is provided in Algorithm 2 in Appendix B.

## 3 EXPERIMENT

**Setup.** We evaluate the proposed PGBC framework across three key tasks: (i) tabular anomaly detection, (ii) time series anomaly detection, and (iii) time series open-set recognition. For each task, PGBC is systematically compared against a diverse range of classical and deep learning baseline methods, including Isolation Forest (Liu et al., 2008), Local Outlier Factor (LOF) (Breunig et al., 2000), $k$-Nearest Neighbors (k-NN) (Peterson, 2009), AutoEncoder (Sakurada & Yairi, 2014), DeepSVDD (Ruff et al., 2018), DAGMM (Zong et al., 2018), the hierarchical Gaussian-mixture method HGAD (Yao et al., 2024), as well as granular-ball-based approaches GBMOD (Cheng et al., 2025) and GBDO (Su et al., 2025).

All experiments are conducted on a single NVIDIA RTX 4090 GPU, with fixed random seeds to ensure reproducibility. Model performance is evaluated using the Area Under the Curve (AUC) metric, averaged over five independent runs for robustness. Detailed descriptions of the datasets, encoder configurations, baseline implementations, and evaluation protocols are provided in Appendix C.

### 3.1 TABULAR ANOMALY DETECTION

**Overall Results.** We evaluate PGBC on 19 tabular datasets spanning manufacturing, cybersecurity, and medical diagnostics (Cheng et al., 2025). Features are normalized and models are trained only on normal samples. This evaluation tests PGBC's ability to model heterogeneous, static data across a range of anomaly ratios. Table 1 presents the mean AUC scores (averaged over five independent runs), with standard deviations provided in Appendix D. PGBC achieves the highest mean AUC on 13 out of 19 datasets and consistently ranks among the top-performing methods, highlighting its robustness and effectiveness across diverse data types.

**Analysis of Representative Scenarios.** *i) Typical Datasets:* On datasets with clear structures, such as the Bands and Tictac variants, PGBC demonstrates exceptional performance by achieving a perfect AUC score of 100.0%, demonstrating reliable performance even in cases where data exhibit no significant structural challenges. *ii) Imbalanced Datasets with Low Anomaly Ratios:* PGBC excels on datasets with extremely low anomaly ratios. On Thyroid (0.81% anomalies), it obtains the state-of-the-art AUC of 71.2%, surpassing all baselines including GBMOD (69.7%). Similarly, on Yeast (0.44% anomalies), it scores a perfect 100.0% AUC, matching the strongest competitors. This robustness stems from its hierarchical scoring strategy, which integrates information across multiple levels of granularity to mitigate majority-class bias in highly imbalanced settings.

Table 1: AUC results (%) on tabular anomaly detection tasks averaged over five runs. Standard deviations are omitted for brevity (see Appendix D for full results). **Bold** indicates the best performance, and underline indicates the second best. Abbreviations: AE = AutoEncoder, D.SV = DeepSVDD, DAG = DAGMM, GBM = GBMOD.

| Datasets | LOF | IForest | KNN | AE | D.SV | DAG | HGAD | GBM | GBDO | Ours |
|---|---|---|---|---|---|---|---|---|---|---|
| Abalone | 75.6 | 70.2 | 69.3 | **79.9** | 68.2 | 57.0 | 75.2 | 78.0 | 76.3 | 72.1 |
| Bands34 | 74.8 | 79.2 | 77.5 | 77.8 | 64.8 | 53.0 | 80.7 | 71.7 | 71.0 | **100.0** |
| Bands42 | 75.9 | 78.3 | 76.3 | 76.1 | 63.1 | 51.0 | 77.9 | 76.9 | 70.4 | **100.0** |
| Cardio | 82.6 | 85.9 | 80.5 | 76.5 | 77.3 | 70.0 | 75.8 | 86.5 | 66.4 | 84.0 |
| Ecoli | 85.1 | 84.1 | 87.6 | 88.1 | 86.8 | 40.0 | 88.3 | 85.1 | **89.8** | 89.4 |
| Iris | 91.6 | **100.0** | 97.8 | 93.8 | 83.4 | 99.0 | 98.5 | **100.0** | 74.8 | **100.0** |
| Musk | **100.0** | 96.9 | **100.0** | **100.0** | 99.7 | **100.0** | **100.0** | 91.1 | 25.9 | **100.0** |
| Pageblocks | 98.2 | 98.3 | 99.6 | 92.2 | 98.4 | 66.0 | 99.7 | 99.5 | 96.2 | **99.9** |
| Pendigits | 99.3 | 97.5 | 98.9 | 94.4 | 91.1 | 77.0 | 99.4 | 98.4 | 74.2 | **99.5** |
| Satellite | 84.5 | 79.4 | **87.6** | 80.6 | 83.3 | 80.0 | 82.3 | 85.5 | 80.4 | 83.1 |
| Sick35 | 73.7 | **89.3** | 89.1 | 85.2 | 83.8 | 57.0 | 88.8 | 89.1 | 83.6 | 87.4 |
| Sick72 | 61.2 | 81.4 | 83.5 | 81.4 | 77.3 | 56.0 | 82.2 | 79.1 | 79.6 | **87.3** |
| Sonar | 98.9 | 99.4 | 99.6 | 99.1 | 76.3 | 79.0 | 98.4 | 98.9 | 63.7 | **100.0** |
| Thyroid | 53.5 | 63.3 | 65.8 | 60.3 | 63.9 | 44.0 | 66.0 | 69.7 | 68.6 | **73.0** |
| Tictac12 | 97.6 | 98.6 | 93.6 | 84.6 | 75.9 | 54.0 | 97.0 | 96.8 | 60.7 | **100.0** |
| Tictac26 | 92.8 | 95.6 | 91.1 | 76.7 | 64.3 | 48.0 | 95.0 | 88.1 | 54.2 | **100.0** |
| Tictac32 | 92.2 | 95.4 | 93.1 | 76.1 | 64.1 | 46.0 | 93.2 | 86.3 | 55.1 | **100.0** |
| Waveform | 76.5 | 70.7 | 76.1 | 52.5 | 60.0 | 56.0 | 73.7 | 74.2 | 65.0 | **78.1** |
| Yeast | 99.1 | 99.7 | 99.4 | 99.6 | 99.9 | 43.0 | 99.7 | **100.0** | 99.2 | **100.0** |
| Average | 84.9 | 87.6 | 87.8 | 83.0 | 78.0 | 61.9 | 87.6 | 87.1 | 71.4 | **91.1** |

Table 2: AUC results (%) on time series anomaly detection tasks, averaged over five runs. Standard deviations are omitted here for clarity (full results in Appendix D). **Bold** indicates the best performance, and underline indicates the second best. AE = AutoEncoder, D.SV = DeepSVDD, DAG = DAGMM, GBM = GBMOD.

| Datasets | LOF | IForest | KNN | AE | D.SV | DAG | HGAD | GBM | GBDO | Ours |
|---|---|---|---|---|---|---|---|---|---|---|
| NAB Traffic | 80.4 | 78.6 | **91.2** | 79.6 | 83.4 | 59.2 | 84.3 | 80.6 | 77.5 | 85.4 |
| WSD WebService | 94.3 | 87.2 | 97.0 | 96.0 | 88.6 | 68.5 | 94.0 | 93.6 | 75.1 | **98.8** |
| SMD Facility | 93.3 | 93.4 | 97.1 | **98.5** | 94.5 | 76.4 | 96.4 | 95.1 | 91.1 | 97.9 |
| IOPS WebService | 91.0 | 81.7 | 85.5 | 77.8 | 79.3 | 62.7 | 87.0 | 79.3 | 44.4 | **96.4** |
| UCR Medical | **95.6** | 90.8 | 93.0 | 66.4 | 93.2 | 75.5 | 88.7 | 88.5 | 67.7 | 93.5 |
| YAHOO Synthetic | 71.3 | 77.2 | 88.6 | 44.8 | 71.4 | 48.1 | 67.2 | 67.0 | 88.9 | **99.8** |
| Average | 87.6 | 84.8 | 92.1 | 77.2 | 85.1 | 65.4 | 86.3 | 84.0 | 74.1 | **95.3** |

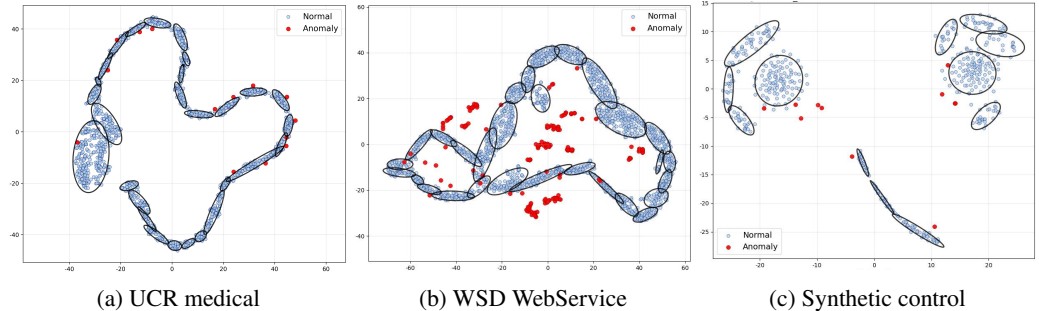

| (a) UCR medical | (b) WSD WebService | (c) Synthetic control |
|---|---|---|

Figure 3: Visualization of probabilistic granular-ball coverings on representative time series datasets. (a) UCR Medical, (b) WSD WebService, and (c) Synthetic Control.

**Summary.** The experimental results align with PGBC's design: covariance-aware, ellipsoidal modeling improves robustness on anisotropic data, while entropy-weighted hierarchical aggregation helps maintain adaptability across varying anomaly ratios.

## 3.2 TIME SERIES ANOMALY DETECTION

We evaluate PGBC on six benchmarks from the TSB-AD suite (Liu & Paparrizos, 2024), which contain real-world monitoring data with accurate anomaly labels. Sequences are segmented into

Table 3: AUC results (%) on time series open-set recognition tasks, averaged over five runs. Standard deviations are omitted for brevity (see Appendix D for detailed tables). **Bold** indicates the best performance, and underline indicates the second best. Abbreviations: AE = AutoEncoder, D.SV = DeepSVDD, DAG = DAGMM, GBM = GBMOD.

| Datasets | LOF | IForest | KNN | AE | D.SV | DAG | HGAD | GBM | GBDO | Ours |
|---|---|---|---|---|---|---|---|---|---|---|
| Adiac | 96.6 | 98.1 | 98.2 | 97.6 | **98.5** | 82.7 | 97.9 | 98.8 | 97.3 | **99.2** |
| CBF | 99.7 | 76.7 | 99.7 | 51.0 | 78.3 | 52.1 | 92.7 | 95.3 | 61.9 | **100.0** |
| Synthetic Control | 98.8 | 69.8 | **100.0** | 84.4 | 87.7 | 75.2 | 96.5 | 80.9 | 66.1 | 99.5 |
| SwedishLeaf | 88.8 | 95.1 | 96.3 | 84.6 | 94.5 | 73.9 | 97.2 | 95.7 | 88.6 | **99.7** |
| Trace | 87.3 | 89.6 | 94.9 | 62.9 | 66.9 | 80.0 | 97.1 | 73.1 | 72.2 | **99.6** |
| Average | 89.8 | 85.9 | 97.8 | 76.1 | 85.2 | 72.8 | 96.3 | 88.8 | 77.2 | **99.6** |

sliding windows and encoded by a lightweight CNN (Krizhevsky et al., 2012). This evaluation examines PGBC's behavior in noisy and dynamic environments. As shown in Table 2, we report mean AUC over five independent runs (std omitted here for clarity; full tables with std are in Appendix D), PGBC achieves the highest or second-highest AUC on all datasets. Furthermore, to demonstrate superiority over specialized deep time-series methods, we conducted an extended comparison against generative and flow-based models such as **OCFlow**, and **OCSVM**. Detailed results are provided in **Appendix E.1**, where PGBC consistently outperforms these baselines. Figure 3(a) and (b) further illustrate the learned granular-ball coverings on the UCR medical and WSD WebService datasets, where each contour corresponds to a $2\sigma$ boundary.

On datasets with extremely low anomaly ratios, such as SMD (0.64%) and Yahoo (0.28%), PGBC remains highly competitive. It achieves an AUC of 97.9% on SMD, slightly below AutoEncoder (98.5%) but higher than all other baselines, and reaches an AUC of 99.8% on Yahoo, surpassing the second-best method (88.9%) by over 10 points. These two datasets also span the extremes of sequence length in our benchmarks (Yahoo with 1,421 time steps and SMD with 22,700), underscoring PGBC's robustness across diverse temporal scales.

On WebService datasets such as WSD and IOPS, the underlying time series exhibit clear periodic patterns, which pose challenges for distance-based methods that cannot adapt to recurring fluctuations. PGBC achieves the highest AUC scores on both datasets: 98.8% on WSD (surpassing KNN's 97.0%) and 96.4% on IOPS (outperforming LOF's 91.0%). This demonstrates its ability to effectively align ellipsoidal granular-balls with intrinsic temporal structures. Figure 4 further illustrates this effect on the IOPS dataset: PGBC produces probabilistic boundaries where high anomaly scores closely coincide with ground-truth anomalies despite the strong periodicity.

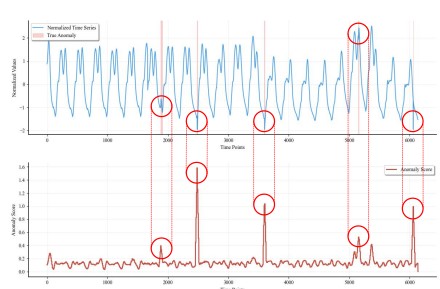

Figure 4: Visualization result of PGBC anomaly detection results on the IOPS WebService dataset.

Compared with tabular data, time series embeddings often lie on smooth manifold-like trajectories, where adjacent segments tend to follow intrinsic orientations. Therefore, modeling such anisotropic and continuous patterns requires flexible local structures. By orienting ellipsoidal granular-balls along these directions, PGBC adapts naturally to temporal data and maintains robustness under noisy and non-stationary conditions.

### 3.3 TIME SERIES OPEN-SET RECOGNITION

We further evaluate PGBC on five datasets from the UCR archive (Chen et al., 2015), repurposed for open-set recognition. The smallest class in each dataset is designated as anomalous; 80% of the remaining classes are used for training and 20% for testing, with an additional 10% anomalous samples injected into the test set. This setting explicitly evaluates the ability to recognize previously unseen categories under distributional shift.

Table 4: Summary of False Positive Rate (FPR %) and False Negative Rate (FNR %) on 19 tabular datasets. Full per-dataset results are provided in **Appendix D**.

| Metric | IForest | LOF | KNN | AE | D.SV | DAG | HGAD | GBM | GBDO | Ours |
|---|---|---|---|---|---|---|---|---|---|---|
| Avg FPR (%) ↓ | 3.14 | 3.19 | 2.54 | 3.31 | 3.88 | 4.29 | 2.40 | 3.07 | 4.57 | **2.36** |
| Avg FNR (%) ↓ | 52.8 | 50.1 | 46.5 | 57.2 | 75.7 | 84.9 | 38.7 | 50.2 | 77.7 | **35.9** |

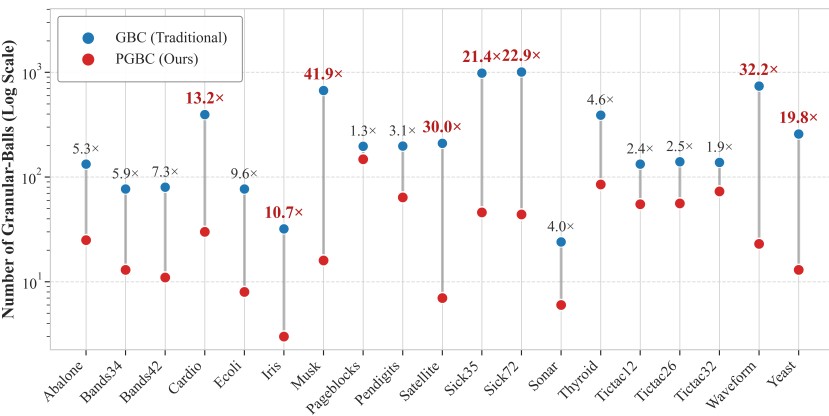

Figure 5: Comparison of model complexity (number of granular-balls) between traditional GBC and PGBC on 19 tabular datasets.

As shown in Table 3, PGBC achieves the best performance on four out of five datasets. PGBC achieves the highest AUC on four datasets—reaching 99.2% on Adiac, 100.0% on CBF, 99.7% on SwedishLeaf, and 99.6% on Trace. Even on the more challenging Synthetic Control dataset, PGBC attains 99.5% AUC, remaining competitive with the best baseline (KNN, 100.0%). These results highlight both robustness across sequence lengths (60–275) and reliable detection of unseen classes, enabled by entropy-weighted score aggregation.

As illustrated in Figure 3(c), PGBC adapts its ellipsoidal granular-balls to cover the interleaved clusters of the Synthetic Control dataset. Overall, these findings demonstrate PGBC's capability to generalize to open-set scenarios by forming precise decision boundaries. Specifically, the **ellipsoidal modeling** allows PGBC to wrap the complex manifolds of known classes tightly, minimizing the inclusion of void space where out-of-distribution samples might erroneously fall. Furthermore, the **entropy-weighted aggregation** ensures that anomaly scores reflect structural deviations across multiple granularities, preventing the model from overfitting to specific scales. This enables reliable rejection of unseen categories even under distributional shifts.

### 3.4 FALSE POSITIVE AND FALSE NEGATIVE RATE ANALYSIS

Balancing False Positive (FPR) and False Negative Rates (FNR) is critical for practical deployment. As shown in Table 4 (full details in **Appendix D**), PGBC achieves both the lowest average FPR (2.36%) and FNR (35.9%) across 19 datasets. Unlike methods such as DeepSVDD and GBDO, which achieve low FPR by being overly conservative (resulting in high FNR), PGBC's probabilistic ellipsoidal modeling constructs a precise decision boundary that minimizes false alarms without sacrificing recall, offering a superior trade-off for safety-critical applications.

### 3.5 EFFICIENCY AND MODEL COMPACTNESS

PGBC drastically reduces model complexity compared to traditional isotropic Granular-Ball Computing, requiring $1.3\times$ to $41.9\times$ fewer components (see Figure 5). As detailed in **Appendix I**, this compactness directly translates to computational efficiency. On an NVIDIA RTX 2060 GPU, PGBC achieves an average total runtime of **13.03s**, nearly $2\times$ faster than the deep probabilistic baseline HGAD (24.67s) and striking an optimal balance between expressiveness and speed.

Table 5: Average AUC (%) comparison on Visual Datasets. (a) Results on CIFAR-10 and FashionM-NIST. (b) Results on MVTec-AD. **Bold** indicates the best performance, and underline indicates the second best. Full details in Appendix F. Abbreviations: AE = AutoEncoder, D.SV = DeepSVDD, DAG = DAGMM, GBM = GBMOD, U.S=U-student.

(a) Comparison on CIFAR-10 and F-MNIST

| Datasets | D.SV | AE | GBM | KNN | HGAD | CutPaste | **Ours** |
|---|---|---|---|---|---|---|---|
| CIFAR-10 | 69.5 | 94.6 | 94.6 | 94.9 | 95.3 | 73.3 | **95.5** |
| FashionMNIST | 58.5 | 94.1 | 92.5 | 93.3 | 94.6 | 70.1 | **94.9** |

(b) Comparison on MVTec-AD

| Datasets | KNN | GBM | HGAD | CutPaste | U.S | PSVDD | **Ours** |
|---|---|---|---|---|---|---|---|
| MVTec-AD | 87.1 | 81.9 | 91.2 | 90.9 | 92.5 | 92.1 | **93.0** |

## 3.6 EXTENSION TO VISUAL ANOMALY DETECTION

To further validate the generalization of PGBC, we extended evaluation to visual domains using pre-trained feature embeddings (512-D ViT) following the ADBench protocol (Han et al., 2022). Our comparison includes feature-based baselines (DeepSVDD, AE, GBMOD, KNN, HGAD) on all datasets, and a comprehensive mixed benchmark on MVTec-AD incorporating both feature-based and image-based SOTA methods (e.g., CutPaste (Li et al., 2021), U-student (Bergmann et al., 2020), P-SVDD (Yi & Yoon, 2020)). Full experimental details are provided in **Appendix F**.

As summarized in Table 5, PGBC consistently achieves the highest average AUC across all benchmarks. On CIFAR-10 and FashionMNIST, PGBC outperforms all feature-based baselines, surpassing the second-best method HGAD (e.g., 95.5% vs. 95.3% on CIFAR-10). Crucially, on the challenging MVTec-AD industrial dataset, PGBC attains a leading AUC of **93.0%**, exceeding both feature-based competitors and specialized image-based SOTA methods such as U-student (92.5%) and P-SVDD (92.1%). This demonstrates that PGBC effectively leverages deep feature representations to model complex visual manifolds, outperforming even methods designed for raw pixel data. The results underscore the advantage of PGBC's probabilistic ellipsoidal modeling over rigid geometric approaches (e.g., DeepSVDD's 69.5% on CIFAR-10), confirming its robustness in high-dimensional spaces.

## 3.7 ABLATION STUDY SUMMARY

We rigorously validate PGBC's design via component-wise ablations (full details in **Appendix G** and **Appendix H**). Regarding **Construction**, results confirm that *dynamic reassignment* is critical for refining local fits, while the *BIC criterion* acts as an essential statistical regularizer—removing it leads to severe overfitting (e.g., component explosion from 25 to 668 on *Abalone*). Regarding **Scoring**, comparisons verify four key elements: (1) **Hierarchy**: Hierarchical aggregation consistently outperforms flat "Leaf-only" strategies; (2) **Metric**: Probabilistic log-likelihood significantly surpasses Euclidean distance (preventing $\approx$18% drop on *Bands34*); and (3) **Weighting & Normalization**: Entropy-based weighting and score normalization are proven essential for adaptively prioritizing informative levels and aligning scales across granularities.

## CONCLUSION

This work presented Probabilistic Granular-ball Computing (PGBC), a hierarchical framework for one-class data description. By leveraging ellipsoidal granular-balls, PGBC effectively aligns with the anisotropic geometry of data while requiring fewer granular-balls to represent complex distributions. Its recursive refinement process governed by statistical criteria ensures precise multi-scale data descriptions. Extensive experiments on tabular, time series, open-set, and visual benchmarks demonstrate consistent improvements and robustness over classical and deep learning baselines, underscoring its effectiveness. Future work could explore extending PGBC to streaming and online anomaly detection, enabling deployment in dynamic, real-world environments.

ETHICS STATEMENT

This work focuses solely on methodological advances in anomaly detection. All datasets employed are standard, publicly available benchmarks, and no human subjects, sensitive personal information, or proprietary data were involved. We therefore do not anticipate direct ethical risks arising from this study. Nonetheless, as with other anomaly detection techniques, the proposed method could be deployed in domains such as surveillance or security, where ethical considerations regarding fairness, privacy, and potential misuse must be carefully assessed by practitioners.

REPRODUCIBILITY STATEMENT

We ensure reproducibility by providing a precise mathematical description of the proposed PGBC framework in the main text, including its model formulation, splitting and reassignment rules, and scoring functions. Additional implementation details, covering datasets, preprocessing procedures, evaluation metrics, model configurations, and experimental settings, are included in the Appendix. To assess robustness, three main experimental tables report the mean and standard deviation over five independent runs. To support transparency and reproducibility, all source code and scripts will be released upon acceptance of this paper.

USE OF LARGE LANGUAGE MODELS

This paper utilized a large language model to aid in the refinement of writing and grammar. Specifically, the model was used for tasks such as rephrasing sentences for clarity, correcting typographical errors, and improving overall stylistic coherence. All content, research ideas, and core arguments remain the sole intellectual property of the authors. The use of the language model was strictly limited to polishing the written text.

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

## A    PRELIMINARIES: GRANULAR-BALL COMPUTING

We briefly review the basic concepts of traditional granular-ball computing (GBC), which form the foundation for our probabilistic extension.

**Definition 2** (Granular-ball). *A granular-ball $gb$ is defined by its center $\mathbf{c}$ and radius $\mathbf{r}$. For a set of objects $\mathbf{o}$ belonging to $gb$, these are typically determined as*

$$\mathbf{c} = \frac{1}{|gb|} \sum_{\mathbf{o} \in gb} \mathbf{o}, \quad \mathbf{r} = \max_{\mathbf{o} \in gb} \|\mathbf{o} - \mathbf{c}\|, \tag{9}$$

*where $\| \cdot \|$ denotes the Euclidean distance.*

Intuitively, a granular-ball represents a localized region in feature space summarizing a group of similar data points. Construction is usually performed in two stages. In the first stage, the dataset is coarsely partitioned into initial granular-balls via K-Means clustering, with the number of clusters often set to $\sqrt{n}$, where $n$ is the dataset size. Each cluster forms a ball with its centroid and maximum intra-cluster radius (Definition 2). This initialization provides a coarse but efficient covering of the data space.

The most widely used quality measure is the *distribution measure (DM)* (Definition 3), which quantifies the average dispersion of points within a ball.

**Definition 3** (Distribution Measure (DM)). *Given a granular-ball $gb$ with center $\mathbf{c}$ and data points $\{\mathbf{o}_i\}_{i=1}^{|gb|}$, the DM score is computed as*

$$DM(gb) = \frac{1}{|gb|} \sum_{\mathbf{o}_i \in gb} \|\mathbf{o}_i - \mathbf{c}\|. \tag{10}$$

A smaller $DM$ value indicates higher internal consistency. To refine representations, balls with large DM are recursively split using 2-means clustering. A split is accepted if the weighted DM of the resulting child balls is lower than that of the parent:

**Definition 4** (Refinement Criterion). *A granular-ball $gb$ is refined into $gb_1$ and $gb_2$ if*

$$DM_w = \frac{|gb_1|}{|gb|} DM(gb_1) + \frac{|gb_2|}{|gb|} DM(gb_2) < DM(gb), \tag{11}$$

*with both sub-balls containing at least $s_{\min}$ points (e.g., $s_{\min} = 8$).*

This recursive refinement continues until no further DM-reducing splits are possible, yielding a hierarchical partition of the dataset into compact granular-balls.

In anomaly detection, methods such as Granular-Ball Mean-Shift Outlier Detector (GB-MOD) (Cheng et al., 2025) and Granular-Ball Density Outlier (GBDO) (Su et al., 2025) apply GBC to identify outliers via heuristic criteria. However, these approaches are non-probabilistic, which limits their applicability in statistical analysis. Our PGBC framework extends GBC by associating each granular-ball with a probabilistic Gaussian component and building a hierarchical structure via statistically grounded refinement.

# B  ALGORITHMS

For clarity and reproducibility, we provide the pseudocode of the PGBC framework. Algorithm 1 describes the construction of the probabilistic granular-ball hierarchy, starting from a global Gaussian model and recursively applying statistical splitting criteria (BIC and log-likelihood improvement). The resulting tree-structured representation provides hierarchical coverage of the data distribution. Algorithm 2 presents the inference procedure for computing anomaly scores, where test samples are evaluated across all levels of the hierarchy, normalized, and aggregated with entropy-based weights.

---

**Algorithm 1:** Probabilistic Granular-ball Construction (PGBC)

---

**Input:** Training data $\mathcal{X} = \{x_1, \ldots, x_N\} \subset \mathbb{R}^d$; regularizer $\varepsilon$
**Output:** Tree-structured hierarchy $\mathcal{T}$ of probabilistic granular-balls

1 **Initialize:**
2 Compute global mean $\mu^{(0)}$ and covariance $\Sigma^{(0)} + \varepsilon I$
3 Create root ball $\mathcal{B}^{(0)}$ with $(\mu^{(0)}, \Sigma^{(0)})$, initialize tree $\mathcal{T}$
4 Queue $\mathcal{Q} \leftarrow [\mathcal{B}^{(0)}]$
5 **while** $\mathcal{Q}$ *not empty* **do**
6     Pop granular-ball $\mathcal{B}$ from $\mathcal{Q}$
7     Fit single Gaussian $\mathcal{M}_1$ and two-component GMM $\mathcal{M}_2$ on data in $\mathcal{B}$
8     Compute $BIC(\mathcal{M}_1)$, $BIC(\mathcal{M}_2)$, and $\Delta \log \mathcal{L} = \log L(\mathcal{M}_2) - \log L(\mathcal{M}_1)$
9     **if** $BIC(\mathcal{M}_2) < BIC(\mathcal{M}_1)$ ***and*** $\Delta \log \mathcal{L} > 0$ **then**
10         Split $\mathcal{B}$ into $\mathcal{B}_1, \mathcal{B}_2$ using GMM responsibilities
11         Add $\mathcal{B}_1, \mathcal{B}_2$ as children of $\mathcal{B}$ in $\mathcal{T}$
12         Push $\mathcal{B}_1, \mathcal{B}_2$ into $\mathcal{Q}$
        // Dynamic reassignment after each split
13         **foreach** $x_i \in \mathcal{X}$ **do**
14             Reassign $x_i$ to $\mathcal{B}^* = \arg\max_{\mathcal{B}_j} \log p_{\mathcal{B}_j}(x_i)$
15     **else**
16         Mark $\mathcal{B}$ as leaf
17 **return** $\mathcal{T}$

---

**Algorithm 2:** Hierarchical Anomaly Scoring

---

**Input:** Test sample $x' \in \mathbb{R}^d$; PGBC tree $\mathcal{T}$ with maximum depth $L$
**Output:** Final anomaly score $s_{\text{final}}(x')$

1 **if** $L = 0$ **then**
2     Let $\mathcal{B}^{(0)}$ be the root granular-ball with $(\mu^{(0)}, \Sigma^{(0)})$
3     **return** $-\log \mathcal{N}(x' \mid \mu^{(0)}, \Sigma^{(0)})$
4 **for** $l = 1$ **to** $L$ **do**
5     Let $\{\mathcal{B}_j^{(l)}\}_{j=1}^{K^{(l)}}$ be granular-balls at level $l$
6     Compute weights $\pi_j^{(l)} = \frac{n_j}{\sum_{k=1}^{K^{(l)}} n_k}$ where $n_j = |\mathcal{B}_j^{(l)}|$
7     Compute level-wise score: $s^{(l)}(x') = -\log\left(\sum_{j=1}^{K^{(l)}} \pi_j^{(l)} \cdot \mathcal{N}(x' \mid \mu_j^{(l)}, \Sigma_j^{(l)})\right)$
8     Compute level entropy: $H^{(l)} = -\sum_{j=1}^{K^{(l)}} \frac{n_j}{N} \log\left(\frac{n_j}{N}\right)$
9     Normalize $s^{(l)}(x')$ to $\tilde{s}^{(l)}(x')$ using training stats
10 Compute entropy weights $w^{(l)} = \frac{H^{(l)}}{\sum_{l'=1}^{L} H^{(l')}}$
11 Compute final score: $s_{\text{final}}(x') = \sum_{l=1}^{L} w^{(l)} \cdot \tilde{s}^{(l)}(x')$
12 **return** $s_{\text{final}}(x')$

---

## C  MORE EXPERIMENTAL SETTINGS

### C.1  DATASETS

Our experiments cover three dataset categories—tabular, benchmark time series (TSB-AD), and repurposed classification datasets (UCR)—spanning both static and temporal anomaly detection scenarios. Below we summarize sources, preprocessing steps, and key statistics; full dataset details are given in Tables 6–8.

**(1) Tabular datasets.**  We evaluate PGBC on 19 real-world tabular datasets collected from public repositories and prior anomaly-detection benchmarks; full per-dataset statistics are reported in Table 6. The collection covers diverse domains (manufacturing, cybersecurity, biology, and healthcare) and varies widely in scale (100–6,870 samples) and dimensionality (7–166 features), creating heterogeneous evaluation conditions. Anomaly proportions span from 0.44% (Yeast) to 31.64% (Satellite), providing both extremely sparse and relatively dense anomaly scenarios. For tabular inputs we apply per-dataset feature normalization (zero mean, unit variance); categorical attributes, when present, are one-hot encoded (see Appendix C.4 for encoder and preprocessing details). This benchmark stresses PGBC across low-anomaly-ratio cases and high-dimensional, anisotropic feature geometries—settings where covariance-aware modeling is particularly beneficial.

Table 6: Information of 19 tabular anomaly detection datasets.

| No. | Datasets | Samples | Features | Anom. (%) | Subject Area |
|---|---|---|---|---|---|
| 1 | Abalone | 4177 | 8 | 1.89 | Biology |
| 2 | Bands34 | 346 | 39 | 9.83 | Phys./Chem. |
| 3 | Bands42 | 354 | 39 | 11.86 | Phys./Chem. |
| 4 | Cardio | 1,688 | 21 | 1.95 | Health/Med. |
| 5 | Ecoli | 336 | 7 | 2.68 | Biology |
| 6 | Iris | 111 | 4 | 9.91 | Biology |
| 7 | Musk | 3,062 | 166 | 3.17 | Biology |
| 8 | Pageblocks | 5,171 | 10 | 4.99 | CS |
| 9 | Pendigits | 6,870 | 16 | 2.27 | CS |
| 10 | Satellite | 6,435 | 36 | 31.64 | Climate/Env. |
| 11 | Sick35 | 3,576 | 29 | 0.98 | Health/Med. |
| 12 | Sick72 | 3,613 | 29 | 1.99 | Health/Med. |
| 13 | Sonar | 107 | 60 | 9.35 | Phys./Chem. |
| 14 | Thyroid | 9,172 | 28 | 0.81 | Health/Med. |
| 15 | Tictac12 | 638 | 9 | 1.88 | Games |
| 16 | Tictac26 | 652 | 9 | 3.99 | Games |
| 17 | Tictac32 | 658 | 9 | 4.86 | Games |
| 18 | Waveform | 3,443 | 21 | 2.9 | Phys./Chem. |
| 19 | Yeast | 1,141 | 8 | 0.44 | Biology |

**(2) Benchmark time series datasets (TSB-AD).**  We use six real-world monitoring datasets from the TSB-AD benchmark, chosen for diversity in source (network traffic, system diagnostics, web services) and sequence length (see Table 7). Each dataset provides point-level anomaly labels; following common practice, we segment continuous sequences into fixed-length overlapping windows and encode each window with the same CNN encoder used across experiments (window size and step are listed in Appendix A.4, Table 8). Windowing preserves local temporal context while enabling batch training and fair comparisons across baseline encoders. Because anomalies are labeled at the point level, reported metrics correspond primarily to window-level detection; where applicable we additionally report sequence-level aggregated results (see main text).

**(3) Repurposed classification datasets (UCR).**  From the UCR archive we select five datasets and adapt them to a one-class / open-set evaluation protocol. For each dataset we designate the smallest class as the anomalous class and treat the remaining classes as normal. Normal samples are split

Table 7: Information of 6 time series anomaly detection datasets.

| No. | Datasets | Length | Anomalies | Anom. (%) |
|-----|----------|--------|-----------|-----------|
| 1 | NAB Traffic | 2,494 | 248 | 9.94 |
| 2 | WSD WebService | 15,403 | 203 | 1.32 |
| 3 | SMD Facility | 22,700 | 146 | 0.64 |
| 4 | IOPS WebService | 6,138 | 53 | 0.86 |
| 5 | UCR iMedical | 12,000 | 45 | 0.38 |
| 6 | YAHOO Synthetic | 1,421 | 4 | 0.28 |

80%/20% into training/testing; the test set is then augmented so that approximately 10% of test samples are anomalous (drawn from the designated anomalous class), creating a controlled open-set scenario. This protocol ensures anomalies are genuine class examples (not synthetic perturbations) while allowing consistent cross-dataset comparisons. Per-dataset statistics (sample counts, length, anomaly counts) are provided in Table 8.

Collectively, Tables 6–8 summarize dataset scales, modalities, and anomaly ratios, forming a comprehensive testbed for evaluating PGBC's robustness and generalization.

Table 8: Information of 5 open-set recognition (UCR) datasets.

| No. | Datasets | Samples | Length | Anomalies | Anom. (%) |
|-----|----------|---------|--------|-----------|-----------|
| 1 | Adiac | 778 | 176 | 17 | 10.00 |
| 2 | CBF | 633 | 128 | 13 | 9.49 |
| 3 | Synthetic Control | 511 | 60 | 11 | 9.91 |
| 4 | SwedishLeaf | 1,073 | 128 | 23 | 9.87 |
| 5 | Trace | 153 | 275 | 3 | 9.09 |

## C.2 BASELINES

We provide implementation details and hyperparameter settings of the baseline methods evaluated in our experiments. To evaluate model stability, we introduced randomness through random sub-sampling (Bootstrap) of the training data across five independent runs, applying this uniformly to all methods. All methods use consistent random seed control via `random_state=seed` to ensure reproducibility.

**Isolation Forest (Liu et al., 2008).**  Isolation Forest is a tree-based ensemble method that isolates anomalies by using random partitioning to create "isolation" for each data point. Anomalies, being few and different, are isolated in fewer steps than normal points. We use the scikit-learn implementation with `contamination` set to the true anomaly ratio, which is the proportion of outliers in the dataset.

**Local Outlier Factor (LOF) (Breunig et al., 2000).**  LOF is a density-based method that identifies anomalies by comparing the local density of a data point to the local densities of its neighbors. A point is considered an outlier if its local density is lower than that of its neighbors. We use a neighborhood size of `n_neighbors=20` and set `contamination` to the known anomaly ratio.

**k-Nearest Neighbors (kNN) (Peterson, 2009).**  The kNN method defines the anomaly score of a point as its distance to the $k$-th nearest neighbor. This score measures how far a point is from its local neighborhood. We set the number of neighbors to `n_neighbors=5`.

**AutoEncoder (Sakurada & Yairi, 2014).**  An AutoEncoder is a neural network with an encoder and a decoder, trained to reconstruct its input. By training only on normal data, it learns a compact representation of the normal distribution, so any point with a high reconstruction error is considered an anomaly. We use a feedforward encoder-decoder network trained to minimize reconstruction error. The network is configured with a `latent_dim=16`, trained for 50 epochs, with a `batch_size=64`, and a learning rate of `1e-3`. All models are trained exclusively on normal data.

**Deep SVDD (Ruff et al., 2018).**  Deep SVDD learns a compact representation of the normal data by training a deep neural network to map the data points into a feature space where they are enclosed within a minimal hypersphere. The radius of this hypersphere is learned during training with a $nu$ parameter that controls the trade-off between the volume of the sphere and the number of allowed outliers. We use a three-layer encoder with `hidden_dims=[128, 64, 32]` and an $output\_dim$=32. The hypersphere radius is learned with $nu$=0.1 using a soft-boundary objective. Pretraining is performed using an AutoEncoder with a `latent_dim=16`, trained for 50 epochs.

**Deep Autoencoding Gaussian Mixture Model (DAGMM) (Zong et al., 2018).**  Deep Autoencoding Gaussian Mixture Model (DAGMM) is a deep learning method that combines a deep autoencoder with a Gaussian Mixture Model (GMM) for unsupervised anomaly detection. It is designed to overcome the limitations of traditional two-stage methods by jointly optimizing the parameters of the autoencoder and GMM in an end-to-end fashion. The model utilizes the autoencoder to generate a low-dimensional representation and reconstruction error, which are then fed into the GMM to estimate the density of the normal data. We configure the autoencoder with a `latent_dim=3`, trained for `epochs=50`, a `batch_size=128`, and a learning rate of `1e-3`. The estimation network parameters are set with `n_components=4` for the GMM, and we use regularization parameters `lambda_energy=0.1` and `lambda_cov_diag=0.005` for the combined loss function.

**Hierarchical Gaussian Mixture Normalizing Flow Modeling (HGAD) (Yao et al., 2024).**  Hierarchical Gaussian Mixture Normalizing Flow Modeling (HGAD) is a novel method for unified anomaly detection that addresses the 'homogeneous mapping' issue in traditional normalizing flow-based models. It achieves this by leveraging a hierarchical probabilistic approach with two key components: inter-class Gaussian mixture modeling and intra-class mixed class centers learning. We extract features from `feature_levels=1` layer. The model is configured with `n_classes=1` for anomaly detection, and `n_intra_centers=5` for intra-class modeling. The loss function weights include `lambda_g=1.0, lambda_g_intra=1.0, lambda_z=1.0, lambda_e=0.1,`

and `lambda_mi=0.1`. The model is trained for `epochs=50` with a `batch_size=64` and a learning rate of `1e-3`.

**GBMOD (Cheng et al., 2025).** GBMOD, or Granular-Ball Mean-shift Outlier Detector, is a granular-ball based anomaly detection method addressing the limitations of traditional mean-shift techniques. It combines neighborhood-based density modeling with deep autoencoding. It uses granular-balls as anchors to guide the mean-shift process, which effectively avoids the influence of noisy points and improves efficiency. We use `k=10` nearest neighbors, `iteration_number=3`, and pretrain an AutoEncoder with the same configuration as described above.

**GBDO (Su et al., 2025)** GBDO, or Granular-ball Discrimination Outlier, is a method that improves density-based anomaly detection by operating at a granular-ball level. It detects anomalies based on local granular-ball density, which is calculated using the local reachability similarity among granular-balls. We set `k_neighbors=15` and `min_points_ratio=0.1` to control neighborhood and granularity.

### C.3 METRICS

**Area Under the Receiver Operating Characteristic Curve (AUC)** For all experiments, we use AUC as the primary evaluation metric. AUC is a widely adopted performance measure in anomaly detection, particularly suitable for highly imbalanced datasets where the number of anomalies is significantly smaller than normal instances. It quantifies the model's ability to distinguish between normal and anomalous samples across various classification thresholds, providing a comprehensive assessment independent of a specific decision threshold. To evaluate the robustness and reproducibility of our model and baselines on tabular datasets, we performed multiple independent runs and reported the average AUC with its standard deviation (AUC ± std).

**False Positive Rate (FPR)** In addition to AUC, we also evaluate our model's performance using the False Positive Rate (FPR). FPR is crucial in scenarios where the cost of misclassifying a normal instance as anomalous (i.e., a false alarm) is high. By analyzing both AUC and FPR, we provide a more holistic view of the model's performance, balancing its overall discriminative power with its precision in minimizing false alarms. We report the FPR for 19 tabular datasets.

**False Negative Rate (FNR)** Complementing FPR, we incorporate the False Negative Rate (FNR) to evaluate the model's sensitivity and reliability. FNR quantifies the proportion of actual anomalies that are incorrectly classified as normal (i.e., missed detections). This metric is paramount in safety-critical or high-risk applications—such as mechanical failure prediction or disease diagnosis—where failing to identify an anomaly can lead to severe consequences. By reporting FNR alongside FPR, we ensure that the model achieves a robust trade-off, verifying that a low false alarm rate is not obtained at the expense of missing actual threats.

## C.4 ENCODERS & COMPARISONS

To ensure fair comparisons across all baselines and to provide a unified vector input for different data types, we employ a feature encoder to transform our data. For tabular datasets, we apply standard feature normalization without additional preprocessing. For time series datasets, a sliding window mechanism is used to segment sequences into fixed-length windows. These windows are then encoded by a lightweight Convolutional Neural Network (CNN) into a fixed-dimensional vector. The detailed architecture and key parameters for our CNN encoder and the time series preprocessing steps are summarized in Table 10.

While our proposed PGBC model is agnostic to the choice of encoder—meaning its core architecture can operate on any fixed-dimensional vector representation—the quality of learned features significantly affects downstream detection accuracy. To investigate this impact, we conducted a comparative study using five common encoder types: MLP, CNN, LSTM, Transformer, and ResNet. These encoders were evaluated on four representative datasets from the TSB-AD and UCR benchmark suites, each representing a distinct feature learning paradigm.

Our experimental findings, with complete results summarized in Table 9, show that the lightweight CNN encoder consistently achieves strong performance, ranking first or second in AUC on all four datasets. Considering its computational efficiency and strong empirical performance, we adopt the CNN architecture as our default feature encoder for all subsequent experiments. This design ensures a fair and robust evaluation across all baselines while maintaining a practical and effective model architecture.

Table 9: Ablation study on feature encoders. AUC results (%) of PGBC using different encoder architectures on selected datasets from TSB-AD and UCR. Bold indicates the best performance. Trans = Transformer.

| Dataset | CNN | MLP | LSTM | Trans | ResNet |
|---|---|---|---|---|---|
| Adiac | 99.6 | **100.0** | 97.8 | 98.7 | 99.5 |
| Synthetic Control | **100.0** | 80.0 | 98.2 | 98.7 | **100.0** |
| WSD WebService | 99.5 | 99.4 | 77.7 | **100.0** | **100.0** |
| YAHOO Synthetic | **100.0** | **100.0** | 63.8 | 75.9 | 99.2 |

Table 10: Configuration of key parameters. Details for the CNN encoder architecture and time series preprocessing steps used in the experiments.

| Parameter | Value | Description |
|---|---|---|
| *CNN Encoder Architecture* | | |
| Convolutional Layers | 3 | 3-layer convolutional encoder, mirrored by 3-layer transposed convolutional decoder |
| Kernel Size | 5 | 5×1 kernel for all (transposed) convolutional layers |
| Stride | 2 | Stride of 2 for downsampling/upsampling in all layers |
| Padding | 2 | Padding of 2 for all (transposed) convolutional layers |
| Channel Progression | 1→16→32→64 | Encoder channel increase across layers |
| Activation Function | ReLU | ReLU activation for all intermediate layers |
| Latent Dimension | 16 | Default latent space dimension, configurable via `ae_latent_dim` |
| *Time Series Processing* | | |
| Window Size | 32 | Sliding window size for time series segmentation |
| Window Step | 8 | Step size for sliding window |
| AE Latent Dimension | 16 | Autoencoder latent space dimension |
| AE Epochs | 50 | Number of training epochs for autoencoder |
| AE Batch Size | 64 | Batch size for autoencoder training |
| AE Learning Rate | $10^{-3}$ | Learning rate for autoencoder training |
| Device | Auto | GPU/CPU, automatically selected |

## D  MORE RESULTS

In this section, we present a more comprehensive view of our experimental results by including the standard deviation (std) of the AUC scores, which reflects model stability.

**Tabular anomaly detection.**  The full results are detailed in Table 11. A key observation is that PGBC, in addition to achieving the highest average AUC, also demonstrates superior stability with the second-lowest average standard deviation among all baselines. This indicates that the high performance of our method is consistently reproducible across different data splits, a crucial factor for reliable real-world applications.

Table 11: Full AUC results (%) with mean and standard deviation on 19 tabular datasets. Results are averaged over five independent runs. Bold indicates the best performance. Abbreviations: AE = AutoEncoder, D.SV = DeepSVDD, DAG = DAGMM, GBM = GBMOD.

| Datasets | LOF | IForest | KNN | AE | D.SV | DAG | HGAD | GBM | GBDO | Ours |
|---|---|---|---|---|---|---|---|---|---|---|
| Abalone | 75.6 | 70.2 | 69.3 | **79.9** | 68.2 | 57.0 | 75.2 | 78.0 | 76.3 | 72.1 |
| | ±1.2 | ±1.1 | ±1.0 | ±1.3 | ±1.9 | ±4.3 | ±2.5 | ±1.3 | ±1.0 | ±1.1 |
| Bands34 | 74.8 | 79.2 | 77.5 | 77.8 | 64.8 | 53.0 | 80.7 | 71.7 | 71.0 | **100.0** |
| | ±0.8 | ±1.3 | ±2.7 | ±1.2 | ±5.6 | ±6.8 | ±1.6 | ±0.4 | ±4.4 | ±0.0 |
| Bands42 | 75.9 | 78.3 | 76.3 | 76.1 | 63.1 | 51.0 | 77.9 | 76.9 | 70.4 | **100.0** |
| | ±0.8 | ±1.3 | ±2.7 | ±1.2 | ±5.6 | ±5.6 | ±1.4 | ±0.4 | ±4.4 | ±0.0 |
| Cardio | 82.6 | 85.9 | 80.5 | 76.5 | 77.3 | 70.0 | 75.8 | 86.5 | 66.4 | 84.0 |
| | ±1.0 | ±1.8 | ±1.6 | ±2.5 | ±8.7 | ±7.5 | ±2.0 | ±0.7 | ±1.4 | ±0.8 |
| Ecoli | 85.1 | 84.1 | 87.6 | 88.1 | 86.8 | 40.0 | 88.3 | 85.1 | **89.8** | 89.4 |
| | ±0.5 | ±0.9 | ±1.1 | ±2.0 | ±1.5 | ±33.5 | ±2.1 | ±0.2 | ±0.5 | ±0.4 |
| Iris | 91.6 | **100.0** | 97.8 | 93.8 | 83.4 | 99.0 | 98.5 | **100.0** | 74.8 | **100.0** |
| | ±3.8 | ±0.0 | ±1.0 | ±10.0 | ±13.0 | ±1.7 | ±0.9 | ±0.0 | ±2.6 | ±0.0 |
| Musk | **100.0** | 96.9 | **100.0** | **100.0** | 99.7 | **100.0** | **100.0** | 91.1 | 25.9 | **100.0** |
| | ±0.0 | ±1.6 | ±0.0 | ±0.0 | ±0.6 | ±0.5 | ±0.0 | ±1.7 | ±1.5 | ±0.0 |
| Pageblocks | 98.2 | 98.3 | 99.6 | 92.2 | 98.4 | 66.0 | 99.7 | 99.5 | 96.2 | **99.9** |
| | ±0.1 | ±0.2 | ±0.1 | ±0.8 | ±0.1 | ±23.4 | ±0.1 | ±0.1 | ±0.2 | ±0.0 |
| Pendigits | 99.3 | 97.5 | 98.9 | 94.4 | 91.1 | 77.0 | 99.4 | 98.4 | 74.2 | **99.5** |
| | ±0.1 | ±0.8 | ±0.3 | ±3.3 | ±6.0 | ±17.9 | ±0.3 | ±0.6 | ±2.6 | ±0.6 |
| Satellite | 84.5 | 79.4 | **87.6** | 80.6 | 83.3 | 80.0 | 82.3 | 85.5 | 80.4 | 83.1 |
| | ±0.2 | ±1.3 | ±0.1 | ±0.6 | ±5.0 | ±4.9 | ±0.3 | ±0.4 | ±0.2 | ±1.5 |
| Sick35 | 73.7 | **89.3** | 89.1 | 85.2 | 83.8 | 57.0 | 88.8 | 89.1 | 83.6 | 87.4 |
| | ±1.3 | ±1.2 | ±0.8 | ±1.4 | ±2.7 | ±15.1 | ±0.4 | ±0.4 | ±0.6 | ±3.3 |
| Sick72 | 61.2 | 81.4 | 83.5 | 81.4 | 77.3 | 56.0 | 82.2 | 79.1 | 79.6 | **87.3** |
| | ±1.7 | ±2.3 | ±0.4 | ±2.1 | ±2.6 | ±4.8 | ±1.1 | ±0.2 | ±1.1 | ±3.6 |
| Sonar | 98.9 | 99.4 | 99.6 | 99.1 | 76.3 | 79.0 | 98.4 | 98.9 | 63.7 | **100.0** |
| | ±0.1 | ±0.5 | ±0.3 | ±0.2 | ±8.7 | ±12.6 | ±0.7 | ±0.3 | ±8.3 | ±0.0 |
| Thyroid | 53.5 | 63.3 | 65.8 | 60.3 | 63.9 | 44.0 | 66.0 | 69.7 | 68.6 | **73.0** |
| | ±0.9 | ±1.9 | ±0.7 | ±3.3 | ±6.5 | ±10.9 | ±3.2 | ±0.7 | ±0.3 | ±4.8 |
| Tictac12 | 97.6 | 98.6 | 93.6 | 84.6 | 75.9 | 54.0 | 97.0 | 96.8 | 60.7 | **100.0** |
| | ±0.5 | ±0.6 | ±3.5 | ±6.4 | ±6.4 | ±8.1 | ±1.5 | ±0.9 | ±3.0 | ±0.0 |
| Tictac26 | 92.8 | 95.6 | 91.1 | 76.7 | 64.3 | 48.0 | 95.0 | 88.1 | 54.2 | **100.0** |
| | ±0.9 | ±0.9 | ±1.8 | ±5.7 | ±6.5 | ±4.5 | ±2.6 | ±0.8 | ±0.8 | ±0.0 |
| Tictac32 | 92.2 | 95.4 | 93.1 | 76.1 | 64.1 | 46.0 | 93.2 | 86.3 | 55.1 | **100.0** |
| | ±1.3 | ±0.9 | ±0.8 | ±5.9 | ±9.8 | ±8.3 | ±2.7 | ±2.9 | ±0.7 | ±0.0 |
| Waveform | 76.5 | 70.7 | 76.1 | 52.5 | 60.0 | 56.0 | 73.7 | 74.2 | 65.0 | **78.1** |
| | ±0.5 | ±3.9 | ±0.6 | ±3.3 | ±3.6 | ±13.6 | ±2.3 | ±0.2 | ±1.2 | ±0.0 |
| Yeast | 99.1 | 99.7 | 99.4 | 99.6 | 99.9 | 43.0 | 99.7 | **100.0** | 99.2 | **100.0** |
| | ±0.0 | ±0.0 | ±0.2 | ±0.4 | ±0.1 | ±29.6 | ±0.1 | ±0.0 | ±1.1 | ±0.0 |
| Average | 84.9 | 87.6 | 87.8 | 83.0 | 78.0 | 61.9 | 87.6 | 87.1 | 71.4 | **91.1** |
| | ±13.3 | ±11.4 | ±10.5 | ±12.4 | ±12.8 | ±17.7 | ±10.7 | ±9.8 | ±16.2 | ±10.0 |

**Time series anomaly detection.** The complete results for the six TSB-AD benchmarks are shown in Table 12. PGBC not only achieves the highest or second-highest mean AUC across all datasets, but also maintains low variance compared with deep baselines such as AutoEncoder and DeepSVDD. This demonstrates that the probabilistic granular-ball hierarchy provides stable anomaly detection under the noisy and dynamic conditions typical of real-world monitoring data.

Table 12: Full AUC results (%) with mean and standard deviation on 6 time series datasets. Results are averaged over five independent runs. Bold indicates the best performance.

| Datasets | LOF | IForest | KNN | AE | D.SV | DAG | HGAD | GBM | GBDO | Ours |
|---|---|---|---|---|---|---|---|---|---|---|
| NAB Traffic | 80.4 | 78.6 | **91.2** | 79.6 | 83.4 | 59.2 | 84.3 | 80.6 | 77.5 | 85.4 |
| | ±0.4 | ±0.6 | ±0.9 | ±2.3 | ±2.0 | ±5.9 | ±1.7 | ±0.7 | ±0.3 | ±3.8 |
| WSD WebService | 94.3 | 87.2 | 97.0 | 96.0 | 88.6 | 68.5 | 94.0 | 93.6 | 75.1 | **98.8** |
| | ±0.2 | ±0.9 | ±0.3 | ±0.2 | ±1.9 | ±13.5 | ±1.1 | ±0.6 | ±0.5 | ±0.4 |
| SMD Facility | 93.3 | 93.4 | 97.1 | **98.5** | 94.5 | 78.3 | 96.4 | 95.1 | 91.1 | 97.9 |
| | ±0.7 | ±0.3 | ±0.6 | ±0.1 | ±1.6 | ±6.9 | ±1.0 | ±1.5 | ±0.7 | ±0.0 |
| IOPS WebService | 91.0 | 81.7 | 85.5 | 77.8 | 79.3 | 62.7 | 87.0 | 79.3 | 44.4 | **96.4** |
| | ±0.5 | ±2.4 | ±0.4 | ±14.7 | ±4.5 | ±4.0 | ±3.4 | ±4.5 | ±1.2 | ±0.1 |
| UCR Medical | **95.6** | 90.8 | 93.0 | 66.4 | 93.2 | 75.5 | 88.7 | 88.5 | 67.7 | 93.5 |
| | ±0.7 | ±3.0 | ±0.3 | ±1.2 | ±5.3 | ±9.1 | ±2.3 | ±1.2 | ±0.3 | ±0.1 |
| YAHOO Synthetic | 71.3 | 77.2 | 88.6 | 44.8 | 71.4 | 48.1 | 67.2 | 67.0 | 88.9 | **99.8** |
| | ±7.4 | ±2.1 | ±4.9 | ±8.8 | ±21.9 | ±12.5 | ±29.8 | ±8.8 | ±0.0 | ±0.0 |
| Average | 87.6 | 84.8 | 92.1 | 77.2 | 85.1 | 65.4 | 86.3 | 84.0 | 74.1 | **95.3** |
| | ±1.7 | ±1.6 | ±1.2 | ±4.6 | ±6.2 | ±8.7 | ±6.6 | ±2.9 | ±0.5 | ±0.7 |

**Time series open-set recognition.** For the UCR-based open-set recognition task, the full results are summarized in Table 13. Across the five datasets, PGBC achieves competitive or superior mean AUC while keeping consistently small standard deviations. This stability highlights the robustness of the entropy-weighted hierarchical scoring mechanism, which balances coarse and fine representations and prevents overfitting to specific runs. Overall, PGBC generalizes well under distributional shifts, as evidenced by both high average accuracy and low variability across repeated experiments.

Table 13: Full AUC results (%) with mean and standard deviation on 5 open-set recognition tasks. Results are averaged over five independent runs. Bold indicates the best performance.

| Datasets | LOF | IForest | KNN | AE | D.SV | DAG | HGAD | GBM | GBDO | Ours |
|---|---|---|---|---|---|---|---|---|---|---|
| Adiac | 96.6 | 98.1 | 98.2 | 97.6 | **98.5** | 82.7 | 97.9 | 98.8 | 97.3 | **99.2** |
| | ±0.7 | ±0.5 | ±0.1 | ±0.2 | ±0.3 | ±14.9 | ±0.2 | ±0.6 | ±0.0 | ±0.2 |
| CBF | 99.7 | 76.7 | 99.7 | 51.0 | 78.3 | 52.1 | 92.7 | 95.3 | 61.9 | **100.0** |
| | ±0.1 | ±3.9 | ±0.2 | ±3.7 | ±14.0 | ±17.5 | ±2.9 | ±2.4 | ±0.4 | ±0.0 |
| Synthetic Control | 98.8 | 69.8 | **100.0** | 84.4 | 87.7 | 75.2 | 96.5 | 80.9 | 66.1 | 99.5 |
| | ±0.4 | ±4.8 | ±0.1 | ±4.6 | ±3.3 | ±18.0 | ±1.6 | ±2.9 | ±1.5 | ±1.1 |
| SwedishLeaf | 88.8 | 95.1 | 96.3 | 84.6 | 94.5 | 73.9 | 97.2 | 95.7 | 88.6 | **99.7** |
| | ±1.1 | ±0.6 | ±0.1 | ±4.6 | ±3.4 | ±16.7 | ±0.9 | ±0.5 | ±0.1 | ±0.1 |
| Trace | 87.3 | 89.6 | 94.9 | 62.9 | 66.9 | 80.0 | 97.1 | 73.1 | 72.2 | **99.6** |
| | ±6.1 | ±2.0 | ±5.5 | ±23.1 | ±35.4 | ±34.2 | ±4.2 | ±11.8 | ±0.0 | ±1.0 |
| Average | 89.8 | 85.9 | 97.8 | 76.1 | 85.2 | 72.8 | 96.3 | 88.8 | 77.2 | **99.6** |
| | ±2.9 | ±2.4 | ±1.6 | ±7.2 | ±18.1 | ±20.3 | ±1.9 | ±3.6 | ±0.4 | ±0.5 |

**False Positive Rate (FPR) Results.** In this section, we provide the complete false positive rate (FPR) results for all 19 tabular datasets, as detailed in Table 14. This comprehensive view complements the main text and highlights the superior reliability of PGBC in controlling false alarms across diverse data distributions. As shown in the table, PGBC achieves the lowest average FPR of **2.36%**. A closer inspection reveals that methods relying on isotropic boundaries (e.g., DeepSVDD, 3.88%) or simple geometric distances (e.g., GBDO, 4.57%) tend to generate higher false alarms, particularly on datasets with complex cluster shapes. In contrast, PGBC's advantage is most pronounced in these scenarios, as its probabilistic ellipsoidal granular-balls can stretch to fit the normal data tightly without encompassing the surrounding void space, thereby minimizing the risk of misclassifying normal boundary points as anomalies.

Table 14: Full False Positive Rate (FPR %) results on 19 tabular datasets. Lower values indicate fewer false alarms. Abbreviations: AE = AutoEncoder, D.SV = DeepSVDD, DAG = DAGMM, GBM = GBMOD.

| Datasets | LOF | IForest | KNN | AE | D.SV | DAG | HGAD | GBM | GBDO | Ours |
|---|---|---|---|---|---|---|---|---|---|---|
| Abalone | 4.56 | 4.73 | **4.37** | 4.61 | 4.88 | 5.05 | 4.66 | 4.71 | 5.05 | 4.64 |
| Bands34 | 4.17 | 5.13 | 2.88 | 4.49 | 5.13 | 5.45 | **0.00** | 4.17 | 6.09 | **0.00** |
| Bands42 | 3.85 | 4.81 | 2.88 | 3.85 | 5.13 | 5.45 | **0.00** | 3.21 | 3.21 | **0.00** |
| Cardio | 4.53 | 4.35 | **3.93** | 4.71 | 4.53 | 4.65 | 4.35 | 4.05 | 4.95 | 4.65 |
| Ecoli | **3.06** | **3.06** | **3.06** | **3.06** | **3.06** | 5.20 | **3.06** | **3.06** | 3.98 | **3.06** |
| Iris | **0.00** | 2.00 | **0.00** | 3.00 | 2.00 | 2.00 | **0.00** | **0.00** | 5.00 | **0.00** |
| Musk | **1.92** | **1.92** | **1.92** | **1.92** | **1.92** | 2.02 | **1.92** | **1.92** | 5.19 | **1.92** |
| Pageblocks | 1.61 | 0.83 | 0.49 | 0.55 | 1.04 | 3.91 | 0.57 | 1.14 | 2.06 | **0.22** |
| Pendigits | 3.66 | **2.80** | **2.80** | 3.71 | 4.71 | 4.72 | **2.80** | 3.05 | 4.54 | **2.80** |
| Satellite | **0.00** | **0.00** | **0.00** | 0.02 | 1.20 | 0.23 | **0.00** | **0.00** | 0.43 | **0.00** |
| Sick35 | 4.94 | 4.94 | 4.63 | 4.83 | 4.74 | 5.06 | 4.80 | 4.66 | 5.06 | **4.57** |
| Sick72 | 5.00 | 4.91 | **4.57** | 4.77 | 4.74 | 5.17 | 4.83 | 4.69 | 5.06 | 4.80 |
| Sonar | 1.03 | **0.00** | **0.00** | **0.00** | 2.06 | 2.06 | 1.03 | 1.03 | 5.15 | **0.00** |
| Thyroid | 5.02 | 4.96 | 4.95 | 4.98 | 4.95 | 5.05 | 4.98 | 4.90 | 5.06 | **4.89** |
| Tictac12 | **3.19** | 3.35 | **3.19** | 3.99 | 4.79 | 5.11 | **3.19** | 3.35 | 5.75 | **3.19** |
| Tictac26 | 2.24 | 2.24 | **1.12** | 2.40 | 4.63 | 4.95 | 1.28 | 3.19 | 4.63 | **1.12** |
| Tictac32 | 1.44 | 1.76 | **0.16** | 2.24 | 4.79 | 5.27 | 0.32 | 2.40 | 5.75 | **0.16** |
| Waveform | 4.82 | 4.10 | **2.57** | 5.12 | 4.67 | 5.00 | 3.20 | 4.07 | 5.12 | 4.07 |
| Yeast | **4.67** | **4.67** | **4.67** | **4.67** | 4.75 | 5.11 | **4.67** | **4.67** | 4.75 | 4.75 |
| Average | 3.14 | 3.19 | 2.54 | 3.31 | 3.88 | 4.29 | 2.40 | 3.07 | 4.57 | **2.36** |

**False Negative Rate (FNR) Results.** While a low FPR is essential for system usability, a low False Negative Rate (FNR) is critical for safety, as missing a genuine anomaly can have severe consequences. The full FNR results in Table 15 reveal a significant trade-off made by many baselines. PGBC achieves the **lowest average FNR of 35.9%**, edging out the second-best HGAD (38.7%) and drastically surpassing DeepSVDD (75.7%) and GBDO (77.7%). The extremely high FNRs of these methods suggest they achieve reasonable FPRs only by being *overly conservative*—resulting in an overly loose decision boundary that fails to capture a significant portion of anomalies. In contrast, when combined with the FPR analysis in Section 3.4 (where PGBC also leads with 2.36%), these results confirm that PGBC does not trivially trade off precision for recall. Instead, its hierarchical probabilistic structure provides a fundamentally more accurate description of the data, successfully identifying diverse anomalies that other methods miss.

Table 15: Full False Negative Rate (FNR %) results on 19 tabular datasets. Lower values indicate fewer missed anomalies. Abbreviations: AE = AutoEncoder, D.SV = DeepSVDD, DAG = DAGMM, GBM = GBMOD.

| Datasets | IForest | LOF | KNN | AE | D.SV | DAG | HGAD | GBM | GBDO | Ours |
|---|---|---|---|---|---|---|---|---|---|---|
| Abalone | 72.2 | 81.0 | 74.7 | 86.1 | 86.1 | 91.1 | 74.7 | 88.6 | 83.5 | **75.9** |
| Bands34 | 85.3 | 94.1 | 97.1 | 79.4 | 82.4 | 97.1 | **47.1** | 79.4 | 97.1 | **47.1** |
| Bands42 | 85.7 | 92.9 | 95.2 | **63.3** | 97.6 | 97.6 | 57.1 | 88.1 | 97.6 | 57.1 |
| Cardio | 69.7 | 60.6 | 69.7 | 66.7 | 81.8 | 100.0 | 60.6 | 57.6 | 84.8 | **48.5** |
| Ecoli | **22.2** | **22.2** | **22.2** | 88.9 | 44.4 | 100.0 | 11.1 | **22.2** | **22.2** | **22.2** |
| Iris | 45.5 | 63.6 | 45.5 | 45.5 | 100.0 | 81.8 | 45.5 | 45.5 | 100.0 | **45.5** |
| Musk | **0.0** | **0.0** | **0.0** | **0.0** | 18.6 | 13.4 | **0.0** | **0.0** | 84.5 | **0.0** |
| Pageblocks | 30.2 | 15.5 | 10.9 | 22.5 | 39.1 | 70.9 | 12.0 | 22.5 | 38.8 | **3.1** |
| Pendigits | 37.2 | **0.0** | **0.0** | 92.3 | 100.0 | 94.9 | **0.0** | 17.9 | 64.7 | **0.0** |
| Satellite | 84.2 | 84.2 | 84.2 | 84.2 | 84.2 | 85.5 | 84.2 | 84.2 | 94.7 | **84.2** |
| Sick35 | 88.6 | 88.6 | 71.4 | 74.3 | 91.4 | 100.0 | 71.4 | 71.4 | 85.7 | **45.7** |
| Sick72 | 94.4 | 90.3 | 86.1 | 83.3 | 95.8 | 100.0 | 84.7 | 86.1 | 93.1 | **72.2** |
| Sonar | 50.0 | **40.0** | **40.0** | 50.0 | 70.0 | 90.0 | **40.0** | 50.0 | 70.0 | **40.0** |
| Thyroid | 97.3 | 89.2 | 89.2 | 89.2 | 83.8 | 97.3 | 91.9 | 82.4 | 98.6 | **79.7** |
| Tictac12 | **0.0** | 8.3 | 8.3 | **0.0** | 83.3 | 100.0 | 8.3 | **0.0** | 91.7 | **0.0** |
| Tictac26 | 26.9 | 26.9 | 11.5 | 30.8 | 100.0 | 96.2 | **3.8** | 42.3 | 84.6 | **0.0** |
| Tictac32 | 25.0 | 31.2 | 9.4 | 50.0 | 90.6 | 96.9 | 9.4 | 50.0 | 90.6 | **0.0** |
| Waveform | 88.0 | 64.0 | 69.0 | 81.0 | 89.0 | 100.0 | **33.0** | 66.0 | 95.0 | 61.0 |
| Yeast | **0.0** | **0.0** | **0.0** | **0.0** | **0.0** | **0.0** | **0.0** | **0.0** | **0.0** | **0.0** |
| Average | 52.8 | 50.1 | 46.5 | 57.2 | 75.7 | 84.9 | 38.7 | 50.2 | 77.7 | **35.9** |

# E  EXTENDED BASELINES AND METRICS

In this section, we provide detailed comparisons with additional baselines requested by the reviewers to demonstrate the superiority of PGBC across different task domains.

## E.1  ADDITIONAL BASELINES ON TIME-SERIES ANOMALY DETECTION

To demonstrate the superiority of PGBC against a broader range of state-of-the-art methods in time-series anomaly detection, we incorporated additional baselines: **OCFlow** (Maziarka et al., 2022) and **OCSVM**. We evaluated them on our six Time-Series Benchmark datasets (TSB-AD) to compare our approach against representative normalizing flow and kernel-based methods.

**Implementation Details.** To ensure a fair comparison, we configured each baseline with key hyper-parameters optimized for the task. For **OCFlow**, a flow-based model, we utilized 8 coupling layers (`n_couplings=8`, `n_layers=4`) with a hidden dimension of 512 and variable Jacobian determinants (`det_type='var'`) to map data to a latent hypersphere. For **OCSVM**, we utilized the standard RBF kernel with `gamma='scale'`, dynamically setting the `nu` parameter equal to the ground-truth anomaly ratio of each dataset.

**Analysis.** As shown in Table 16, PGBC consistently outperforms these baselines by a significant margin. While normalizing flows (OCFlow) achieve a decent average AUC of 81.5%, they show significant variance (e.g., $\pm$ 26.4% on *YAHOO*). This instability indicates the inherent difficulty Normalizing Flows face in accurately mapping the complex, locally varying geometry of time-series embeddings to a regular latent density, compared to PGBC's adaptive granular-balls. Furthermore, traditional kernels (OCSVM) perform reasonably well (Avg 75.3%) but still lag behind PGBC (e.g., 41.0% vs. 99.8% on *YAHOO*), confirming that fixed kernels are insufficient to model locally varying anisotropic structures.

Table 16: Comparison with additional baselines on time-series anomaly detection (AUC %). **Bold** indicates the best performance.

| Datasets | OCSVM | OCFlow | **Ours** |
|---|---|---|---|
| NAB Traffic | $78.9 \pm 2.3$ | $74.0 \pm 2.7$ | $\mathbf{85.4 \pm 3.8}$ |
| WSD WebService | $92.9 \pm 0.4$ | $94.8 \pm 1.0$ | $\mathbf{98.8 \pm 0.4}$ |
| SMD Facility | $88.0 \pm 2.8$ | $97.1 \pm 2.2$ | $\mathbf{97.9 \pm 0.0}$ |
| IOPS WebService | $84.4 \pm 1.9$ | $84.0 \pm 4.4$ | $\mathbf{96.4 \pm 0.1}$ |
| UCR Medical | $66.3 \pm 4.7$ | $80.7 \pm 8.6$ | $\mathbf{93.5 \pm 0.1}$ |
| YAHOO Synthetic | $41.0 \pm 8.4$ | $58.4 \pm 26.4$ | $\mathbf{99.8 \pm 0.0}$ |
| **Average** | $75.3 \pm 3.4$ | $81.5 \pm 7.6$ | $\mathbf{95.3 \pm 0.7}$ |

## E.2  COMPARISON WITH OSR METHOD

To further validate the effectiveness of PGBC from an Open-Set Recognition (OSR) perspective, we compared it against the foundational baseline method: OpenMax (Bendale & Boult, 2016), which estimates the probability of an input being from an unknown class by fitting Weibull distributions to the tail activation distances of known classes. We followed the same OSR protocol described in Section 3.3, evaluating on five datasets from the UCR archive.

**Analysis.** As shown in Table 17, PGBC demonstrates overwhelming superiority over OpenMax, achieving an average AUC of **99.6%** compared to 68.7%. OpenMax relies on calibrating SoftMax scores, which implicitly assumes that known classes form tight, separable clusters in the penultimate layer. However, strictly enforcing such compactness on time-series data can be challenging, leading to feature distributions that defy OpenMax's distributional assumptions. Consequently, OpenMax struggles significantly, performing near random guessing on datasets like *CBF* (49.5%). In contrast, PGBC is inherently well-suited for this data geometry. Its **adaptive ellipsoidal granular-balls** can stretch and rotate to locally approximate these manifold structures, creating a tight and precise boundary around the normal data. This geometric alignment enables robust rejection of open-set samples regardless of the classification margin.

Table 17: Comparison with the Open-Set Recognition (OSR) baseline on time series tasks. Reported metric is AUC (%). PGBC significantly outperforms the classic method OpenMax. **Bold** indicates the best performance.

| Datasets | OpenMax | **Ours** |
|----------|---------|----------|
| Adiac | $92.4 \pm 7.4$ | $\mathbf{99.2 \pm 0.2}$ |
| CBF | $49.5 \pm 0.3$ | $\mathbf{100.0 \pm 0.0}$ |
| Synthetic Control | $53.0 \pm 5.0$ | $\mathbf{99.8 \pm 1.1}$ |
| SwedishLeaf | $63.9 \pm 4.8$ | $\mathbf{99.7 \pm 0.1}$ |
| Trace | $84.9 \pm 1.8$ | $\mathbf{99.6 \pm 1.0}$ |
| **Average** | $68.7 \pm 3.9$ | $\mathbf{99.6 \pm 0.5}$ |

### E.3 PRECISION, RECALL, AND F1-SCORE ANALYSIS

To provide a comprehensive evaluation beyond threshold-independent metrics like AUC, we report the Precision (P), Recall (R), and F1-score on 12 representative tabular datasets. The decision threshold for all methods is strictly determined by the ground-truth anomaly ratio (contamination) of each dataset.

**Analysis.** As detailed in Table 18, PGBC demonstrates exceptional robustness, achieving the **highest F1-score on all 12 datasets**. First, on datasets with complex structures such as *Bands42* and *Pendigits*, PGBC achieves perfect performance (100.0% F1), whereas the strong probabilistic baseline HGAD lags behind (90.2% and 95.6%, respectively), and distance-based methods like KNN struggle significantly. Second, the results on the challenging *Sick72* dataset highlight the precision-recall trade-off. While baselines like DAGMM achieve 100% Recall, they suffer from catastrophic Precision (2.0%), indicating they classify almost all samples as anomalies. In contrast, PGBC maintains a significantly higher Precision (**63.6%** vs. next best 17.6%), securing the highest F1-score (29.8%). This confirms that PGBC's high performance stems from a precise, well-fitted decision boundary rather than loose over-coverage.

Table 18: Precision (P), Recall (R), and F1-score (%) on 12 representative tabular datasets. **Bold** indicates the best performance, and underline indicates the second best. Abbreviations: AE = AutoEncoder, D.SV = DeepSVDD, DAG = DAGMM, GBM = GBMOD.

| Method | Bands34 | | | Bands42 | | | Ecoli | | | Iris | | | Musk | | | Pageblocks | | |
|---|---|---|---|---|---|---|---|---|---|---|---|---|---|---|---|---|---|---|
| | P | R | F1 | P | R | F1 | P | R | F1 | P | R | F1 | P | R | F1 | P | R | F1 |
| IForest | 35.6 | 61.8 | 45.2 | 28.6 | 85.7 | 42.9 | 58.3 | 77.8 | 66.7 | 100.0 | 100.0 | 100.0 | 96.8 | 92.8 | 94.7 | 68.0 | 74.8 | 71.2 |
| LOF | 32.7 | 52.9 | 40.4 | 34.4 | 50.0 | 40.8 | 77.8 | 77.8 | 77.8 | 83.3 | 90.9 | 87.0 | 100.0 | 100.0 | 100.0 | 84.0 | 85.7 | 84.8 |
| KNN | 29.9 | 58.8 | 39.6 | 33.3 | 54.8 | 41.4 | 87.5 | 77.8 | 82.4 | 91.7 | 100.0 | 95.7 | 100.0 | 100.0 | 100.0 | 89.0 | 90.7 | 89.8 |
| AE | 30.6 | 76.5 | 43.7 | 35.3 | 71.4 | 47.2 | 13.6 | 33.3 | 19.4 | 100.0 | 90.9 | 95.2 | 100.0 | 100.0 | 100.0 | 81.5 | 75.2 | 78.2 |
| D.SV | 11.8 | 85.3 | 20.7 | 12.5 | 100.0 | 22.2 | 36.8 | 77.8 | 50.0 | 10.8 | 90.9 | 19.2 | 83.5 | 78.4 | 80.9 | 57.7 | 69.8 | 63.2 |
| DAG | 13.4 | 73.5 | 22.6 | 12.6 | 100.0 | 22.3 | 2.7 | 100.0 | 5.2 | 42.9 | 81.8 | 56.2 | 100.0 | 76.3 | 86.5 | 28.4 | 31.4 | 29.8 |
| HGAD | 84.6 | 97.1 | 90.4 | 92.5 | 88.1 | 90.2 | 100.0 | 77.8 | 87.5 | 100.0 | 100.0 | 100.0 | 100.0 | 100.0 | 100.0 | 86.5 | 91.9 | 89.1 |
| GBM | 25.6 | 61.8 | 36.2 | 40.8 | 47.6 | 44.0 | 77.8 | 77.8 | 77.8 | 100.0 | 100.0 | 100.0 | 53.6 | 15.5 | 24.0 | 74.7 | 81.4 | 77.9 |
| GBDO | 21.6 | 55.9 | 31.1 | 24.5 | 61.9 | 35.1 | 70.0 | 77.8 | 73.7 | 53.3 | 72.7 | 61.5 | 100.0 | 100.0 | 100.0 | 64.6 | 78.7 | 71.0 |
| **Ours** | **94.4** | **100.0** | **97.1** | **100.0** | **100.0** | **100.0** | **100.0** | **77.8** | **87.5** | **100.0** | **100.0** | **100.0** | **100.0** | **100.0** | **100.0** | **100.0** | **98.4** | **99.2** |

| Method | Pendigits | | | Sonar | | | Sick72 | | | Tictac12 | | | Tictac26 | | | Tictac32 | | |
|---|---|---|---|---|---|---|---|---|---|---|---|---|---|---|---|---|---|---|
| | P | R | F1 | P | R | F1 | P | R | F1 | P | R | F1 | P | R | F1 | P | R | F1 |
| IForest | 45.1 | 47.4 | 46.2 | 83.3 | 100.0 | 90.9 | 4.8 | 91.7 | 9.2 | 63.2 | 100.0 | 77.4 | 84.2 | 61.5 | 71.1 | 77.4 | 75.0 | 76.2 |
| LOF | 83.9 | 93.6 | 88.5 | 76.9 | 100.0 | 87.0 | 6.3 | 25.0 | 10.1 | 90.9 | 83.3 | 87.0 | 90.5 | 73.1 | 80.9 | 90.5 | 59.4 | 71.7 |
| KNN | 91.0 | 97.4 | 94.1 | 90.0 | 90.0 | 90.0 | 6.7 | 90.3 | 12.4 | 100.0 | 91.7 | 95.7 | 85.2 | 88.5 | 86.8 | 87.9 | 90.6 | 89.2 |
| AE | 32.7 | 62.8 | 43.0 | 90.9 | 100.0 | 95.2 | 8.5 | 75.0 | 15.2 | 25.0 | 25.0 | 25.0 | 64.0 | 61.5 | 62.7 | 65.2 | 46.9 | 54.5 |
| SVDD | 2.3 | 100.0 | 4.5 | 100.0 | 30.0 | 46.2 | 5.3 | 38.9 | 9.3 | 9.5 | 16.7 | 12.1 | 6.2 | 53.8 | 11.1 | 7.2 | 31.2 | 11.7 |
| DAGMM | 7.1 | 72.4 | 12.9 | 19.0 | 80.0 | 30.8 | 2.0 | 100.0 | 3.9 | 2.1 | 100.0 | 4.1 | 4.3 | 61.5 | 8.1 | 4.9 | 100.0 | 9.3 |
| HGAD | 93.3 | 98.1 | 95.6 | 100.0 | 100.0 | 100.0 | 14.7 | 13.9 | 14.3 | 100.0 | 83.3 | 90.9 | 96.0 | 92.3 | 94.1 | 96.7 | 90.6 | 93.5 |
| GBMOD | 67.5 | 66.7 | 67.1 | 83.3 | 100.0 | 90.9 | 17.6 | 8.3 | 11.3 | 64.3 | 75.0 | 69.2 | 45.7 | 61.5 | 52.5 | 48.7 | 59.4 | 53.5 |
| GBDO | 29.3 | 34.6 | 31.8 | 44.4 | 40.0 | 42.1 | 5.5 | 95.8 | 10.3 | 11.1 | 8.3 | 9.5 | 12.1 | 15.4 | 13.6 | 10.1 | 21.9 | 13.9 |
| **Ours** | **100.0** | **100.0** | **100.0** | **100.0** | **100.0** | **100.0** | **63.6** | **19.4** | **29.8** | **100.0** | **100.0** | **100.0** | **100.0** | **100.0** | **100.0** | **100.0** | **100.0** | **100.0** |

## F    EXTENDED EXPERIMENTS ON VISUAL DATASETS

To demonstrate the versatility of PGBC on high-dimensional data beyond tabular and time-series domains, we conducted experiments on three widely recognized visual benchmarks: CIFAR-10, FashionMNIST, and MVTec-AD.

**Experimental Setup.** Adhering to the standard protocol established by ADBench (Han et al., 2022), we utilized the provided pre-extracted deep feature embeddings as input for all methods. This standardized setup allows for a direct and focused comparison regarding the ability to model complex, anisotropic distributions inherent in semantic feature spaces. Specifically, the input data consists of **512-dimensional** feature vectors derived from a pre-trained **Vision Transformer (ViT)** backbone.

**Implementation Details and Baselines.** We compared PGBC against a diverse set of baselines. For the CIFAR-10 and FashionMNIST datasets, we compare PGBC against five classical feature-based methods: DeepSVDD, AutoEncoder (AE), GBMOD, KNN, and HGAD. For all these methods (**DeepSVDD, AE, GBMOD, KNN, HGAD**), we maintained the hyperparameter configurations described in Appendix C.2, adjusting only the input layer size to match the 512-dimensional feature vectors. PGBC was applied directly to these feature vectors without any additional fine-tuning.

For the MVTec-AD industrial benchmark, we adopt a comprehensive mixed comparison strategy. The results of the five aforementioned feature-based baselines and PGBC are obtained by running them on the 512-dimensional ViT feature vectors (ADBench protocol). However, to benchmark against leading methods that inherently rely on raw spatial information, we also include results from three prominent *image-based* methods, namely **CutPaste** (Li et al., 2021), **U-student** (Uninformed students) (Bergmann et al., 2020), and **P-SVDD** (Patch-level SVDD) (Yi & Yoon, 2020). Since the core mechanisms of these image-based methods (such as self-supervised spatial augmentation) cannot be fairly tested on pre-extracted feature vectors, their performance on MVTec-AD is cited directly from the original CutPaste paper (Li et al., 2021).

**Results and Discussion.** The detailed results for CIFAR-10 and FashionMNIST subsets are reported in Table 19, while the MVTec-AD results, including the image-based baselines, are reported separately in Table 20. PGBC consistently achieves the highest average AUC scores across all three benchmarks: **95.5%** on CIFAR-10, **94.9%** on FashionMNIST, and **93.0%** on MVTec-AD.

First, when comparing with feature-based baselines, PGBC outperforms the second-best method, HGAD, across all three datasets. On CIFAR-10 and FashionMNIST, PGBC achieves 95.5% and 94.9% respectively, both exceeding HGAD (95.3% and 94.6%). On the MVTec-AD dataset, PGBC (93.0% AUC) also clearly outperforms the best feature-based competitor, HGAD (91.2% AUC). This stable leading performance across diverse visual domains demonstrates that our adaptive granular-ball construction is superior to standard density estimation techniques like hierarchical Gaussian mixtures in modeling complex feature distributions.

Second, we analyze the competitive landscape on MVTec-AD, which includes image-based state-of-the-art methods. PGBC (**93.0%** Avg AUC) achieves the highest overall average score, surpassing specialized image-based methods such as **CutPaste** (90.9% Avg AUC), **P-SVDD** (92.1% Avg AUC), and the competitive **U-student** (92.5% Avg AUC). This superior performance is highly significant: it confirms that PGBC effectively leverages powerful deep feature representations to achieve state-of-the-art detection on challenging industrial data, even when benchmarked against methods that operate directly on raw pixels.

Finally, we emphasize the advantage provided by PGBC's probabilistic ellipsoidal modeling. We observe that methods relying on simple geometric boundaries, such as DeepSVDD, show limited capability when dealing with complex visual feature manifolds. For instance, on the CIFAR-10 and F-MNIST datasets shown in Table 19, DeepSVDD exhibits significantly lower average AUCs (69.5% and 58.5%, respectively), which is consistent with the rigidity of its hyperspherical boundary assumption being ill-suited for complex data distributions. In contrast, PGBC's probabilistic ellipsoidal modeling successfully captures these inherent geometries, providing consistently superior and stable performance across all benchmarks.

Table 19: Full quantitative results (AUC ↑ and FPR ↓ in %) on CIFAR-10 and FashionMNIST. **Bold** indicates the best performance, and underline indicates the second best. Abbreviations: GBM=GBMOD, U.S=U-student.

| Datasets | Subset | DeepSVDD AUC | DeepSVDD FPR | AutoEncoder AUC | AutoEncoder FPR | GBMOD AUC | GBMOD FPR | KNN AUC | KNN FPR | HGAD AUC | HGAD FPR | Ours AUC | Ours FPR |
|---|---|---|---|---|---|---|---|---|---|---|---|---|---|
| CIFAR10 | 0 | 78.7 | 15.4 | 93.5 | 8.0 | 92.0 | 8.4 | 91.9 | 7.0 | 94.1 | 6.3 | **94.4** | **2.6** |
| | 1 | 78.9 | 21.4 | 95.7 | 3.3 | 95.9 | 2.1 | 95.7 | 4.6 | 96.6 | **1.1** | **96.7** | 3.2 |
| | 2 | 67.1 | 37.2 | 89.0 | 11.3 | 89.0 | 12.6 | **91.0** | 13.8 | 90.1 | 9.0 | 90.6 | 6.8 |
| | 3 | 68.4 | 19.9 | 92.7 | 7.1 | 93.3 | 4.9 | **94.1** | 4.9 | 93.3 | 7.5 | 93.9 | 3.5 |
| | 4 | 66.8 | 48.2 | 95.7 | **2.1** | 95.2 | 4.0 | 95.5 | 4.6 | 96.1 | 4.4 | **96.2** | 6.7 |
| | 5 | 63.3 | 64.1 | 92.7 | **3.3** | 94.0 | 5.2 | **94.1** | 4.5 | 94.0 | 4.8 | **94.1** | 3.6 |
| | 6 | 67.7 | 35.1 | 96.8 | 3.2 | 96.2 | 3.2 | 97.1 | 4.4 | 97.2 | **2.9** | **97.4** | **2.9** |
| | 7 | 66.3 | 37.8 | 97.1 | 5.4 | 96.7 | **4.3** | 96.9 | 6.9 | 97.4 | 5.0 | **97.5** | 4.5 |
| | 8 | 73.3 | 25.4 | 96.7 | 3.5 | **97.0** | 3.3 | 96.9 | 3.5 | **97.0** | 2.8 | **97.0** | 3.3 |
| | 9 | 64.4 | 31.5 | 95.9 | 3.3 | 96.7 | 3.5 | 95.9 | 3.6 | 96.8 | **1.4** | **96.9** | 4.4 |
| | **Avg** | 69.5 | 33.6 | 94.6 | 5.1 | 94.6 | 5.2 | 94.9 | 5.8 | 95.3 | 4.5 | **95.5** | **4.2** |
| F-MNIST | 0 | 47.8 | 99.9 | 91.3 | 7.1 | 89.9 | 5.6 | 90.1 | 4.5 | 92.2 | 6.4 | **92.6** | **3.0** |
| | 1 | 55.1 | 67.8 | 99.7 | 1.2 | 98.9 | 3.7 | 99.7 | **0.9** | **99.8** | 1.5 | **99.8** | 1.1 |
| | 2 | 55.3 | 23.6 | 93.4 | 4.8 | 93.1 | 5.7 | 93.7 | **2.4** | 93.8 | 4.5 | 93.8 | 4.1 |
| | 3 | 58.5 | 31.8 | 90.4 | 9.7 | 85.9 | 8.2 | 86.6 | **3.9** | 91.7 | 7.1 | **92.2** | 5.9 |
| | 4 | 61.8 | 33.0 | 89.2 | 2.7 | 87.2 | 4.1 | 88.2 | 6.2 | 89.7 | 5.5 | **90.0** | **2.6** |
| | 5 | 77.6 | 18.2 | 94.8 | **0.4** | 94.0 | 1.8 | 93.8 | 0.5 | 96.2 | 1.1 | **96.8** | 0.5 |
| | 6 | 55.8 | 79.7 | 87.6 | 2.2 | 83.9 | 9.8 | 87.1 | 2.2 | 87.7 | **1.8** | **88.1** | 2.0 |
| | 7 | 71.8 | 31.6 | 98.3 | 0.8 | 97.5 | 1.8 | 97.8 | 1.0 | 98.6 | 1.2 | **98.7** | **0.6** |
| | 8 | 50.5 | 78.1 | 98.3 | 4.4 | 96.4 | 4.6 | 98.5 | 4.2 | **98.6** | 3.6 | **98.6** | 4.2 |
| | 9 | 50.4 | 100.0 | 98.0 | **0.9** | 97.9 | 1.4 | 98.1 | 1.2 | **98.3** | 1.3 | **98.3** | 1.2 |
| | **Avg** | 58.5 | 56.4 | 94.1 | 3.4 | 92.5 | 4.7 | 93.3 | 2.7 | 94.6 | 3.4 | **94.9** | **2.5** |

Table 20: Full AUC results (%) on the MVTec-AD dataset. Abbreviations: GBM=GBMOD, U.S=U-student. **Bold** indicates the best performance, and underline indicates the second best.

| Datasets | KNN | GBM | HGAD | CutPaste | U.S | PSVDD | Ours |
|---|---|---|---|---|---|---|---|
| bottle | 99.7 | 99.3 | **99.9** | 99.2 | 96.7 | 98.6 | **99.9** |
| cable | 94.6 | 90.0 | 93.7 | 87.1 | 82.3 | 90.3 | **96.4** |
| capsule | 77.7 | 72.4 | 82.3 | 87.9 | **92.8** | 76.7 | 91.5 |
| carpet | 97.1 | 96.1 | 97.4 | 67.9 | 95.3 | 92.9 | **97.9** |
| grid | 73.9 | 60.6 | 85.0 | **99.9** | 98.7 | 94.6 | 87.6 |
| hazelnut | 82.1 | 77.2 | 87.6 | 91.3 | 91.4 | **92.0** | 87.0 |
| leather | **100.0** | **100.0** | **100.0** | 99.7 | 93.4 | 90.9 | **100.0** |
| metalnut | 89.7 | 82.8 | 93.0 | **96.8** | 94.0 | 94.0 | 95.9 |
| pill | 77.0 | 73.9 | 85.1 | **93.4** | 86.7 | 86.1 | 85.3 |
| screw | 67.1 | 60.3 | 76.7 | 54.4 | **87.4** | 81.3 | 77.0 |
| tile | 98.9 | 98.0 | **99.7** | 95.9 | 95.8 | 97.8 | **99.7** |
| toothbrush | 96.2 | 81.7 | 96.2 | 99.2 | 98.6 | **100.0** | 97.5 |
| transistor | 83.3 | 79.6 | 84.6 | **96.4** | 83.6 | 91.5 | 89.9 |
| wood | 85.2 | 74.5 | 94.9 | 94.9 | 95.5 | 96.5 | **98.0** |
| zipper | 84.5 | 82.0 | 92.3 | **99.4** | 95.8 | 97.9 | 91.7 |
| **Avg** | 87.1 | 81.9 | 91.2 | 90.9 | 92.5 | 92.1 | **93.0** |

# G ABLATION STUDIES

In this section, we conduct a comprehensive ablation analysis to validate the contribution of key components in PGBC: the dynamic reassignment step and the BIC splitting criterion.

## G.1 IMPACT OF DYNAMIC REASSIGNMENT

The dynamic reassignment step (Step 3 in Algorithm 1) ensures that data points are associated with the most likely Gaussian component after each split. To verify its necessity, we compared the performance and runtime of PGBC with and without this step.

As shown in Table 21, enabling dynamic reassignment consistently improves detection accuracy. For instance, on the *Cardio* dataset, AUC increases significantly from 80.41% to **84.04%**, and on *SwedishLeaf*, FPR drops from 2.10% to **1.81%**. This confirms that refining the data assignment after splitting allows the granular-balls to better fit the local data geometry. While this step involves iterative computation, the overall efficiency of our method remains high, as detailed in the comprehensive runtime analysis in **Appendix I**.

Table 21: Ablation study on the Dynamic Reassignment step. Impact on AUC (%) and FPR (%) across six representative datasets. **Bold** indicates the better performance.

| Datasets | Reassign | AUC% ($\uparrow$) | FPR% ($\downarrow$) |
|---|---|---|---|
| NAB Traffic | $\times$ | $85.06 \pm 0.79$ | $7.81 \pm 0.24$ |
| | $\checkmark$ | $\mathbf{85.68 \pm 3.04}$ | $\mathbf{7.58 \pm 0.83}$ |
| WSD WebService | $\times$ | $98.58 \pm 0.01$ | $0.50 \pm 0.00$ |
| | $\checkmark$ | $\mathbf{98.69 \pm 0.04}$ | $\mathbf{0.44 \pm 0.00}$ |
| Abalone | $\times$ | $70.46 \pm 0.76$ | $1.78 \pm 0.00$ |
| | $\checkmark$ | $\mathbf{72.10 \pm 0.33}$ | $\mathbf{1.72 \pm 0.01}$ |
| Cardio | $\times$ | $80.41 \pm 0.07$ | $1.74 \pm 0.06$ |
| | $\checkmark$ | $\mathbf{84.04 \pm 0.96}$ | $\mathbf{1.63 \pm 0.12}$ |
| Adiac | $\times$ | $98.22 \pm 1.20$ | $2.09 \pm 1.05$ |
| | $\checkmark$ | $\mathbf{98.58 \pm 1.11}$ | $\mathbf{1.96 \pm 1.17}$ |
| SwedishLeaf | $\times$ | $98.10 \pm 0.17$ | $2.10 \pm 0.38$ |
| | $\checkmark$ | $\mathbf{98.66 \pm 0.39}$ | $\mathbf{1.81 \pm 0.36}$ |

## G.2 IMPACT OF BIC CRITERION

A critical challenge in hierarchical density estimation is determining the optimal stopping condition to prevent over-partitioning, where the model fits local noise rather than the underlying distribution. To validate the efficacy of the Bayesian Information Criterion (BIC) as a statistical regularizer in PGBC, we compared our proposed method ("Full") against a baseline variant ("No BIC") that executes splits solely based on positive Log-Likelihood Gain (LLG), effectively removing the penalty for model complexity.

The quantitative results in Table 22 demonstrate that the BIC criterion serves as an essential defense against overfitting. Without the BIC penalty ("No BIC"), the algorithm aggressively pursues marginal likelihood gains, leading to an explosion in the number of granular-balls. For instance, on the *Cardio* dataset, the number of components surges from a parsimonious 17 to an excessive 721, and on *Abalone*, it increases from 25 to 668. This uncontrolled growth has severe consequences for computational efficiency, with runtimes increasing by orders of magnitude (e.g., from 3.13s to 183.41s on *Abalone* and from 1.46s to 37.33s on *WSD WebService*). This trend is consistent across both tabular datasets and time-series embeddings (e.g., NAB and WSD), indicating that without the BIC penalty, the model tends to interpret insignificant local data variations as distinct structural components. By incorporating the BIC term, PGBC successfully balances data fit with model complexity, ensuring a representation that is both statistically significant and computationally efficient.

Table 22: Ablation study on the impact of the BIC Criterion. Comparison of PGBC ("Full") against a variant without the BIC penalty ("w/o BIC") across 7 datasets. (a) Model Complexity (Number of Granular-balls). (b) Efficiency (Runtime in seconds). **Bold** indicates the more compact model and efficient runtime.

(a) Model Complexity (Number of Granular-balls)

| Strategy | Abalone | Bands34 | Bands42 | Cardio | Ecoli | NAB | WSD |
|---|---|---|---|---|---|---|---|
| **Full (Ours)** | **25** | **10** | **14** | **17** | **5** | **6** | **9** |
| w/o BIC | 668 | 312 | 161 | 721 | 314 | 269 | 382 |

(b) Efficiency (Runtime in seconds)

| Strategy | Abalone | Bands34 | Bands42 | Cardio | Ecoli | NAB | WSD |
|---|---|---|---|---|---|---|---|
| **Full (Ours)** | **3.13** | **0.62** | **0.72** | **1.83** | **0.32** | **0.25** | **1.46** |
| w/o BIC | 183.41 | 6.44 | 6.29 | 42.80 | 5.07 | 4.35 | 37.33 |

### G.3 DOES THE METHOD SUPPORT A SINGLE PRINCIPAL COMPONENT?

PGBC naturally supports data dominated by one or few principal components without any architectural modification. On three 2D toy datasets with highly elongated distributions (Figure 6), the BIC criterion correctly terminates splitting at the root level in all cases, producing exactly one ellipsoidal granular-ball $\mathcal{B}^{(0)}$ that precisely aligns with the dominant principal direction(s). This demonstrates that PGBC gracefully degenerates to a single anisotropic Gaussian whenever the data structure warrants it, confirming its full adaptivity across both complex high-rank and simple low-rank scenarios.

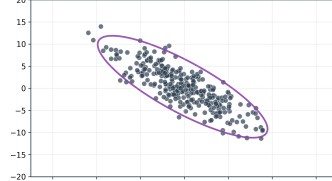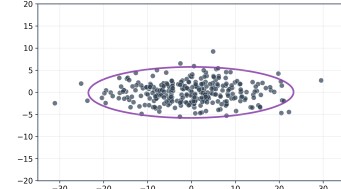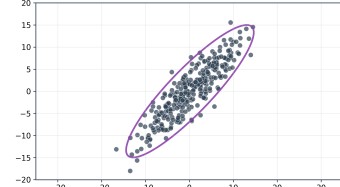

Figure 6: Visualization of PGBC on three 2D toy datasets with highly anisotropic distributions. In all cases, PGBC automatically stops splitting and fits a single elongated ellipsoid $\mathcal{B}^{(0)}$ perfectly aligned with the data manifold, demonstrating degeneration to a single principal component Gaussian.

# H ANOMALY SCORING MECHANISMS

To rigorously evaluate the effectiveness of our proposed hierarchical anomaly scoring mechanism, we conducted a component-wise comparative study. We analyze four distinct aspects of the scoring strategy in the following order: (1) the necessity of hierarchical aggregation, (2) the effectiveness of entropy-based weighting schemes, (3) the impact of the probabilistic scoring metric, and (4) the importance of score normalization. The detailed analysis for each component is provided below.

## H.1 IMPACT OF HIERARCHICAL AGGREGATION (HIERARCHY VS. LEAF-ONLY)

To validate the necessity of the hierarchical structure, we compared PGBC against a "Leaf-only" strategy, which utilizes the finest-grained granular-balls at the leaf nodes to compute anomaly scores.

As shown in Table 23, the hierarchical PGBC consistently outperforms the flat "Leaf-only" approach across all datasets. For instance, on *Thyroid*, the AUC rises from 67.8% to 73.0%, and on *Sick72* from 84.2% to 87.3%. This result demonstrates that relying solely on fine-grained leaf nodes is insufficient, as they may overfit to local variations. In contrast, intermediate layers in the hierarchy capture valuable multi-scale structural information that is critical for robust anomaly detection.

To visually illustrate this, Figure 7 plots the layer-wise AUC on the *Thyroid* dataset. Performance fluctuates significantly across layers, and the finest granularity (leaves) is not necessarily optimal. PGBC's hierarchical aggregation effectively integrates these complementary scales.

Table 23: Comparison of Hierarchical Aggregation vs. Leaf-only strategy. Metric: AUC (%).

| Method | Bands34 | Bands42 | Ecoli | Sick72 | Thyroid | Waveform |
|---|---|---|---|---|---|---|
| Leaf-only | $97.4 \pm 2.4$ | $99.1 \pm 0.7$ | $88.8 \pm 0.3$ | $84.2 \pm 1.0$ | $67.8 \pm 2.2$ | $77.6 \pm 0.0$ |
| **Ours** | $\mathbf{100.0 \pm 0.0}$ | $\mathbf{100.0 \pm 0.0}$ | $\mathbf{89.4 \pm 0.4}$ | $\mathbf{87.3 \pm 3.6}$ | $\mathbf{73.0 \pm 4.8}$ | $\mathbf{78.1 \pm 0.0}$ |

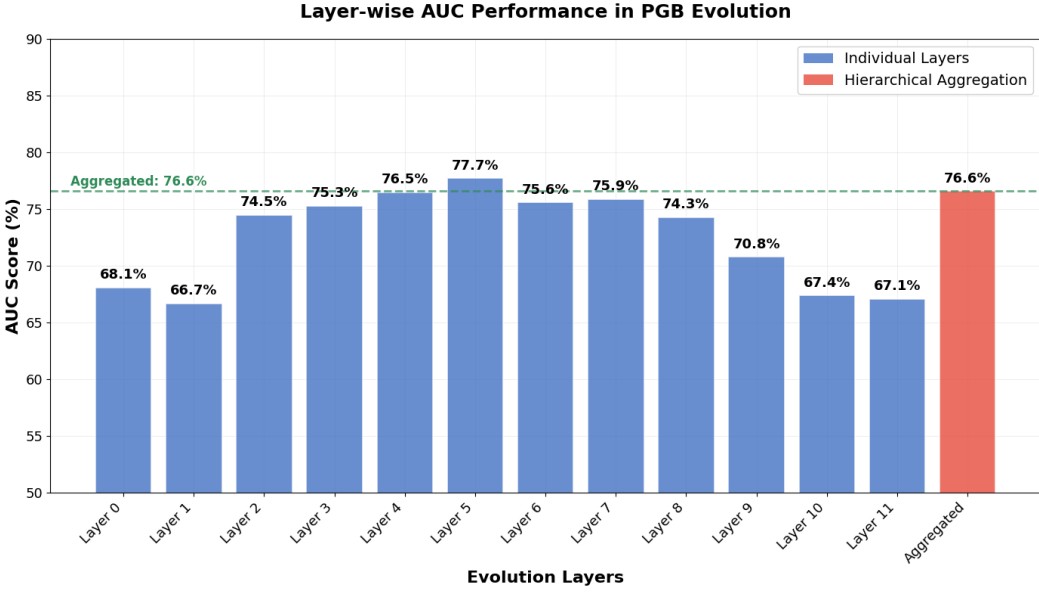

Figure 7: Layer-wise AUC performance on the Thyroid dataset. The chart compares the performance of individual layers against the hierarchical aggregation (Ours), validating the necessity of multi-granularity scoring.

## H.2 IMPACT OF WEIGHTING SCHEMES (ENTROPY VS. UNIFORM)

We examined the effectiveness of our entropy-based weighting scheme by comparing it against a "Uniform" strategy, where every hierarchical level contributes equally to the final score.

Table 24 shows that the entropy-based weighting consistently yields superior results. For example, on *Bands34*, the AUC drops to 96.4% with uniform weights, and on *Sick72*, it falls drastically to 78.3% (a decrease of 9.0%). This significant performance gap highlights the limitation of the "Uniform" strategy: it treats coarse global approximations and fine local details identically. Consequently, levels with lower discriminative power can dilute the precise anomaly signals captured by more informative levels. In contrast, our entropy-based weighting adapts dynamically to the data complexity. Specifically, when the data structure is complex and necessitates deeper recursive splitting, the entropy metric naturally assigns higher weights to the fine-grained levels that capture intricate local patterns. Consequently, the influence of coarse-grained levels—which provide only rough global statistics—is automatically attenuated, ensuring that the detection is driven by the most detailed and informative resolution.

Table 24: Comparison of Entropy-based vs. Uniform Weighting. Metric: AUC (%).

| Method | Bands34 | Bands42 | Ecoli | Sick72 | Thyroid | Waveform |
|---|---|---|---|---|---|---|
| Uniform | $96.4 \pm 2.6$ | $97.8 \pm 0.9$ | $88.4 \pm 0.1$ | $78.3 \pm 2.4$ | $72.9 \pm 6.1$ | $74.1 \pm 0.0$ |
| **Ours** | $\mathbf{100.0 \pm 0.0}$ | $\mathbf{100.0 \pm 0.0}$ | $\mathbf{89.4 \pm 0.4}$ | $\mathbf{87.3 \pm 3.6}$ | $\mathbf{73.0 \pm 4.8}$ | $\mathbf{78.1 \pm 0.0}$ |

## H.3 IMPACT OF SCORING METRIC (LOG-LIKELIHOOD VS. EUCLIDEAN DISTANCE)

To isolate the contribution of our probabilistic scoring mechanism, we conducted an ablation study using two alternative scoring strategies based on Euclidean distance. Crucially, these baselines share the exact same hierarchical structure and granular-ball centers generated by PGBC, differing solely in how the anomaly score is computed:

- **Min-Distance:** Calculates the Euclidean distance to the *nearest* granular-ball center. This represents a "vanilla" GBC approach, treating granular-balls as isotropic spheres and ignoring local shape information.

- **Avg-Distance:** Calculates the *average* Euclidean distance to *all* granular-ball centers. This strategy, suggested for comparison, incorporates global geometric information rather than local density.

The results in Table 25 reveal a significant performance gap between these distance-based metrics and our proposed probabilistic scoring. First, **Min-Distance** consistently underperforms PGBC. For instance, on the *Bands* datasets, the AUC drops from 100.0% (Ours) to approximately 82–84%. This confirms that simple geometric distance fails to capture the anisotropic structures (e.g., elongated clusters) inherent in the data, whereas our log-likelihood scoring successfully leverages the covariance matrix to model local orientation. Second, **Avg-Distance** proves to be an unstable metric for anomaly detection. While it outperforms Min-Distance on *Thyroid* (67.6% vs. 56.2%) and *Ecoli*, it performs worse on *Bands42* (75.6%). This inconsistency suggests that averaging distances incorporates irrelevant global geometric information that obscures the precise local anomaly signals. Ultimately, PGBC (Log-Likelihood) surpasses both distance baselines across all datasets, demonstrating that explicit probabilistic modeling is essential for precise data description.

Table 25: Ablation study on scoring metrics. "Min-Dist" and "Avg-Dist" denote the minimum and average Euclidean distances to granular-ball centers, respectively. Metric: AUC (%).

| Method | Bands34 | Bands42 | Ecoli | Sick72 | Thyroid | Waveform |
|---|---|---|---|---|---|---|
| Min-Dist | $83.9 \pm 1.8$ | $81.8 \pm 2.6$ | $79.5 \pm 2.8$ | $80.9 \pm 1.4$ | $56.2 \pm 1.5$ | $71.6 \pm 0.0$ |
| Avg-Dist | $81.3 \pm 1.4$ | $75.6 \pm 3.8$ | $84.3 \pm 0.4$ | $79.6 \pm 0.5$ | $67.6 \pm 0.2$ | $68.8 \pm 0.0$ |
| **Ours** | $\mathbf{100.0 \pm 0.0}$ | $\mathbf{100.0 \pm 0.0}$ | $\mathbf{89.4 \pm 0.4}$ | $\mathbf{87.3 \pm 3.6}$ | $\mathbf{73.0 \pm 4.8}$ | $\mathbf{78.1 \pm 0.0}$ |

### H.4 IMPACT OF SCORE NORMALIZATION

Finally, to assess the importance of aligning scores across levels, we tested a variant "w/o Normalization", which aggregates raw log-likelihood scores without min-max scaling.

As presented in Table 26, removing normalization leads to clear performance degradation. For instance, the AUC on *Sick72* drops from 87.3% to 81.2%. This degradation arises because the raw log-likelihood scores at different hierarchical levels often exhibit vastly different value ranges, reflecting the varying granularity of the data description from global to local scales. Without normalization, levels with numerically larger score ranges could inadvertently overshadow the contributions of other levels, biasing the final result. Normalization ensures that the contribution of each level is governed strictly by its structural informativeness (entropy), rather than arbitrary differences in numerical magnitude.

Table 26: Impact of Score Normalization. Metric: AUC (%).

| Method | Bands34 | Bands42 | Ecoli | Sick72 | Thyroid | Waveform |
|---|---|---|---|---|---|---|
| w/o Norm | $99.6 \pm 0.7$ | $99.1 \pm 0.7$ | $88.8 \pm 0.3$ | $81.2 \pm 1.8$ | $71.3 \pm 6.5$ | $77.8 \pm 0.0$ |
| **Ours** | $\mathbf{100.0 \pm 0.0}$ | $\mathbf{100.0 \pm 0.0}$ | $\mathbf{89.4 \pm 0.4}$ | $\mathbf{87.3 \pm 3.6}$ | $\mathbf{73.0 \pm 4.8}$ | $\mathbf{78.1 \pm 0.0}$ |

# I  COMPUTATIONAL EFFICIENCY ANALYSIS

In this section, we provide a theoretical analysis of the time complexity of PGBC and report empirical runtime comparisons against all baseline methods.

## I.1  THEORETICAL COMPLEXITY

The computational cost of PGBC is primarily determined by the hierarchical construction process using the Expectation-Maximization (EM) algorithm. For a dataset with $N$ samples and $d$ dimensions, let $L$ denote the tree depth and $t$ denote the average number of EM iterations per split. Since the summation of samples across all granular-balls at any specific level is bounded by $N$, the computational complexity for one level is approximately $\mathcal{O}(t \cdot N \cdot d^2)$. Consequently, the total training complexity is $\mathcal{O}(L \cdot t \cdot N \cdot d^2)$. This is generally more efficient than deep neural networks, where the number of training epochs (typically 50-100) far exceeds the tree depth $L$ (typically $< 10$). For inference, computing the anomaly score involves evaluating Gaussian densities across the constructed hierarchy, resulting in a complexity of $\mathcal{O}(K_{total} \cdot d^2)$, where $K_{total}$ is the total number of granular-balls, ensuring fast retrieval.

Table 27: Average runtime (seconds) comparison on 19 tabular datasets. **Bold** indicates the proposed method. Abbreviations: AE = AutoEncoder, D.SV = DeepSVDD, DAG = DAGMM, GBM = GBMOD.

| Datasets | IForest | LOF | KNN | AE | D.SV | DAG | HGAD | GBM | GBDO | **Ours** |
|---|---|---|---|---|---|---|---|---|---|---|
| Abalone | 0.22 | 0.13 | 0.04 | 16.63 | 13.04 | 29.95 | 40.38 | 1.14 | 21.13 | 6.28 |
| Bands34 | 0.24 | 0.25 | 0.01 | 8.42 | 5.96 | 2.87 | 3.31 | 3.45 | 2.29 | 0.98 |
| Bands42 | 0.18 | 0.25 | 0.01 | 16.63 | 5.31 | 2.67 | 3.27 | 2.61 | 7.14 | 2.60 |
| Cardio | 0.19 | 0.27 | 0.01 | 7.43 | 5.67 | 11.02 | 16.78 | 15.62 | 14.00 | 4.92 |
| Ecoli | 0.19 | 0.01 | 0.01 | 6.91 | 0.21 | 2.59 | 3.68 | 0.14 | 5.87 | 0.89 |
| Iris | 0.20 | 0.00 | 0.00 | 6.28 | 0.10 | 1.02 | 1.28 | 1.12 | 2.06 | 0.36 |
| Musk | 0.23 | 0.30 | 0.04 | 25.58 | 4.97 | 19.66 | 28.98 | 28.64 | 17.04 | 21.64 |
| Pageblocks | 0.22 | 0.23 | 0.08 | 18.88 | 7.99 | 31.86 | 46.76 | 1.54 | 22.15 | 58.46 |
| Pendigits | 0.26 | 0.35 | 0.05 | 45.70 | 10.70 | 42.87 | 63.44 | 8.06 | 25.68 | 49.19 |
| Satellite | 0.23 | 0.32 | 0.04 | 15.81 | 7.19 | 28.73 | 42.89 | 5.78 | 21.40 | 5.80 |
| Sick35 | 0.21 | 0.28 | 0.02 | 26.51 | 5.81 | 22.79 | 33.96 | 43.46 | 21.94 | 11.99 |
| Sick72 | 0.21 | 0.28 | 0.02 | 28.29 | 5.85 | 22.83 | 33.85 | 43.58 | 21.97 | 12.31 |
| Sonar | 0.22 | 0.26 | 0.01 | 19.19 | 0.14 | 1.24 | 1.37 | 0.97 | 1.88 | 1.11 |
| Thyroid | 0.26 | 0.90 | 0.10 | 40.77 | 24.85 | 57.76 | 62.22 | 21.39 | 37.22 | 21.12 |
| Tictac12 | 0.19 | 0.02 | 0.01 | 8.93 | 0.27 | 4.14 | 6.04 | 0.30 | 7.54 | 15.16 |
| Tictac26 | 0.19 | 0.02 | 0.01 | 9.19 | 0.28 | 4.31 | 6.16 | 0.31 | 7.83 | 15.27 |
| Tictac32 | 0.19 | 0.02 | 0.01 | 9.45 | 0.29 | 4.56 | 6.36 | 0.31 | 7.71 | 15.55 |
| Waveform | 0.21 | 0.28 | 0.02 | 27.79 | 5.63 | 22.63 | 32.83 | 29.62 | 16.19 | 1.44 |
| Yeast | 0.19 | 0.05 | 0.02 | 12.80 | 0.52 | 7.66 | 10.98 | 0.54 | 11.45 | 2.41 |
| **Average** | 0.21 | 0.22 | 0.03 | 18.48 | 7.09 | 16.90 | 23.40 | 10.98 | 14.34 | **13.03** |

## I.2  EMPIRICAL RUNTIME COMPARISON

To evaluate real-world efficiency, we measured the total runtime for all methods on the 19 tabular datasets. All runtime experiments in this section were conducted on a machine equipped with an NVIDIA RTX 2060 GPU (6 GB), an AMD Ryzen 7 4800H CPU, and 16 GB of RAM. This setup differs from the main experiments (which used an RTX 4090 server), serving to demonstrate the accessibility and efficiency of our method on standard hardware.

The runtime results in Table 27 illustrate the trade-off between computational cost and model complexity. Traditional distance-based methods like KNN and IForest are extremely fast due to their algorithmic simplicity, but as shown in the main text, they often fail to capture complex anisotropic patterns, resulting in lower detection accuracy. In contrast, deep learning-based approaches such as AutoEncoder and HGAD are significantly slower (averaging 18.59s and 24.67s, respectively) due to the necessity of iterative gradient descent over many epochs. PGBC achieves an average runtime

of **13.03s**, positioning it advantageously between these two extremes. It is nearly **2x faster** than HGAD and consistently outperforms other iterative methods like DAGMM. This efficiency indicates that PGBC successfully avoids the heavy computational burden of deep neural networks while providing a sophisticated probabilistic description that surpasses simple distance-based baselines, making it a highly practical solution for real-world anomaly detection tasks.

### I.3 MODEL COMPACTNESS ANALYSIS

**Tabular Data Analysis.** To complement the reduction factor analysis in the main text, Figure 8 presents the number of components required by traditional GBC versus PGBC across all 19 tabular datasets. It is evident that PGBC achieves a consistent and dramatic reduction in model complexity. For instance, on the *Sick72* dataset, traditional GBC requires 1009 isotropic balls to cover the data, whereas PGBC achieves a more precise coverage with only 44 ellipsoidal components. Similarly, on *Waveform*, the count drops from 740 to 23. This confirms that PGBC's covariance-aware modeling effectively eliminates the need for excessive recursive splitting in real-world anisotropic regions.

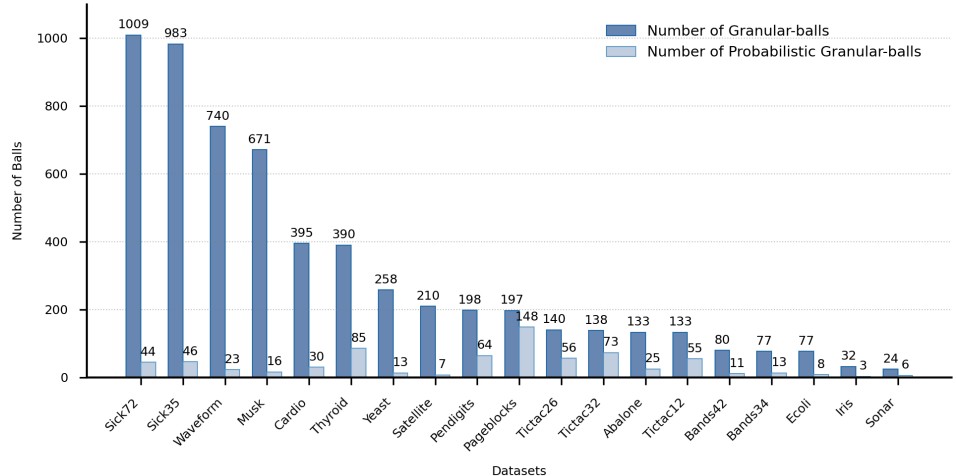

Figure 8: Comparison of model complexity (absolute number of granular-balls) on 19 tabular datasets. Dark blue bars represent traditional GBC, while light blue bars represent PGBC. PGBC consistently maintains a significantly more compact representation.

**Synthetic Data Verification.** To evaluate the efficiency of PGBC in idealized scenarios, we analyzed four synthetic datasets: Two Circles, Two Moons, Intersecting, and DB. Figure 9 reports the number of granular-balls required by traditional isotropic Granular-Ball Computing (GBC) versus the proposed PGBC. Quantitatively, PGBC achieves a dramatic reduction in model size, reducing the component count by factors ranging from 7.5× to 26.8×. As visually demonstrated in **Appendix K** (see Figure 11), this compactness stems from PGBC's superior adaptivity: unlike GBC, PGBC aligns ellipsoidal boundaries with the underlying data geometry, eliminating the need for unnecessary splits in anisotropic regions. This results in a highly parsimonious representation that significantly lowers computational cost without sacrificing data coverage or detection accuracy.

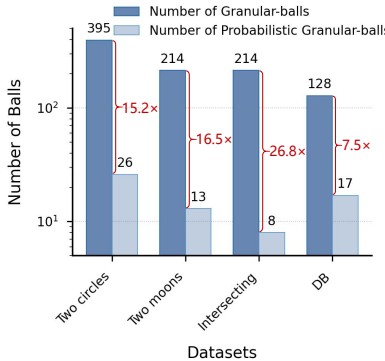

Figure 9: Comparison of model complexity on 4 synthetic datasets. The chart highlights the reduction in the number of granular-balls required by PGBC compared to traditional GBC.

# J    ROBUSTNESS TO CONTAMINATION

To evaluate the robustness of PGBC against label noise, we conducted experiments by introducing varying ratios of anomalies (from 0% to 5%) into the training sets of eight representative datasets. These datasets span diverse domains, including six tabular datasets (e.g., *Bands34*, *Iris*, *Sonar*), one time series dataset (*WSD WebService*), and one open-set recognition dataset (*SwedishLeaf*), ensuring a comprehensive assessment across different data modalities.

As shown in Figure 10, PGBC demonstrates remarkable stability across most scenarios. On datasets with clear structural separation like *Iris* and *WSD WebService*, the performance drop is negligible ($\leq 1\%$), indicating near-perfect immunity to contamination. Even on challenging datasets such as *Bands34* and *Bands42*, where the AUC decreases by approximately 10%, the model exhibits a graceful degradation, maintaining absolute scores above 89%. This resilience stems from the statistical rigor of the granular-ball splitting process, which effectively isolates sparse noisy samples and prevents them from distorting the learned normal data distribution.

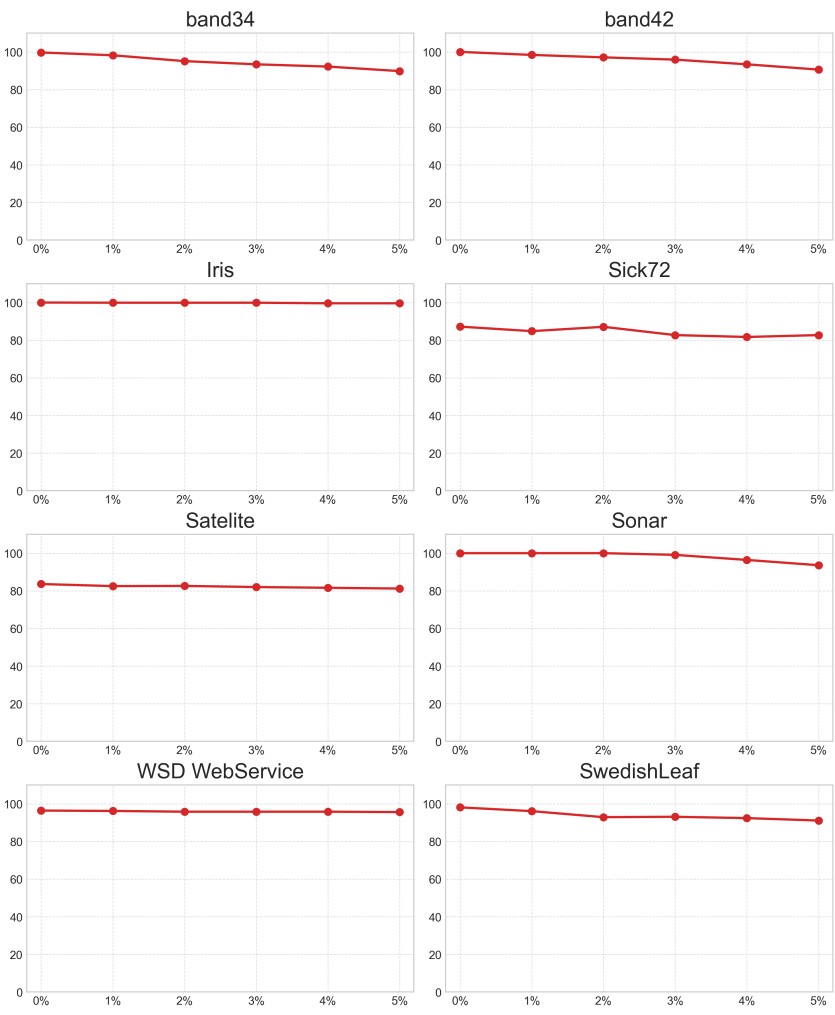

Figure 10: Robustness analysis on eight datasets under varying training data contamination rates (0% to 5%). PGBC demonstrates high stability across tabular, time series, and open-set tasks, with minimal performance loss in most scenarios.

# K  VISUALIZATION STUDIES

Beyond the quantitative results reported in the main text, we include visual comparisons that highlight the covering behavior of different granular-ball approaches. Figure 11 shows how probabilistic granular-balls, unlike traditional isotropic granular-balls, adapt their shape and orientation to local data structure. On synthetic datasets with curved or elongated clusters, this adaptivity allows PGBC to cover distributions with fewer and more compact components, providing an intuitive complement to the quantitative gains discussed earlier.

To further illustrate the refinement behavior of PGBC, we visualize the probabilistic granular-ball splitting process on four representative datasets: two tabular datasets (*abalone*, *pageblocks*), one benchmark time series dataset (*SMD Facility*), and one repurposed UCR dataset (*Synthetic Control*). For tabular data with large sample sizes and inherent clustering structures, PGBC progressively splits coarse coverings into compact ellipsoidal components that align with cluster boundaries (Figures 12 and 13). For time series data, embeddings often lie on smooth manifold-like trajectories; here PGBC adaptively stretches ellipsoids along principal directions, preserving continuity while capturing subtle deviations (Figure 14). For UCR datasets with multiple interleaved classes, PGBC refines granular-balls into anisotropic coverings that disentangle overlapping patterns and highlight out-of-distribution behaviors (Figure 15). Together, these visualizations demonstrate PGBC's ability to handle heterogeneous structures across static and temporal domains.

To complement the quantitative results, we further visualize anomaly detection outcomes on the six time series datasets used in our experiments. To generate the continuous point-level anomaly score curves shown in these figures, we aggregated the window-level outputs: specifically, the score for each time step is calculated by averaging the hierarchical log-likelihoods of all overlapping sliding windows covering that point. Each plot overlays the raw sequence with ground-truth anomalies and those detected by PGBC. As shown in Figure 16 and Figure 17, the detected anomalies closely follow the true labels, capturing both sharp point anomalies and subtle contextual deviations across diverse temporal settings.

Building on these insights, we further provide a specific case study of our method's anomaly detection performance on the UCR Adiac dataset, a benchmark for challenging time series. The visualization in Figure 18 provides a dual-panel view. The top panel compares the Dynamic Time Warping Barycenter Averaging (DBA) mean curves of normal samples, ground-truth anomalies, and the anomalies detected by PGBC. The close alignment of the detected anomaly mean curve with that of the ground-truth, and its significant divergence from the normal samples' mean, visually validates our method's ability to precisely capture the intrinsic patterns that define an anomaly. The bottom panel of Figure 18 overlays all samples, where the detected anomalies are shown to align perfectly with the ground-truth anomalies. This powerful visual evidence confirms that our method accurately pinpoints the exact locations and shapes of true anomalies, thus complementing the quantitative performance metrics with a clear demonstration of our method's high precision.

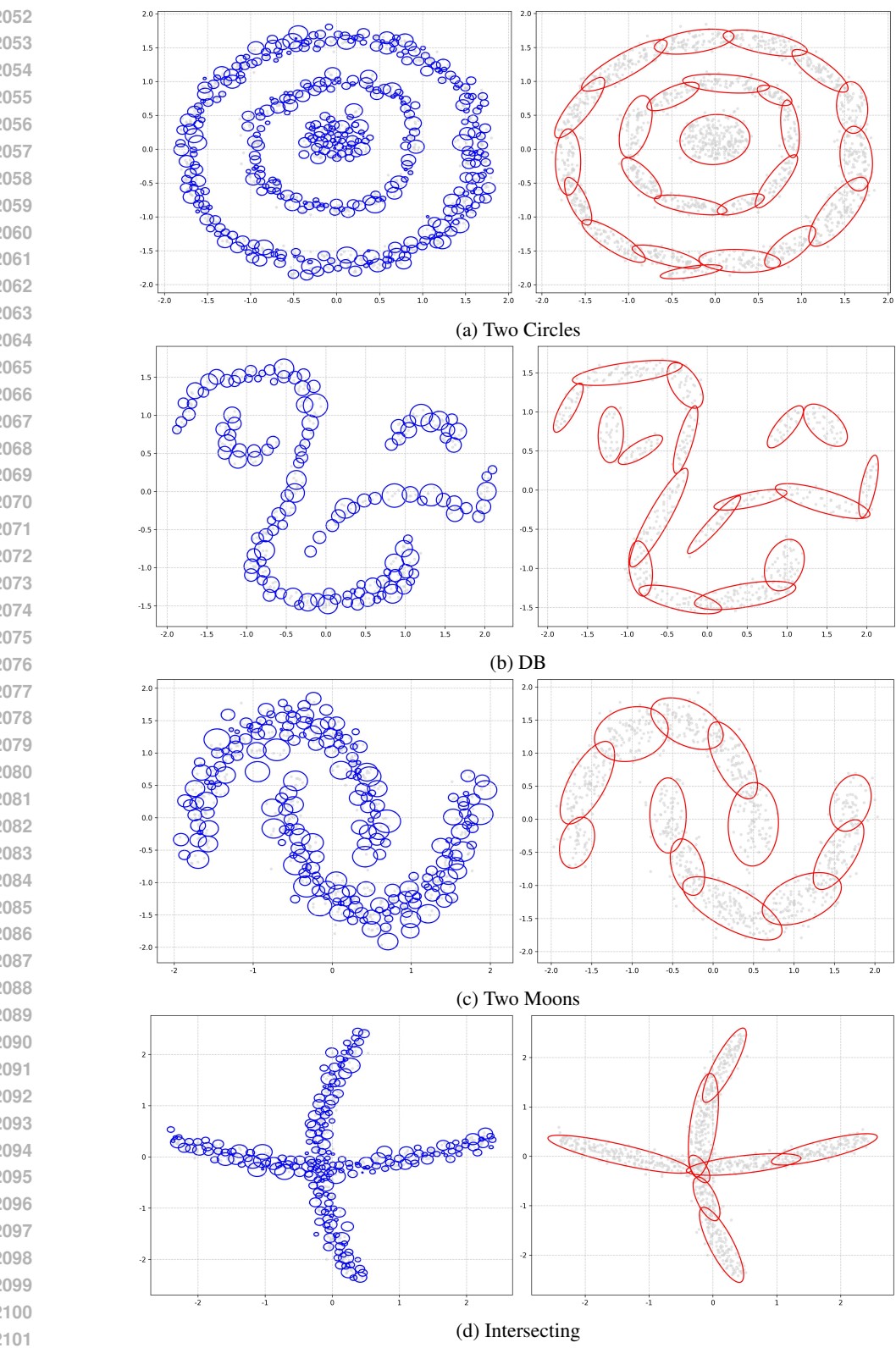

(a) Two Circles

(b) DB

(c) Two Moons

(d) Intersecting

Figure 11: Visualization of granular-ball generation on synthetic datasets. Comparison between traditional isotropic (left) and probabilistic ellipsoidal (right) granular-balls.

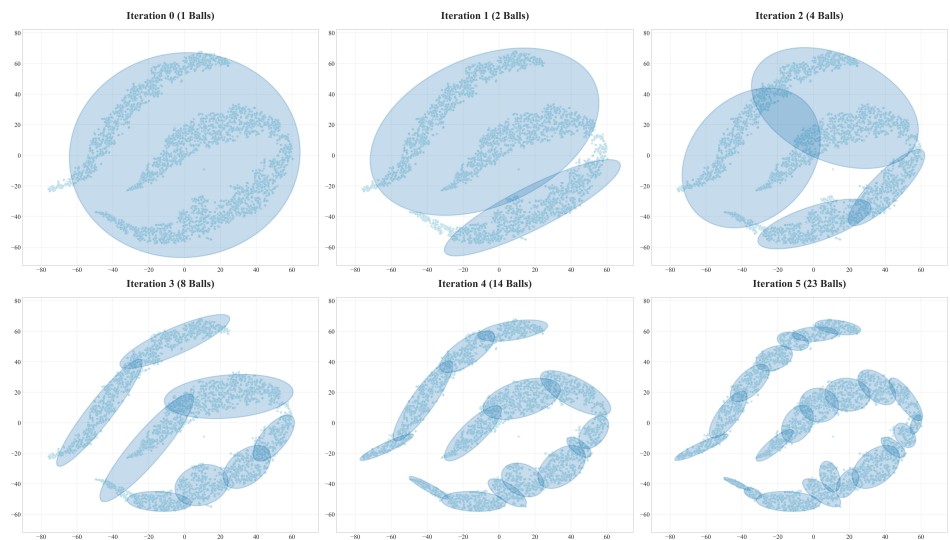

Figure 12: Visualization of the probabilistic splitting process on the *Abalone* dataset. PGBC refines coarse coverings into ellipsoids that align with natural group boundaries.

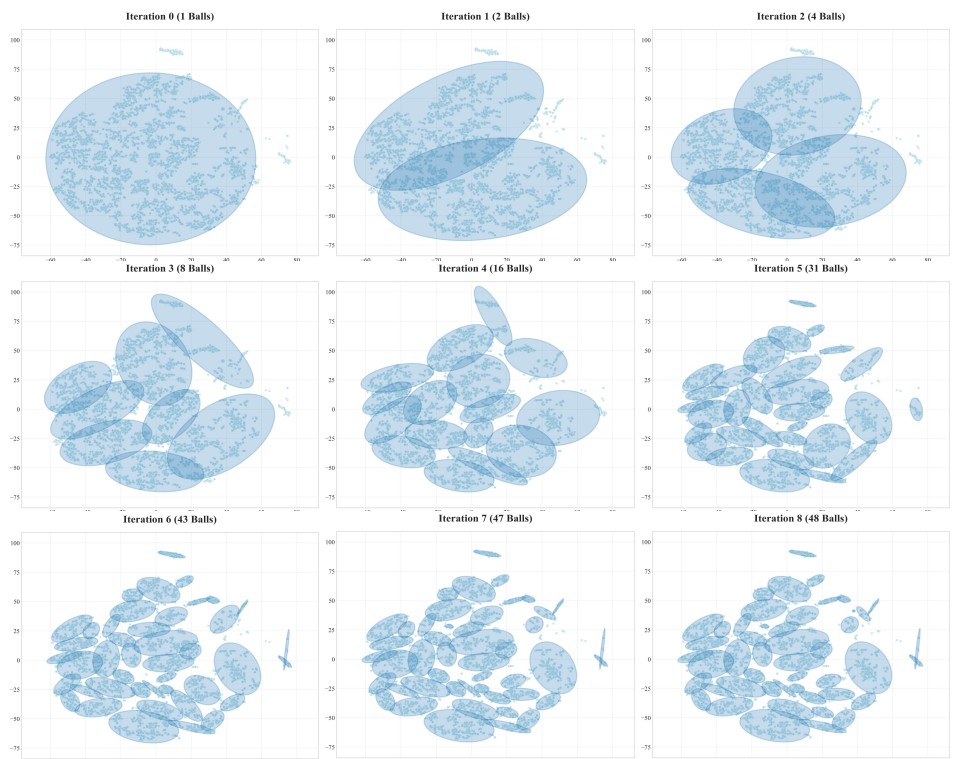

Figure 13: Visualization of the probabilistic splitting process on the *Pageblocks* dataset. PGBC adapts to heterogeneous distributions by stretching ellipsoids along anisotropic directions.

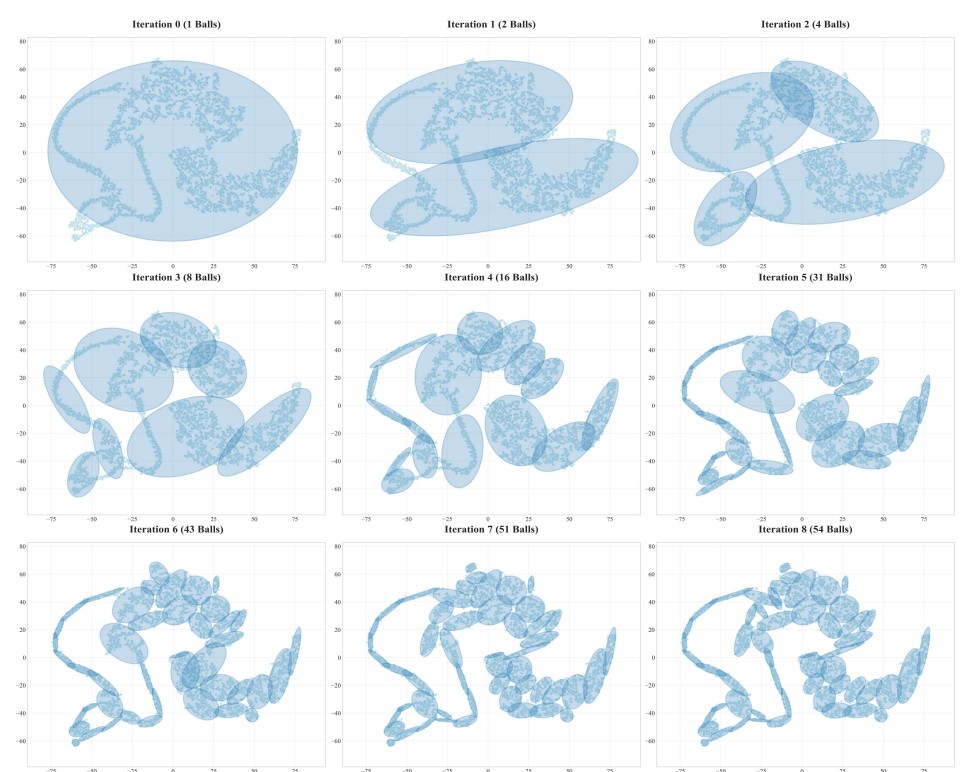

Figure 14: Visualization of the probabilistic splitting process on the *SMD Facility* time series dataset. PGBC aligns ellipsoids with smooth manifold-like embeddings, preserving temporal continuity.

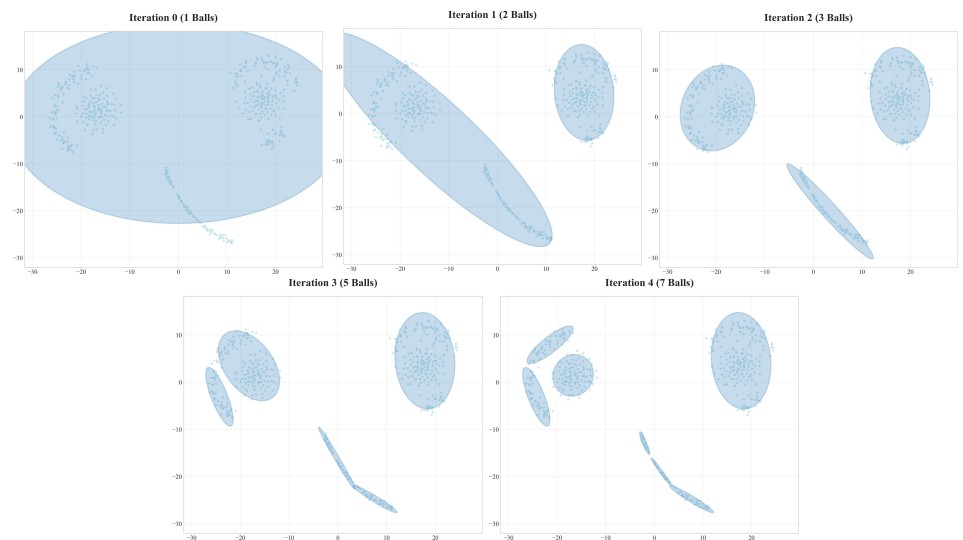

Figure 15: Visualization of the probabilistic splitting process on the *Synthetic Control* dataset. PGBC disentangles interleaved class structures by forming anisotropic coverings.

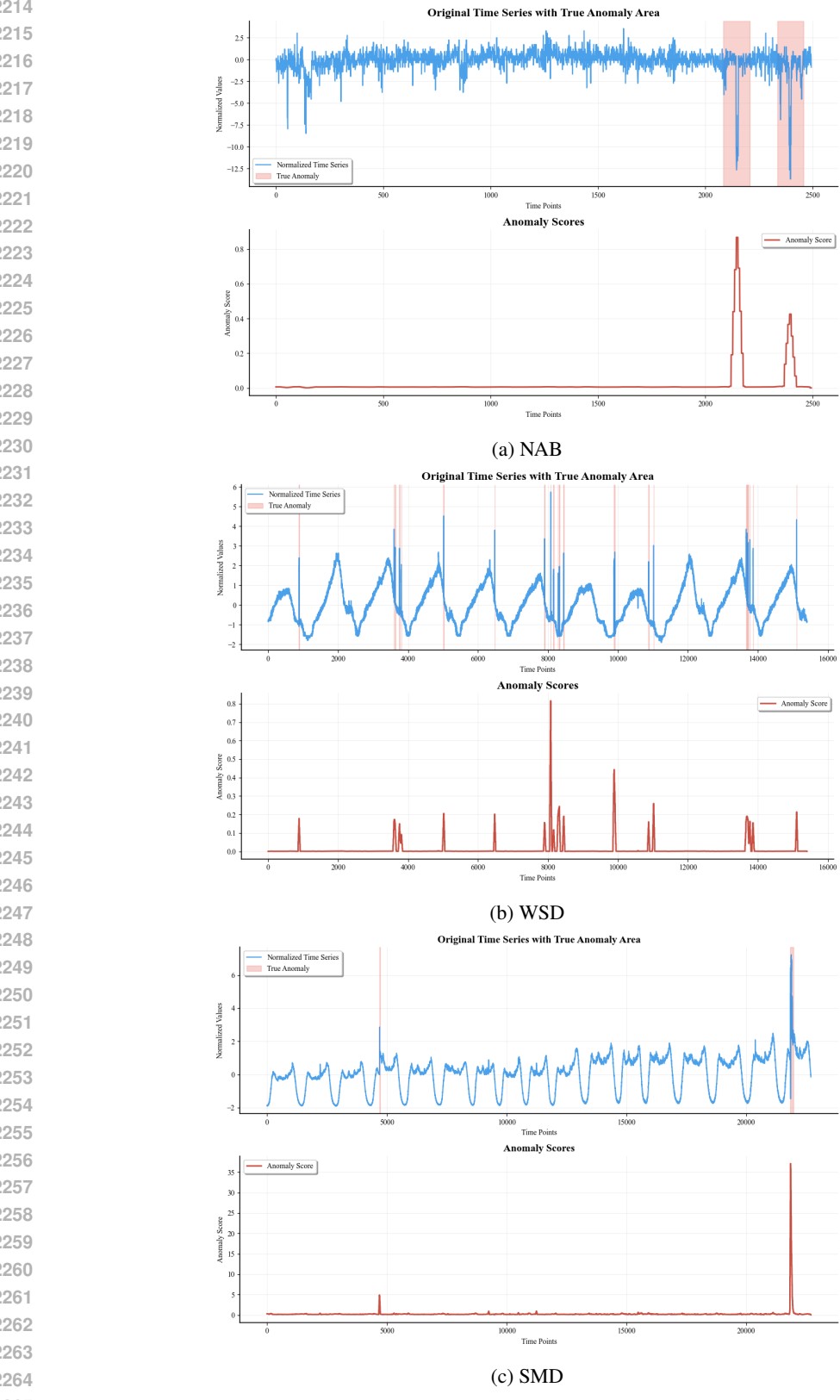

(a) NAB

(b) WSD

(c) SMD

Figure 16: Visualization of anomaly detection results on time series datasets (Part 1). (a) NAB, (b) WSD, and (c) SMD. Panels show raw sequences (blue), ground-truth anomalies (red shading/bars), and detected anomaly scores (red lines).

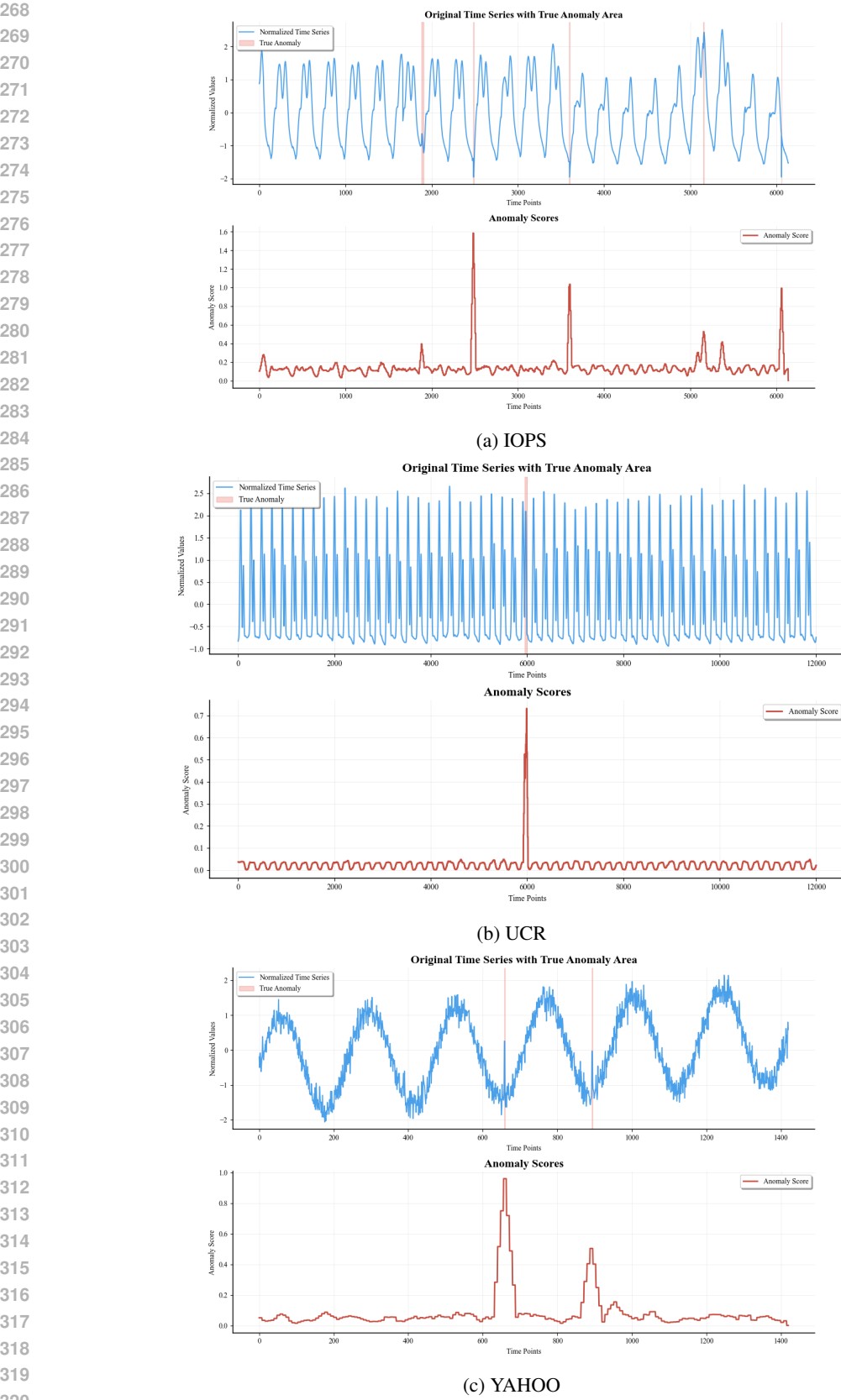

Figure 17: Visualization of anomaly detection results on time series datasets (Part 2). (a) IOPS, (b) UCR, and (c) YAHOO. Panels show raw sequences, ground-truth anomalies, and detected anomaly scores.

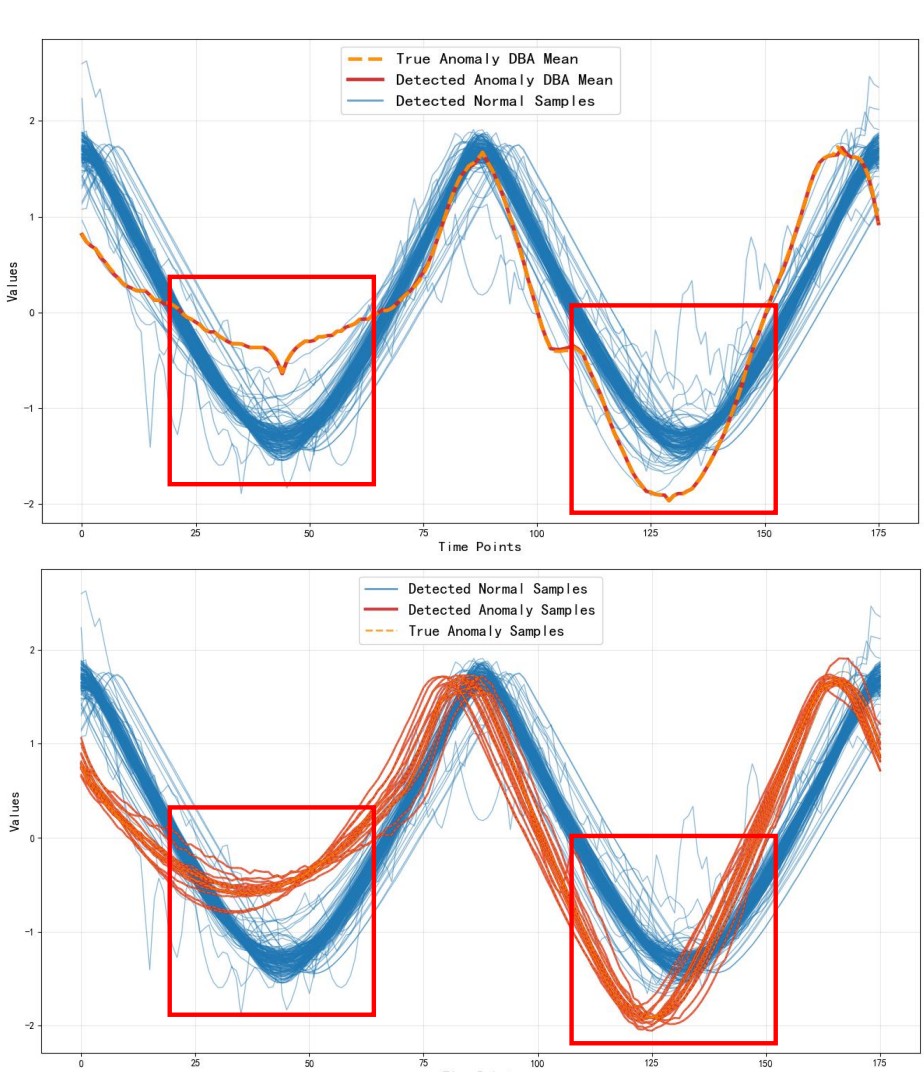

Figure 18: Case study on the UCR Adiac dataset. (Top) Comparison of DBA mean curves for normal, anomalous, and detected samples. (Bottom) Visualization of detected anomalies (red boxes) aligning with ground-truth patterns.

