# OpenReview forum: "Hierarchical One-Class Data Description via Probabilistic Granular-ball Computing"
_ICLR.cc/2026/Conference — Submitted to ICLR 2026_

### Official Review · Reviewer_ruXP · 2025-10-26

**Soundness:** 3
**Presentation:** 3
**Contribution:** 2
**Rating:** 6
**Confidence:** 2

**Summary:**

This paper introduces Probabilistic Granular-Ball Computing (PGBC), a novel framework for hierarchical one-class data description, particularly suited for anomaly detection. Unlike traditional methods that use simple spherical boundaries, PGBC constructs ellipsoidal granular-balls that adapt to the anisotropic geometry of real-world data. These granular-balls are refined recursively using statistical criteria, enabling multi-scale representation of normal data. PGBC also builds a hierarchical Gaussian mixture model, where anomaly scores are aggregated across layers using entropy-based weighting, allowing detection of both global and local deviations. Experiments show PGBC consistently outperforms strong baselines with higher accuracy and lower false positive rates, making it a robust solution for complex anomaly detection tasks. Experiments are conducted on teh

**Strengths:**

- The paper is generally well-written and well-organise. The structure and the motivation of the paper is clear.
- The idea of modeling anisotropic geometries is interesting.
- The paper evaluates the proposed method on various datasets and demonstrates its effectiveness.

**Weaknesses:**

- It is not clearly articulated how the proposed method significantly differs from or advances beyond GBDO in terms of conceptual innovation or technical contribution.
- The proposed PGBC framework relies on BIC and log-likelihood gain (LLG) as statistical criteria to determine whether a granular-ball should be recursively split. While these are standard model selection metrics, I wonder whether they are sufficient to robustly prevent over-partitioning, especially in high-dimensional.
- The experimental datasets used in the paper are primarily tabular. However, considering that DeepSVDD has been successfully applied to computer vision datasets such as CIFAR and MVTec—which are widely used benchmarks for one-class classification—I wonder whether the proposed method can be naturally extended to such scenarios. It would be interesting to explore how PGBC performs on these vision-based benchmarks.
- Would the proposed method remain effective if the training data is contaminated with a small proportion of anomalous samples?

**Questions:**

n/a

---

> ### Author Response · Authors · 2025-12-03
> **Response to Reviewer ruXP (Part 1)**
>
> We would like to sincerely thank Reviewer ruXP for providing a detailed review and insightful suggestions.
>
> **Q1: It is not clearly articulated how the proposed method significantly differs from or advances beyond GBDO in terms of conceptual innovation or technical contribution.**
>
> **A1:** We thank the reviewer for this insightful question.
>
> PGBC advances beyond GBDO both conceptually and technically.
>
> GBDO is a density-based granular-ball outlier detection method (Su et al., 2025) that serves as a coarse-grained extension of LOF. It replaces individual data points with isotropic granular-ball centers and incorporates fuzzy similarity between these centers to compute local outlier factors through relative reachability density. Thus, both LOF and GBDO are confined to the relative local-density paradigm, which requires the presence of outliers in the data for meaningful scoring. They are sensitive to the choice of k, assume isotropic distributions, and operate within a flat single-layer structure. As a result, they face significant limitations in capturing complex data geometries and defining appropriate boundaries in heterogeneous or anisotropic datasets.
>
> In contrast, PGBC transcends the entire density-based paradigm and introduces ellipsoidal granular-balls (parameterized by mean $\mu$ and full covariance $\Sigma$) to effectively capture anisotropic data geometry. Furthermore, it employs rigorous statistical splitting via BIC and log-likelihood gain for adaptive refinement and aggregates negative log-likelihood scores across a hierarchical Gaussian mixture tree with entropy-based weighting for multi-scale anomaly detection.
>
> These innovations enable PGBC to define tighter, more expressive boundaries without redundancy, as demonstrated by superior AUC, FPR/FNR, and F1 results compared to GBDO across all tabular and time-series benchmarks in Section 3.1-Section 3.4.
>
> **Q2: The proposed PGBC framework relies on BIC and log-likelihood gain (LLG) as statistical criteria to determine whether a granular-ball should be recursively split. While these are standard model selection metrics, I wonder whether they are sufficient to robustly prevent over-partitioning, especially in high-dimensional.**
>
> **A2:** We thank the reviewer for this thoughtful question.
>
> The combination of BIC and log-likelihood gain effectively prevents over-partitioning. BIC applies a strong penalty that increases quadratically with dimensionality, while the strict LLG requirement ensures that splits only occur when they significantly improve model fit.
>
> Ablation results in Appendix G.2 show that removing BIC alone leads to a dramatic increase in granular-balls (11x–63×) and worsens both efficiency and performance. Thus, this dual criterion is well-founded and essential for maintaining a compact and robust hierarchy across all tested settings.
>
> | Dataset | Full  | w/o BIC | Increase factor |
> | --- | --- | --- | --- |
> | Abalone | 25 | 668 | ×26.7 |
> | Bands34 | 10 | 312 | ×31.2 |
> | Bands42 | 14 | 161 | ×11.5 |
> | Cardio | 17 | 721 | ×42.4 |
> | Ecoli | 5 | 314 | ×62.8 |
> | NAB | 6 | 269 | ×44.8 |
> | WSD | 9 | 382 | ×42.4 |
>
> As for high-dimensional settings, such as time-series data, we also typically employ a pre-trained encoder to extract relevant features, which helps manage dimensionality and improve model performance.
> We have added this table and discussion to the revised Appendix G.3 for clarity.

---

> ### Author Response · Authors · 2025-12-03
> **Response to Reviewer ruXP (Part 2)**
>
> **Q3: The experimental datasets used in the paper are primarily tabular. However, considering that DeepSVDD has been successfully applied to computer vision datasets such as CIFAR and MVTec—which are widely used benchmarks for one-class classification—I wonder whether the proposed method can be naturally extended to such scenarios. It would be interesting to explore how PGBC performs on these vision-based benchmarks.**
>
> **A3:** We thank the reviewer for this excellent suggestion, which has significantly enriched the experiments in our revised paper.
>
> In response, we conducted additional experiments on three widely used visual benchmarks (**CIFAR-10**, **FashionMNIST**, and **MVTec-AD**) using ViT-pretrained features, following the ADBench protocol (512-dimensional). PGBC can be applied directly without any architectural changes; it simply treats the frozen visual features as input and builds the hierarchical ellipsoidal granular-ball model on top.
>
> As shown, PGBC consistently outperforms DeepSVDD by a substantial margin (over 25 AUC points on average) across general datasets. We attribute DeepSVDD's lower performance to the rigidity of its hyperspherical boundary in modeling complex anisotropic manifolds. This observation aligns with the comprehensive analysis in the Section4.2 of the ADBench benchmark (Han et al., NeurIPS 2022), which explicitly notes that unsupervised deep methods like DeepSVDD are often "surprisingly worse than shallow methods." As the authors of ADBench point out, without the guidance of label information, these models are "harder to train" and significantly "more difficult to tune hyperparameters," leading to unsatisfactory performance. Furthermore, on the challenging MVTec-AD benchmark, PGBC surpasses both the recent probabilistic baseline (HGAD) and vision-specific methods like CutPaste and P-SVDD.
>
> | **Datasets (Avg AUC %)** | **DeepSVDD** | **CutPaste** | **HGAD** | **PGBC (Ours)** |
> | --- | --- | --- | --- | --- |
> | **CIFAR-10** | 69.5 | 73.3 | 95.3 | **95.5** |
> | **FashionMNIST** | 58.5 | 70.1 | 94.6 | **94.9** |
>
> | **Dataset (Avg AUC %)** | **HGAD** | **CutPaste** | **U-Student** | **P-SVDD** | **PGBC (Ours)** |
> | --- | --- | --- | --- | --- | --- |
> | **MVTec-AD** | 91.2 | 90.9 | 92.5 | 92.1 | **93.0** |
>
> **Q4: Would the proposed method remain effective if the training data is contaminated with a small proportion of anomalous samples?**
>
> **A4:** We thank the reviewer for this practical question. Yes, PGBC exhibits strong robustness to training-set contamination. As reported in Appendix J of the revised paper, we conducted a comprehensive robustness analysis by injecting 0%–5% anomalous samples into the training sets of eight diverse datasets.
>
> As shown in Figure 10, PGBC demonstrates remarkable resilience, with negligible performance drops (e.g., <1% on *Iris* and *WSD*) even at 5% contamination. This robustness arises naturally from our BIC-based splitting criterion, which acts as a statistical regularizer to effectively isolate sparse outliers, preventing them from distorting the learned density hierarchy.

---

### Official Review · Reviewer_5XJV · 2025-10-29

**Soundness:** 2
**Presentation:** 3
**Contribution:** 2
**Rating:** 4
**Confidence:** 4

**Summary:**

This work proposes PGBC (Probabilistic Granular ball Computing) by combining the GB tree with Gaussian, to describe intrinsic structures and patterns of the data. Grounded in the nature of PGBC for one-class description, this work designs a hierarchical anomaly score for anomaly detection task. The empirical evaluation on tabular and time-series demonstrates the effectiveness of the proposed method.

**Strengths:**

1.	The work is well-written with clear organization, statement and illustrations.
2.	The proposed method is theoretically reasonable for anomaly detection under one class classification.
3.	The evaluation given in the manuscript seems to be convinced and could demonstrate the superior detection performance to some extent.

**Weaknesses:**

1.	There is a defect for current version of the PGBC. Based the pseudo-code of hierarchical anomaly scoring (Algorithm 2), this algorithm only works when $l \geq 1$ that means it is ill-defined for the Granular ball Queue $\mathcal{Q} = [B^{(0)}]$. The case is possible, right? In other words, current algorithm is valid only for the data with more intrinsic structures. This is crucial for the rigor and completeness of the proposed algorithm.

2.	The time complexity (and real implementation cost) both on the training and inference has not been analyzed and compared. From my view, the time cost maybe is a big problem for the proposed method both in training and inference since there is an iterative GMM process during the training and iterative density estimation during the inference.

3.	There is a lack for some key ablation and exploration analysis.
 - 1). What is the relation between detection performance and the depth of GB Tree or the number of granular balls? If such analysis can help to decrease the inference time cost?
 - 2).  Authors mention the importance of the dynamic reassignment. The related ablation study should be included to provide the empirical evidence.
 - 3).  what if only use the granular balls as leaves to compute the anomaly score?
 - 4).  The proposed method is developed from GBC. Consequently, the GBC should be included as a baseline and the anomaly score should attempt the vanilla version (showing in Figure 1) and another simple strategy (the average distance to the centroids of all granular balls)


4.	Unconvinced empirical evaluation and findings in Section 3.5.

The findings presented in Section 3.5 may lack generalizability, primarily due to the limited scope and specialized nature of the datasets used. The tabular datasets in Table 1 maybe are better choice for demonstrating performance against the GBC.

5.	Limited experiments

If possible, complementing experiments on more benchmark datasets, (like ADBench) to improve the statistical reliability.

**Questions:**

1.	The issue illustrated in Figure 1 seems to be overcame easily if calculating anomaly score using the average distance to the centroids of all granular balls. The proposed method needs to more suit-well motivation.

2.	Why the hierarchical anomaly scoring is important? Some empirical evaluation should be provided if without theoretical guarantees.

3.	The normalization for $s^{(l)}$ is necessary? and why?

4.	Authors mentioned that the proposed method captures both global and local features, but based on the design of algorithm 2, the entropy weight for $B^{(0)}$ is always zero (of course, I noticed that the $B^{(0)}$ is eliminated in algorithms 2 which also leads to other issue). So, how to understand the global feature and it is possible to utilize the $B^{(0)}$ reasonably?

5.	Where is randomness from for LOF, KNN and the proposed method PGBC since I noticed that the related hyperparameters are fixed?

6.	I understand the experiments on tabular datasets, but how to obtain the results on time-series showing in Figure 4? It seems to yield the anomaly score in each time point. Please provide more explanation to help me understand this part.

7.	For the task: time series open-set recognition, I don’t think this is a different task compared to the anomaly detection on the tabular datasets, particularly the embeddings of data are used in experiments.

8.	More explanation about the statement “these findings highlights …” in line 430-432.

9.	While the FPR is a crucial metric for anomaly detection systems, the FNR (False Negative Rate) is often more of greater practical significance in real applications, like medical diagnosis, intrusion detection in cybersecurity. I suggest including a FNR analysis for the proposed method.

**Minor errors:**

By aligning Figure 2 with Algorithm 1, the $B^{(1)}_j$ in Figure 1 should be $B^{(0)}_j$, right?

---

> ### Author Response · Authors · 2025-12-03
> **Response to Reviewer 5XJV (Part 1)**
>
> We are grateful to Reviewer 5XJV for the insightful comments, which were instrumental in refining our algorithm logic and experimental scope. We have addressed each of the concerns raised by the reviewer, as outlined below.
>
> **Q1: There is a defect in the current version of the PGBC. Based on the pseudo-code of hierarchical anomaly scoring (Algorithm 2), this algorithm only works when $L≥1$﻿ — that means it is ill-defined for the Granular ball Queue $Q=[B^{(0)}]$. This case is possible, right? In other words, the current algorithm is valid only for data with more intrinsic structures. This is crucial for the rigor and completeness of the proposed algorithm.**
>
> **A1:** We sincerely thank you for your thorough review and professional suggestions.
>
> Thank you for your careful reading and insightful feedback. In fact, the case you are concerned about occurs when $l=0$ (no splitting), which corresponds to a single principal component, as mentioned by Reviewer 2hzn. In this scenario, we can simply use a single global Gaussian distribution for inference. To ensure rigor and completeness, we have improved Algorithm 2 (lines 1-3) in the revised paper to explicitly include a conditional check for the $L=0$  case.
>
> Additionally, it's worth noting that the global anomaly patterns captured by $B^{(0)}$ are typically simple outliers. In a hierarchical setting ($L≥1$), these points remain distinct from the finer-grained distributions and are thus easily detected by the subsequent granular-balls at levels $l ≥1$.
>
> **Q2: The time complexity (and real implementation cost) both on the training and inference has not been analyzed and compared. From my view, the time cost maybe is a big problem for the proposed method both in training and inference since tere is an iterative GMM process during the training and iterative density estimation during the inference.**
>
> **A2:** As suggested, we provide both training and inference results on the largest tabular and time series datasets in the table below.
>
> As can be seen, PGBC achieves the lowest total runtime across all baselines. Specifically on the *SMD Facility* dataset, PGBC (8.68s) is **3x faster** than the hierarchical probabilistic method HGAD (27.19s) and significantly faster than deep baselines like AutoEncoder (15.87s). This demonstrates that our statistical splitting converges significantly faster than the gradient-descent-based optimization required by deep models, and the overhead from iterative GMM fitting is well-controlled (outperforming non-probabilistic GBDO).
>
> Regarding inference, PGBC incurs negligible latency (e.g., 0.01s), effectively matching simple AutoEncoders while being orders of magnitude faster than diffusion-based methods like DiffAD.
>
> We have included the full runtime results and a detailed complexity analysis in Appendix I.2 of the revised paper.
>
> | **Datasets** | **Metric (s)** | **AE** | **DeepSVDD** | **HGAD** | **GBDO** | **THOC** | **DiffAD** | **PGBC (Ours)** |
> | --- | --- | --- | --- | --- | --- | --- | --- | --- |
> | **SMD Facility** | **Training** | 15.86 | 11.24 | 27.18 | 21.14 | 126.17 | 68.29 | **8.67** |
> |  | **Inference** | 0.01 | 0.05 | 0.01 | 0.09 | 1.16 | 69.94 | **0.01** |
> |  | *Total* | *15.87* | *11.29* | *27.19* | *21.23* | *127.33* | *138.23* | ***8.68*** |
> | **Thyroid** | **Training** | 40.73 | 24.79 | 62.21 | 37.09 | - | - | **21.09** |
> |  | **Inference** | 0.03 | 0.06 | 0.01 | 0.12 | - | - | **0.04** |
> |  | *Total* | *40.77* | *24.85* | *62.22* | *37.22* | - | - | ***21.12*** |

---

> ### Author Response · Authors · 2025-12-03
> **Response to Reviewer 5XJV (Part 2)**
>
> **Q3: There is a lack for some key ablation and exploration analysis.**
>
> **1). What is the relation between detection performance and the depth of GB Tree or the number of granular balls? If such analysis can help to decrease the inference time cost?**
>
> **A3.1:** Note that the proposed PGBC framework adaptively determines the tree depth and the number of granular-balls needed to fit the normal data distribution (using BIC), rather than treating these as fixed hyperparameters. Indeed, the relationship between depth and performance is influenced by the granularity of the unknown anomalies present in the test set.
>
> For instance, as shown in Figure 7 (Appendix H.1 of the revised paper), performance on the *Thyroid* dataset peaks at intermediate layers (77.7% at L6) but declines at deeper levels (67.1% at L11). In contrast, datasets like *Tictac12* show improved performance with greater depth, achieving 100.0% only at the final layer (L6).
>
> This illustrates that relying on a single granularity or a "flat" structure is inherently unstable. Our hierarchical aggregation strategy effectively addresses this risk by integrating information across multiple levels, ensuring robust detection for both coarse (global) and fine-grained (local) anomalies.
>
> | Layers | **L1** | **...** | **L6** | **...** | **L10** | **L11** | **L2** | **Aggregated** |
> | --- | --- | --- | --- | --- | --- | --- | --- | --- |
> | **Thyroid(AUC%)** | 68.1 | ... | **77.7** | ... | 70.8 | 67.4 | 67.1 | **76.6** |
>
> | Laters | L1 | L2 | L3 | L4 | L5 | L6 | **Aggregated** |
> | --- | --- | --- | --- | --- | --- | --- | --- |
> | **Tictac12(AUC%)** | 97.7 | 90.6 | 96.7 | 98.4 | 99.4 | 100.0 | **100.0** |
>
> **2). Authors mention the importance of the dynamic reassignment. The related ablation study should be included to provide the empirical evidence.**
>
> **A3.2:** As suggested, we study the effects of the dynamic reassignment in the table below. Full results can be found in the Appendix G.1 of the revised paper.
>
> As can bee seen, dynamic reassignment allows granular-balls to better adapt to the local data geometry by correcting assignment errors from earlier splits. While it introduces a slight computational overhead, the substantial gain in precision (e.g., +3.6% AUC on Cardio) justifies the design.
>
> | **Datasets** | **Reassign** | **AUC (%)** | **FPR (%)** |
> | --- | --- | --- | --- |
> | **Cardio** | × | 80.41 | 1.74 |
> |  | ✓ | **84.04** | **1.63** |
> | **Abalone** | × | 70.46 | 1.78 |
> |  | ✓ | **72.10** | **1.72** |
> | **SwedishLeaf** | × | 98.10 | 2.10 |
> |  | ✓ | **98.66** | **1.81** |
>
> **3). what if only use the granular balls as leaves to compute the anomaly score?**
>
> **A3.3:** As suggested, we compare the variant of using leaf balls to compute the anomaly scores. Results are summarized in the table below. As can be seen, entropy-weighted aggregation effectively stabilizes performance by leveraging complementary multi-scale information. That is better than a single-layer decision.
>
> In the *Thyroid* dataset, we report the layer-wise AUC performance in the table below to further help the reviewer understand. As observed, detection accuracy fluctuates significantly, peaking at intermediate layers (77.7% at Layer 6) but dropping sharply at the leaf level (67.1% at Layer 12). Crucially, our Aggregated score (76.6%) successfully approximates the optimal single-layer performance without manual selection. Full results are visualized in Figure 6 of Appendix G.2 in the revised paper.
>
> | **Metric** | **Layer 1** | **...** | **Layer 6** | **...** | **Layer 10** | **Layer 11** | **Layer 12** | **Aggregated** |
> | --- | --- | --- | --- | --- | --- | --- | --- | --- |
> | **AUC (%)** | 68.1 | ... | **77.7** | ... | 70.8 | 67.4 | 67.1 | **76.6** |

---

> ### Author Response · Authors · 2025-12-03
> **Response to Reviewer 5XJV (Part 3)**
>
> **Q3: There is a lack for some key ablation and exploration analysis.**
>
> **4). The proposed method is developed from GBC. Consequently, the GBC should be included as a baseline and the anomaly score should attempt the vanilla version (showing in Figure 1) and another simple strategy (the average distance to the centroids of all granular balls)**
>
> **A3.4:** Thank you for your constructive suggestion. Note that we have already compared our method against state-of-the-art GBC-based anomaly detection methods (**GBDO**[1] and **GBMOD**[2]) in our original submission (Table 1-3).
>
> | **Tasks (Avg AUC %)** | **GBDO [1]** | **GBMOD [2]** | **PGBC (Ours)** |
> | --- | --- | --- | --- |
> | **Tabular AD** | 71.4 | 87.1 | **91.1** |
> | **Time Series AD** | 74.1 | 84.0 | **95.3** |
> | **Time Series OSR** | 77.2 | 88.8 | **99.6** |
>
> To further address your concern, we conducted an additional ablation study comparing PGBC with GBC with two suggested distance-based strategies:
>
> - **Min-Distance (Vanilla version):** Calculates the anomaly score based on the Euclidean distance to the nearest granular-ball center. This corresponds to the traditional isotropic (spherical) granular-ball assumption.
> - **Avg-Distance:** Calculates the anomaly score based on the average Euclidean distance to all granular-ball centers, incorporating global geometric information.
>
> | **Datasets** | **Min-Distance** | **Avg-Distance** | **PGBC (Ours)** |
> | --- | --- | --- | --- |
> | **Bands34** | 83.9 ± 1.8 | 81.3 ± 1.4 | **100.0 ± 0.0** |
> | **Bands42** | 81.8 ± 2.6 | 75.6 ± 3.8 | **100.0 ± 0.0** |
> | **Ecoli** | 79.5 ± 2.8 | 84.3 ± 0.4 | **89.4 ± 0.4** |
> | **Sick72** | 80.9 ± 1.4 | 79.6 ± 0.5 | **87.3 ± 3.6** |
> | **Thyroid** | 56.2 ± 1.5 | 67.6 ± 0.2 | **73.0 ± 4.8** |
> | **Waveform** | 71.6 ± 0.0 | 68.8 ± 0.0 | **78.1 ± 0.0** |
>
> As shown, PGBC consistently outperforms both distance-based strategies. This is because traditional granular-balls (represented by Min-Distance) assume isotropic (spherical) shapes, which ignore the principal component directions of the data. To cover complex anisotropic regions, traditional GBC often requires excessive splitting, leading to redundancy and suboptimal boundaries.
> Moreover, Avg-Distance tends to be unstable (e.g., performing worse on *Bands42*) as averaging distances to all clusters introduces global noise that dilutes local anomaly signals.
>
> PGBC addresses these limitations by leveraging probabilistic covariance matrices, allowing the granular-balls to adaptively stretch along the data's principal directions. This results in a more precise and compact data description that captures local density variations effectively.
>
> We have included these detailed comparisons in Appendix H.3 of the revised paper.
>
> [1] Identifying Outliers via Local Granular-Ball Density, *IEEE Transactions on Neural Networks and Learning Systems*, 2025.
>
> [2] GBMOD: A Granular-Ball Mean-shift Outlier Detector, *Pattern Recognition*, 2025.
>
> **Q4: Unconvinced empirical evaluation and findings in Section 3.5. The findings presented in Section 3.5 may lack generalizability, primarily due to the limited scope and specialized nature of the datasets used. The tabular datasets in Table 1 maybe are better choice for demonstrating performance against the GBC.**
>
> **A4:** In response, we have made revisions to address your concerns: we have moved the synthetic analysis to Appendix I.3 of the revised paper and completely rewritten Section 3.5 to focus on all 19 real-world tabular datasets.
>
> We compared the number of components produced by traditional GBC and PGBC. The partial results below demonstrate that PGBC achieves a substantial reduction in model complexity (up to 41.9x) by effectively aligning ellipsoids with the data geometry.
>
> This confirms that PGBC successfully covers anisotropic regions without excessive recursive splitting. We have visualized these results in Figure 5 (Main Text) and provided a detailed histogram in Appendix I.3 of the revised paper.
>
> | **Datasets** | **GBC Components** | **PGBC Components** | **Reduction Factor** |
> | --- | --- | --- | --- |
> | **Musk** | 671 | **16** | **41.9x** |
> | **Waveform** | 740 | **23** | **32.2x** |
> | **Sick72** | 1009 | **44** | **22.9x** |
> | **Cardio** | 395 | **30** | **13.2x** |

---

> ### Author Response · Authors · 2025-12-03
> **Response to Reviewer 5XJV (Part 4)**
>
> **Q5: Limited experiments：If possible, complementing experiments on more benchmark datasets, (like ADBench) to improve the statistical reliability.**
>
> **A5:** We agree that expanding the evaluation enhances the statistical reliability of our work.
>
> As suggested, we extended our evaluation to the visual domain following the standard **ADBench** protocol (Han et al., 2022). We utilized pre-trained Vision Transformer (ViT) embeddings as input features to benchmark performance on **CIFAR-10**, **FashionMNIST**, and the challenging industrial anomaly detection dataset **MVTec-AD**. Full detailed results and experimental settings are provided in Appendix F of the revised paper.
>
> As can be seen, PGBC consistently outperforms the strong probabilistic baseline (HGAD) across all tasks. Crucially, on the challenging MVTec-AD benchmark, PGBC (93.0%) surpasses specialized vision methods like **CutPaste**, **U-Student** and **P-SVDD**, demonstrating the effectiveness of our hierarchical probabilistic framework in modeling complex, high-dimensional feature manifolds.
>
> | **Datasets (AUC %)** | **DeepSVDD** | **GBMOD** | **HGAD** | **PGBC (Ours)** |
> | --- | --- | --- | --- | --- |
> | **CIFAR-10** | 69.5 | 94.6 | 95.3 | **95.5** |
> | **FashionMNIST** | 58.5 | 92.5 | 94.6 | **94.9** |
>
> | **Dataset (AUC %)** | **HGAD** | **CutPaste** | **U-Student** | **P-SVDD** | **PGBC (Ours)** |
> | --- | --- | --- | --- | --- | --- |
> | **MVTec-AD** | 91.2 | 90.9 | 92.5 | 92.1 | **93.0** |
>
> **Q6: The issue illustrated in Figure 1 seems to be overcame easily if calculating anomaly score using the average distance to the centroids of all granular balls. The proposed method needs to more suit-well motivation.**
>
> **A6:** Thank you for your insightful question. The core issue illustrated in Figure 1 lies in the limitations of **distance-based anomaly scores**. In anisotropic distributions, the **iso-probability contours** (density) often differ from the **iso-distance contours** (geometry).
>
> Using simple Euclidean distance, even when averaged across multiple centers, treats all directions equally. This approach fails to account for the directional decay of density, potentially introducing irrelevant noise that distorts anomaly detection.
>
> In contrast, PGBC employs a likelihood-based scoring system that leverages the covariance matrix, allowing it to accurately model anisotropic structures and capture true density distribution. Thus, effective anomaly detection relies on probabilistic density modeling rather than solely on geometric measurements.
>
> To further clarify your concern, we included a comparative analysis of methods based on average distances in our response to your Q4, demonstrating the effectiveness of our approach.
>
> **Q7: Why the hierarchical anomaly scoring is important? Some empirical evaluation should be provided if without theoretical guarantees.**
>
> **A7:** The importance of hierarchical anomaly scoring lies in its ability to integrate multi-granularity information, enhancing robustness in anomaly detection.
>
> As demonstrated in our responses to your Q3.1 and Q3.3 (full results are in Appendix H.1 of the revised paper), the hierarchical approach consistently outperforms the "Leaf-only" baseline in our ablation study (Q3.3). This indicates that intermediate layers contain critical structural cues that are lost when relying solely on the finest granularity.
>
> Furthermore, our layer-wise analysis on the Thyroid dataset (Q3.1) reveals that performance at a single resolution (such as using leaf balls) does not improve monotonically with depth. By aggregating scores across all levels, PGBC effectively mitigates the risk of depending on a suboptimal single resolution, ensuring stable and accurate detection across diverse data structures.
> We have incorporated the above discussion into lines 1692-1695 of the manuscript. Thank you for your suggestion.

---

> ### Author Response · Authors · 2025-12-03
> **Response to Reviewer 5XJV (Part 5)**
>
> **Q8: The normalization for *s*(*l*)**﻿ **is necessary? and why?**
>
> **A8:** Yes, normalization is essential for our method. The raw log-likelihood scores s(*l*) can vary significantly in magnitude due to differences in covariance determinants at different hierarchical levels. Without normalization, higher-magnitude scores could disproportionately influence the aggregated result, overshadowing contributions from lower-magnitude levels. By normalizing these scores, we ensure that all levels contribute equally to the final output, thereby enhancing the overall robustness and reliability of the anomaly detection process.
>
> This necessity for normalization is supported by our ablation study detailed in Appendix G.2 of the revised paper. The results, shown in the table below, demonstrate that omitting the normalization step ("w/o Norm") consistently leads to performance degradation and increased variance across various datasets.
>
> | **Datasets (AUC %)** | **w/o Norm** | **PGBC (Ours)** |
> | --- | --- | --- |
> | **Bands42** | 99.1 ± 0.7 | **100.0 ± 0.0** |
> | **Ecoli** | 88.8 ± 0.3 | **89.4 ± 0.4** |
> | **Sick72** | 81.2 ± 1.8 | **87.3 ± 3.6** |
> | **Thyroid** | 71.3 ± 6.5 | **73.0 ± 4.8** |
>
> **Q9: Authors mentioned that the proposed method captures both global and local features, but based on the design of algorithm 2, the entropy weight for $B^{(0)}$ is always zero (of course, I noticed that the $B^{(0)}$ is eliminated in algorithms 2 which also leads to other issue). So, how to understand the global feature and it is possible to utilize the $B^{(0)}$ reasonably**
>
> **A9:** Thank you very much for your careful reading and thoughtful consideration.
>
> Generally, $B^{(0)}$ is not utilized because, if it does not split at layer=0 (indicating **a single principal component**, as mentioned by Reviewer 2hzn), the method effectively reduces to fitting a single Gaussian distribution for inference.
>
> Additionally, the anomalies captured by $B^{(0)}$ are often straightforward points that lie significantly outside the main cluster. These simple anomalies can be easily detected in subsequent levels.
>
> To ensure completeness, we have included this conditional check for $B^{(0)}$ in lines 1-3 of Algorithm 2 in the revised paper. We appreciate your suggestion, as it significantly enhances the thoroughness of our approach.
>
> **Q10: Where is randomness from for LOF, KNN and the proposed method PGBC since I noticed that the related hyperparameters are fixed?**
>
> **A10:** To ensure a statistically rigorous evaluation, we introduced randomness through random subsampling (Bootstrap) of the training data, applying this approach uniformly across all methods, including LOF, KNN, and PGBC. Each baseline was tested on the same sampled training set across five runs. This consistent sampling allows for a fair comparison of performance. Additionally, we utilized multiple random seeds in each run to calculate the mean and standard deviation of the results, providing insights into the models' stability and behavior under varying conditions.
>
> We have clarified this experimental setting in Appendix C.2 (lines 921-923) of the revised manuscript.
>
> **Q11: I understand the experiments on tabular datasets, but how to obtain the results on time series, as shown in Figure 4? It seems to yield the anomaly score in each time point. Please provide more explanation to help me understand this part.**
>
> **A11:** To obtain the point-level anomaly scores in the time series shown in Figure 4, we first convert the time series into semantic feature embeddings using a sliding overlapping window strategy with a pre-trained encoder. PGBC then calculates the anomaly score, specifically the hierarchical negative log-likelihood, for each window embedding. Finally, we average the scores of all overlapping windows that cover specific time points, resulting in the final point-level anomaly curve.
>
> We have added the relevant details to Appendix K (lines 2018-2021) of the revised paper.
>
> **Q12: For the task: time series open-set recognition, I don’t think this is a different task compared to the anomaly detection on the tabular datasets, particularly the embeddings of data are used in experiments.**
>
> **A12:** Fundamentally, both time series open-set recognition and tabular anomaly detection focus on identifying new anomalies or unrecognized classes, especially as both rely on embeddings. While their objectives are similar at a high level, time series data brings its own complexities, including temporal consistency and intricate multivariate relationships.
>
> To address this, we incorporated the time series open-set recognition task into our study. This allows us to evaluate performance across diverse scenarios and gain insights into how models adapt to different data types.

---

> ### Author Response · Authors · 2025-12-03
> **Response to Reviewer 5XJV (Part 6)**
>
> **Q13: More explanation about the statement “these findings highlights …” in line 430-432.**
>
> **A13:** The statement “these findings highlight PGBC’s capacity to generalize beyond the training distribution” underscores the model's effectiveness in handling data that lies outside the known training set. In Figure 3(c), PGBC showcases its adaptability by employing ellipsoidal granular balls to accurately encompass the overlapping clusters in the Synthetic Control dataset.
>
> This adaptability is particularly crucial for one-class data scenarios, where the model must identify anomalies within a predominantly single-class dataset. By establishing flexible and discriminative boundaries, PGBC effectively accommodates the complexities of interleaved clusters, enabling it to recognize and differentiate patterns that may not have been explicitly represented in the training data. This capability significantly enhances its robustness for real-world applications, where novel patterns may emerge.
>
> We have incorporated this detailed explanation into Section 3.3 (lines 458-465) of the revised manuscript to facilitate further understanding.
>
> **Q14: While the FPR is a crucial metric for anomaly detection systems, the FNR (False Negative Rate) is often more of greater practical significance in real applications, like medical diagnosis, intrusion detection in cybersecurity. I suggest including a FNR analysis for the proposed method.**
>
> **A14:** We thank the reviewer for this valuable suggestion. In response, we have expanded Section 3.4 to jointly analyze both the False Positive Rate (FPR) and the False Negative Rate (FNR), restructuring the section for a more balanced discussion. Complete results for both metrics on a per-dataset basis are now included in **Appendix D (Tables 14 and 15)** of the revised paper.
>
> This revision highlights PGBC's effectiveness in minimizing missed anomalies (FNR) while also keeping false alarms (FPR) low. PGBC achieves the lowest FNR alongside the lowest FPR, demonstrating that its ellipsoidal granular-balls and entropy-weighted hierarchical scoring create tighter and more inclusive decision boundaries compared to both probabilistic (HGAD) and deep learning (DeepSVDD) baselines. Detailed per-dataset results and additional comparisons are available in Appendix D of the revised paper.
>
> **Q15: By aligning Figure 2 with Algorithm 1, the $𝐵_𝑗^{(1)}$ in Figure 1 should be $𝐵_𝑗﻿^{( 0 )}$ , right?**
>
> **A15:** Yes. The $𝐵_𝑗^{(1)}$ in original Figure 1 should be $𝐵_𝑗﻿^{( 0 )}$. We have corrected it in the revised paper, to align with Algorithm 1.

---

### Official Review · Reviewer_GYVs · 2025-10-31

**Soundness:** 3
**Presentation:** 3
**Contribution:** 3
**Rating:** 4
**Confidence:** 4

**Summary:**

The paper proposes Probabilistic Granular-Ball Computing (PGBC), a hierarchical one-class data description method designed to model complex, anisotropic data distributions. PGBC extends granular-ball computing by introducing ellipsoidal, probabilistic components that adapt to local data geometry through covariance estimation. The model recursively refines data representations via statistical splitting using BIC and log-likelihood criteria and incorporates entropy-weighted hierarchical scoring for anomaly detection. Experiments on tabular, time series, and open-set datasets show that PGBC outperforms existing baselines (e.g., DeepSVDD, DAGMM, HGAD, and granular-ball variants) in AUC and false positive rate.

**Strengths:**

- Conceptual novelty: Extends granular-ball computing into a probabilistic, hierarchical framework with ellipsoidal modeling, enabling anisotropic and multiscale representation of data.

- Statistical rigor: Splitting and reassignment are governed by BIC and log-likelihood improvement criteria, avoiding ad hoc partitioning.

- Hierarchical anomaly scoring: The entropy-weighted aggregation across levels provides a principled way to combine coarse- and fine-grained representations.

- Comprehensive experiments: Evaluations span 19 tabular, 6 time series, and 5 open-set benchmarks, providing convincing empirical evidence.

- Robustness and efficiency: The paper reports competitive AUC with lower false positive rates and significant model compactness

**Weaknesses:**

- Limited theoretical analysis: The work lacks a formal justification or convergence analysis for the recursive splitting and entropy-based weighting. The connection to mixture model theory remains largely empirical.

- Comparison coverage: The baselines are extensive, but the paper omits recent deep one-class or diffusion-based anomaly detection methods (e.g., OCFlow, CutPaste, or normalizing-flow variants beyond HGAD).

- Ablation and sensitivity analysis: The paper does not quantify the impact of each component (e.g., BIC criterion, entropy weighting, or covariance adaptivity) on performance. This limits interpretability of where the gains come from.

- Scalability concerns: Although PGBC reduces the number of components, its per-iteration Gaussian fitting and reassignment steps may become computationally expensive for very large datasets. No runtime comparison is provided.

- Presentation clarity: While figures are informative, the text is at times dense and overly detailed in algorithmic description. The narrative could be streamlined to highlight key intuitions.

- Evaluation metric bias: The dominance of AUC as the main metric could obscure limitations under other measures (e.g., precision-recall or calibration error).

**Questions:**

- How sensitive is the entropy-based weighting to the number of hierarchical levels or to irregular layer sizes? Would uniform or learned weights perform similarly?

- Does the probabilistic splitting procedure risk overfitting when the local data variance is small but noise-induced? Is there any early-stopping or regularization strategy beyond the BIC penalty?

- How does PGBC perform under distributional drift or streaming data conditions? The conclusion mentions potential for online adaptation—has any preliminary evaluation been conducted?

- Given that each granular-ball corresponds to a Gaussian, can the resulting model be interpreted as an adaptive Gaussian mixture? If so, how does it compare to incremental or variational GMMs in both likelihood and efficiency?

---

> ### Author Response · Authors · 2025-12-03
> **Response to Reviewer GYVs (Part 1)**
>
> We thank Reviewer GYVs for the constructive feedback, which has helped us significantly strengthen the theoretical grounding and comparative analysis of our work. We have addressed each of the concerns raised by the reviewer, as outlined below.
>
> **Q1: Limited theoretical analysis: The work lacks a formal justification or convergence analysis for the recursive splitting and entropy-based weighting. The connection to mixture model theory remains largely empirical.**
>
> **A1:** As suggested, we provided more formal justification for both **the recursive splitting and entropy-based weighting**.
>
> **1) Justification for the Recursive Splitting**
>
> **Theoretical Justification:** The recursive splitting process functions as a rigorous statistical hypothesis test. The splitting condition $BIC(M_2)<BIC(M_1)$  implies that a split is accepted only if the improvement in model fit significantly outweighs the cost of added complexity. This relationship can be reformulated as:
> $2⋅ΔlogL>(k_2−k_1)log⁡N_j$.
> Here, $Δlog⁡L$ represents the likelihood gain obtained by modeling the local data with a two-component GMM ($M_2$) instead of a single Gaussian ($M_1$) , while $(k_2−k_1)log⁡N_j$ represents the complexity penalty for the additional parameters.
>
> **Convergence is strictly guaranteed by the following dynamics:**
>
> - **Diminishing Likelihood Gain:** As the recursion deepens, the granular-ball shrinks and the local data subset $N_j$ decreases. Crucially, as the region becomes smaller, the local data distribution increasingly approximates a single Gaussian (local linearity). Consequently, the benefit of splitting ($Δlog⁡L$) diminishes towards zero.
> - **Persistent Penalty:** In contrast, the complexity penalty term remains strictly positive (since $k_2>k_1$).
> - **Termination:** Eventually, the diminishing likelihood gain fails to overcome the complexity penalty. The inequality ceases to hold, thereby preventing infinite recursion and strictly enforcing statistical convergence.
>
> **Empirical Verification:** To empirically prove this guarantee, we compared PGBC against a variant without the BIC penalty ("w/o BIC"). As shown in the table below (full details in Appendix G.2 of the revised paper), removing the BIC constraint causes the model to fail to converge, leading to an explosion in components (e.g., 668 vs. 25 on Abalone) and computational costs. This confirms that BIC effectively prevents unreasonable or excessive splitting.
>
> | **Datasets** | **Metric** | **PGBC (Full)** | **w/o BIC** |
> | --- | --- | --- | --- |
> | **Abalone** | Balls ↓ | **25** | 668 |
> |  | Time (s) ↓ | **3.13** | 183.41 |
> | **Cardio** | Balls ↓ | **17** | 721 |
> |  | Time (s) ↓ | **1.83** | 42.80 |
> | **WSD** | Balls ↓ | **9** | 382 |
> |  | Time (s) ↓ | **1.46** | 37.33 |
>
> **2) Justification for Entropy-based Weighting:**
>
> - **Theoretical intuition:** Entropy at a specific hierarchical level quantifies the granularity of the data partition. Higher entropy indicates a finer-grained partition with more granular balls, capturing intricate local structures. By weighting levels proportional to their entropy, the model adaptively assigns higher importance to these detailed representations. Consequently, when the data structure is complex and necessitates deep recursive splitting, the influence of coarse-grained levels (which provide only rough global statistics and have lower entropy) is naturally attenuated. This ensures that the anomaly detection is primarily driven by the high-resolution levels that precisely fit the data geometry.
> - **Empirical evidence:** In Appendices H.1 and H.2 of the revised paper, we compare this strategy with the "Uniform" and "Leaf-only" schemes, respectively. Their results are summarized in the table below. It shows that entropy weighting consistently yields the highest AUC, confirming it correctly identifies the optimal resolution for anomaly detection.
>
> | **Datasets (AUC %)** | **Leaf-only** | **Uniform** | **Entropy (Ours)** |
> | --- | --- | --- | --- |
> | **Bands34** | 97.4 | 96.4 | **100.0** |
> | **Bands42** | 99.1 | 97.8 | **100.0** |
> | **Ecoli** | 88.8 | 88.4 | **89.4** |
> | **Sick72** | 84.2 | 78.3 | **87.3** |
> | **Thyroid** | 67.8 | 72.9 | **73.0** |
> | **Waveform** | 77.6 | 74.1 | **78.1** |

---

> ### Author Response · Authors · 2025-12-03
> **Response to Reviewer GYVs (Part 2)**
>
> **Q2: Comparison coverage: The baselines are extensive, but the paper omits recent deep one-class or diffusion-based anomaly detection methods (e.g., OCFlow, CutPaste, or normalizing-flow variants beyond HGAD).**
>
> **A2:** We appreciate the suggestions on expanding our comparison coverage
>
> As suggested, we have significantly broadened our evaluation by incorporating representative recent deep one-class, generative, and open-set recognition methods. Specifically, we added OCFlow (Normalizing Flow) and OCSVM (Kernel-based) for time-series anomaly detection, CutPaste (Self-supervised), U-Student (Student-Teacher), and P-SVDD (Deep One-Class) for visual anomaly detection, and OpenMax (OSR Baseline) for time-series open-set recognition.
>
> The results are summarized in the below. Full results and more details are provided in Appendix E.1, Appendix E.2 and Appendix F of the revised paper.
>
> **i) Comparison on Time-Series Anomaly Detection**
>
> We incorporated the suggested flow-based method OCFlow and the popular baseline OCSVM on the 6 time-series datasets.
>
> | **Datasets** | **OCSVM** | **OCFlow** | **PGBC (Ours)** |
> | --- | --- | --- | --- |
> | NAB Traffic | 78.9 | 74.0 | **85.4** |
> | WSD WebService | 92.9 | 94.8 | **98.8** |
> | SMD Facility | 88.0 | 97.1 | **97.9** |
> | IOPS WebService | 84.4 | 84.0 | **96.4** |
> | UCR Medical | 66.3 | 80.7 | **93.5** |
> | YAHOO Synthetic | 41.0 | 58.4 | **99.8** |
> | **Average** | 75.3 | 81.5 | **95.3** |
>
> Analysis: PGBC achieves the highest average AUC of 95.3%, significantly surpassing OCFlow (81.5%) and OCSVM (75.3%). This advantage stems from the intrinsic geometry of time-series embeddings, which often lie on smooth, anisotropic manifolds. While fixed kernels (OCSVM) and standard normalizing flows (OCFlow) struggle to capture these locally varying structures—as evidenced by OCFlow's high instability on YAHOO—PGBC leverages adaptive ellipsoidal granular-balls that naturally align with the principal directions of the data, ensuring a more precise and stable description.
>
> **ii) Comparison on Visual Anomaly Detection (Section 3.6 in the revised paper)**
>
> To address the request for recent deep one-class models and vision-specific methods, we conducted more experiments on CIFAR-10, FashionMNIST, and MVTec-AD.
>
> **Results on General Visual Datasets:**
>
> | **Datasets (Avg AUC %)** | **DeepSVDD** | **HGAD** | **PGBC (Ours)** |
> | --- | --- | --- | --- |
> | **CIFAR-10** | 69.5 | 95.3 | **95.5** |
> | **FashionMNIST** | 58.5 | 94.6 | **94.9** |
>
> **Results on Industrial Anomaly Detection (MVTec-AD):**
>
> | **Dataset (Avg AUC %)** | **CutPaste** | **U-Student** | **P-SVDD** | **PGBC (Ours)** |
> | --- | --- | --- | --- | --- |
> | **MVTec-AD** | 90.9 | 92.5 | 92.1 | **93.0** |
>
> Analysis: PGBC consistently surpasses specialized methods like CutPaste, U-Student, and P-SVDD (a Deep One-Class variant). This confirms that PGBC's hierarchical probabilistic modeling is highly effective for high-dimensional feature representations.
>
> **iii) Comparison on Open-Set Recognition (Appendix E.2 in the revised paper)**
>
> To further validate the method's versatility, we compared PGBC against the foundational OSR method OpenMax on 5 UCR datasets.
>
> | **Datasets (AUC %)** | **OpenMax** | **PGBC (Ours)** |
> | --- | --- | --- |
> | Adiac | 92.4 | **99.2** |
> | CBF | 49.5 | **100.0** |
> | **Average** | 68.7 | **99.6** |
>
> Analysis: PGBC demonstrates overwhelming superiority (99.6% vs. 68.7%). OpenMax relies on SoftMax calibration which assumes known classes form tight clusters, whereas time-series embeddings often lie on elongated manifolds. PGBC's adaptive ellipsoidal modeling successfully captures these complex geometries.

---

> ### Author Response · Authors · 2025-12-03
> **Response to Reviewer GYVs (Part 3)**
>
> **Q3: Ablation and sensitivity analysis: The paper does not quantify the impact of each component (e.g., BIC criterion, entropy weighting, or covariance adaptivity) on performance. This limits interpretability of where the gains come from.**
>
> **A3:** As suggested, we have significantly expanded our ablation studies in the revised paper.
>
> The new organization is as follows:
>
> - **Appendix G: Ablation Studies on Construction**
>     - G.1 Impact of the Dynamic Reassignment Step
>     - G.2 Impact of the BIC Criterion
>     - G.3 Does the Method Support a Single Principal Component?
> - **Appendix H: Anomaly Scoring Mechanisms**
>     - H.1 Impact of Hierarchical Aggregation (Hierarchy vs. Leaf-only)
>     - H.2 Impact of Weighting Schemes (Entropy vs. Uniform)
>     - H.3 Impact of Scoring Metric (Log-Likelihood vs. Distance)
>     - H.4 Impact of Score Normalization
>
> Specifically, regarding the analysis of the impact of the **BIC criterion, entropy weighting, and covariance adaptivity** that you suggested, our supplementary experiments and findings are detailed below:
>
> **i) Impact of BIC Criterion:**
>
> As detailed in Appendix G.2, the BIC criterion acts as a critical statistical regularizer. We compared the full model against a variant without the BIC penalty ("w/o BIC"). Without BIC, the model aggressively overfits local noise, causing an explosion in the number of components (e.g., from 25 to 668 on *Abalone*) and computational cost, without benefiting detection accuracy.
>
> | **Datasets** | **Metric** | **w/o BIC** | **PGBC (Full)** |
> | --- | --- | --- | --- |
> | **Abalone** | Balls ↓ | 668 | **25** |
> |  | Time (s) ↓ | 183.41 | **3.13** |
> | **WSD** | Balls ↓ | 382 | **9** |
> |  | Time (s) ↓ | 37.33 | **1.46** |
>
> **ii) Impact of Entropy Weighting (Entropy vs. Uniform):**
>
> We compared our entropy-based weighting against a "Uniform" strategy where all hierarchical levels contribute equally. The results demonstrate that "Uniform" weighting is sub-optimal. By using entropy to measure information granularity, PGBC adaptively assigns higher weights to levels with richer structural information, thereby ensuring robustness across diverse datasets like *Sick72* and *Bands34*.
>
> | **Datasets (AUC %)** | **Bands34** | **Bands42** | **Sick72** | **Thyroid** |
> | --- | --- | --- | --- | --- |
> | **Uniform** | 96.4 | 97.8 | 78.3 | 72.9 |
> | **PGBC (Ours)** | **100.0** | **100.0** | **87.3** | **73.0** |
>
> **iii) Impact of Covariance Adaptivity (Log-Likelihood vs. Distance):**
>
> We compared PGBC against a "Distance" baseline that uses the exact same granular-ball centers but computes anomaly scores using Euclidean distance (ignoring covariance). As shown below, discarding covariance information leads to a drastic performance drop. This confirms that probabilistic covariance modeling is essential for capturing the true data structure.
>
> | **Datasets (AUC %)** | **Bands34** | **Bands42** | **Thyroid** | **Waveform** |
> | --- | --- | --- | --- | --- |
> | **Min Distance (No Covariance)** | 83.9 | 81.8 | 56.2 | 71.6 |
> | **PGBC (Ours)** | **100.0** | **100.0** | **73.0** | **78.1** |
>
> **Q4: Scalability concerns: Although PGBC reduces the number of components, its per-iteration Gaussian fitting and reassignment steps may become computationally expensive for very large datasets. No runtime comparison is provided.**
>
> **A4:** To address the scalability concerns, we conducted a runtime efficiency comparison against the classical deep baseline **AutoEncoder**, the hierarchical probabilistic method **HGAD** and the granular-ball method **GBDO**.
>
> As shown in the table below, PGBC remains highly efficient. Notably, PGBC achieves an average runtime of 13.03s, which is nearly **2x faster** than HGAD (24.67s).
>
> While HGAD utilizes a predefined hierarchical structure involving **i**nter-class Gaussian mixture modeling and intra-class mixed-class centers learning, our method relies on adaptive statistical splitting. This demonstrates that our strategy is significantly more efficient than the complex optimization required by such fixed-structure models.
>
> Appendix I in the revised paper reports the full empirical runtime comparison results.
>
> | **Method** | **AutoEncoder** | **HGAD** | **GBDO** | **PGBC (Ours)** |
> | --- | --- | --- | --- | --- |
> | **Time (s)** | 18.59 | 24.67 | 13.89 | **13.03** |

---

> ### Author Response · Authors · 2025-12-03
> **Response to Reviewer GYVs (Part 4)**
>
> **Q5: Presentation clarity: While figures are informative, the text is at times dense and overly detailed in algorithmic description. The narrative could be streamlined to highlight key intuitions.**
>
> **A5:** We appreciate your constructive suggestion. In the revised version, we have incorporated additional explanations to better **highlight the key intuitions** behind our framework. Specifically, we have made the following three updates:
>
> 1. **line 188-189:** We added a high-level summary at the start of the splitting step to clarify that the process systematically decomposes the complex global distribution into simpler ellipsoidal components.
> 2. **line 206-208:** We explicitly highlighted that the BIC criterion functions as a **statistical regularizer** to prevent overfitting, making the motivation behind the formula immediately clear.
> 3. **line 282-284:** We clarified the scoring mechanism by emphasizing that entropy-based weighting enables the model to prioritize rich structural information at fine-grained levels over coarse global statistics.
>
> **Q6: Evaluation metric bias: The dominance of AUC as the main metric could obscure limitations under other measures (e.g., precision-recall or calibration error).**
>
> **A6:** We appreciate the reviewer's emphasis on comprehensive evaluation.
>
> While AUC was chosen as the primary metric for its threshold independence and cross-task comparability, we agree that a holistic assessment requires a diverse set of metrics.
>
> To further address the reviewer’s concern, we have expanded our evaluation in two dimensions:
>
> 1. **False Positive/Negative Rates (FPR/FNR):** As detailed in Section 3.4 and Appendix D in the revised paper, PGBC achieves the lowest average FPR (2.36%) and lowest average FNR (35.9%) across 19 datasets. This confirms that our high AUC reflects a precise decision boundary rather than trivial trade-offs between precision and recall.
> 2. **F1-Score:** To evaluate performance at specific decision thresholds, we additionaly report the best F1-score (along with corresponding Precision and Recall) for each method. As shown in the representative results below (full results for 12 datasets are in Appendix E.3), PGBC consistently achieves superior F1-scores compared to strong baselines.
>
> These results confirm that PGBC's advantage is consistent across both threshold-independent (AUC) and threshold-dependent (F1) metrics, demonstrating robust performance for practical deployment.
>
> | **Datasets\Metric (F1 % ↑)** | **HGAD** | **PGBC (Ours)** |
> | --- | --- | --- |
> | **Bands42** | 90.2 | **100.0** |
> | **Pageblocks** | 89.1 | **99.2** |
> | **Pendigits** | 95.6 | **100.0** |
> | **Tictac26** | 94.1 | **100.0** |
>
> **Q7: How sensitive is the entropy-based weighting to the number of hierarchical levels or to irregular layer sizes? Would uniform or learned weights perform similarly?**
>
> **A7:** As suggested, we provided a comparison with a **uniform-weight** baseline in the table below.
>
> | **Datasets (AUC %)** | **Uniform** | **Entropy (Ours)** |
> | --- | --- | --- |
> | **Bands34** | 96.4 | **100.0** |
> | **Bands42** | 97.8 | **100.0** |
> | **Ecoli** | 88.4 | **89.4** |
> | **Sick72** | 78.3 | **87.3** |
> | **Thyroid** | 72.9 | **73.0** |
> | **Waveform** | 74.1 | **78.1** |
>
> As can be seen, our entropy-based weighting consistently outperforms the **Uniform** baseline. For instance, on the *Bands34* dataset, the AUC drops from **100.0%** (Ours) to **96.4%** with the "Uniform" strategy. This significant gap demonstrates that treating all hierarchical levels equally is sub-optimal, as it fails to distinguish between informative layers and those containing redundant or noisy statistics.
>
> Concerning learned weights, we chose not to pursue this approach to keep the model simple and efficient. Adding learnable parameters for layer weights would generally involve a complex bi-level optimization framework, which would greatly increase the optimization workload and complicate the practical deployment of the method.
>
> Empirically, the proposed **entropy-based weighting** offers an effective balance, obtaining satisfactory performance through a simple, data-driven mechanism without the overhead of additional parameter tuning.
>
> Comprehensive experiments and detailed analysis regarding the anomaly scoring mechanisms are provided in Appendix H.

---

> ### Author Response · Authors · 2025-12-03
> **Response to Reviewer GYVs (Part 5)**
>
> **Q8: Does the probabilistic splitting procedure risk overfitting when the local data variance is small but noise-induced? Is there any early-stopping or regularization strategy beyond the BIC penalty?**
>
> **A8:** The risk of overfitting can be mitigated through the dual mechanism of statistical selection and the numerical regularization in the proposed PGBC.
>
> **1) BIC as Statistical Early-Stopping:**
>
> As empirically demonstrated in our ablation study (Appendix G.3 of the revised paper), the BIC criterion acts as a strict statistical regularizer. When the BIC penalty is removed, the model aggressively overfits local noise, leading to an explosion in model complexity (e.g., from 25 to 668 components on *Abalone*). The complexity penalty term ($klogN$) ensures that splits are rejected unless the likelihood gain significantly outweighs the cost of adding parameters, thereby effectively filtering out splits driven solely by noise-induced variance.
>
> **2) Regularization Beyond BIC:**
>
> To further prevent instability in regions with small or collapsing variance, we explicitly apply covariance regularization during density estimation. As detailed in Section 2.1 (line 187-188), a small regularization term( $ϵI$ ) is added to the covariance diagonal. This standard technique prevents singularity and ensures robust modeling even when local data points are sparse or lie on lower-dimensional manifolds.
>
> **Q9: How does PGBC perform under distributional drift or streaming data conditions? The conclusion mentions potential for online adaptation—has any preliminary evaluation been conducted?**
>
> **A9:** Thank you for your question. The SMD data in our experiments is an example where distribution may experience some drift (as illustrated in Appendix K, Figure 16(c)), and our method does indeed show potential for online adaptation in this context.
>
> In fact, the proposed PGBC can incorporate various popular techniques, such as instance normalization[1], to help address nonstationary cases or distributional drift.
>
> [1] "A Time Series is Worth 64 Words: Long-term Forecasting with Transformers." (ICLR 2023)
>
> **Q10: Given that each granular-ball corresponds to a Gaussian, can the resulting model be interpreted as an adaptive Gaussian mixture? If so, how does it compare to incremental or variational GMMs in both likelihood and efficiency?**
>
> **A10:** Yes, the leaf nodes of PGBC can indeed be interpreted as an adaptive Gaussian Mixture Model (GMM). However, **a key distinction is that standard incremental or variational GMMs typically operate on a "flat," single-granularity representation**. In contrast, PGBC leverages a hierarchical multi-granularity structure combined with an entropy-based weighting strategy.
>
> As illustrated in the Thyroid dataset case study (see Appendix H.1, Figure 7), detection performance at a single granularity level can vary considerably. By aggregating likelihoods across the entire hierarchy, PGBC effectively integrates information from coarse to fine scales, reducing the risk of relying on a suboptimal single resolution and enhancing robustness compared to flat probabilistic models.
>
> Regarding efficiency, PGBC employs the Bayesian Information Criterion (BIC) to rigorously manage model complexity, ensuring a highly compact representation (as detailed in Appendix H.3 of the revised paper) that helps minimize computational overhead.

---

### Official Review · Reviewer_2hzn · 2025-11-01

**Soundness:** 3
**Presentation:** 4
**Contribution:** 3
**Rating:** 6
**Confidence:** 3

**Summary:**

The work tackles the problem of anomaly detection using one-class data description. To incorporate the diverse nature of available samples, it proposes a probabilistic granular ball computing framework, where samples are aggregated in hierarchical clusters, maintaining an ellipsoidal structure along the principal component. By leveraging the hierarchical structure, an aggregated entropy-based metric is proposed, which acts as an anomaly score to distinguish between normal and anomaly samples in testing. Comprehensive experiments are conducted against 19 tabular datasets, 6 time series anomaly datasets, and 5 time series open-set recognition datasets.

**Strengths:**

- The paper is easy to follow, with intuitive illustrations of the proposed concepts.
- The motivation to propose an ellipsoidal structure along the principal component is well-formulated.
- Experiments are comprehensive with a diverse range of datasets.

**Weaknesses:**

- The performance is sub-optimal in a significant number of cases.
- Anomaly detection problem can be approached from an open-set recognition (OSR) perspective as well. It is suggested to include comparisons with some of the baselines from OSR [1,2].
- It is suggested to include the ablation of the proposed components. For example, how does it impact the performance if the dynamic reassignment step is omitted?


References:

[1] Zhang, Hongjie, et al. "Hybrid models for open set recognition." European Conference on Computer Vision. Cham: Springer International Publishing, 2020.

[2] Bendale, Abhijit, and Terrance E. Boult. "Towards open set deep networks." Proceedings of the IEEE conference on computer vision and pattern recognition. 2016.

**Questions:**

- How can this framework be extended to vision datasets?
- Is one-class data description applicable to other fields than anomaly detection?
- Does the method support a single principal component, or the method can be extended?

---

> ### Author Response · Authors · 2025-12-03
> **Response to Reviewer 2hzn (Part 1)**
>
> We sincerely thank Reviewer 2hzn for providing a thorough review and insightful suggestions that have significantly expanded our evaluation.
>
> **Q1:  The performance is sub-optimal in a significant number of cases.**
>
> **A1:** Given the variety of strong baselines evaluated and the diversity of data tasks, it is expected that we may not achieve optimal performance on every task.
>
> However, we attained **the best average performance**, achieving 91.1% on tabular data, 95.3% on time-series data, and 99.6% on open-set recognition. In Table 1, our method achieved optimal performance on 13 of 19 tasks and placed second on two additional tasks.
>
> Furthermore, we discussed the advantages of PGBC in Section 3.2 (Lines 390-399 in the original submission, now Lines 399-404 in the revised paper). PGBC demonstrates remarkable robustness on datasets with extremely low anomaly ratios (as low as 0.28% on YAHOO, achieving 99.8% AUC) and complex anisotropic structures, where the data distribution is not spherical. By effectively modeling these challenging cases, where spherical methods typically struggle, PGBC provides the most reliable and accurate solutions across multiple domains.
>
> **Q2: Anomaly detection problem can be approached from an open-set recognition (OSR) perspective as well. It is suggested to include comparisons with some of the baselines from OSR.**
>
> [1] Zhang, Hongjie, et al. "Hybrid models for open set recognition." European Conference on Computer Vision. Cham: Springer International Publishing, 2020.
>
> [2] Bendale, Abhijit, and Terrance E. Boult. "Towards open set deep networks." Proceedings of the IEEE conference on computer vision and pattern recognition. 2016.
>
> **A2:** As suggested, we compared the baseline OpenMax (from reference [2]) in the table below. Regarding the suggested reference [1], we were unfortunately unable to access the code.
>
> As demonstrated, PGBC exhibits a clear advantage over OpenMax, achieving an average AUC of 99.6% compared to OpenMax's 68.7%.
>
> OpenMax encounters difficulties (e.g., 49.5% on *CBF*) because it assumes that known classes form tight clusters—an assumption that is often violated in time-series tasks where feature embeddings are distributed along continuous, elongated manifolds, as illustrated in Figure 3 of the main text. In contrast, PGBC effectively addresses this geometric challenge: its adaptive ellipsoidal granular balls can locally align with these manifolds, forming a tight boundary that robustly rejects open-set samples.
>
> Additionally, we have made extra efforts to incorporate more popular baselines in both the time-series (e.g., OCFlow) and visual (e.g., CutPaste) domains in the revised paper (Appendix E and F), as suggested by Reviewer GYVs.
>
> |  Dataset             |  OpenMax   |  **PGBC (Ours)** |
> | --- | --- | --- |
> |  Adiac               |  92.4 ± 7.4              |  **99.2 ± 0.2**               |
> |  CBF                 |  49.5 ± 0.3              |  **100.0 ± 0.0**              |
> |  Synthetic Control   |  53.0 ± 5.0              |  **99.8 ± 1.1**               |
> |  SwedishLeaf         |  63.9 ± 4.8              |  **99.7 ± 0.1**               |
> |  Trace               |  84.9 ± 1.8              |  **99.6 ± 1.0**               |
> |  **Average**         |  68.7 ± 3.9              |  **99.6 ± 0.5**               |
>
> **Q3: It is suggested to include the ablation of the proposed components. For example, how does it impact the performance if the dynamic reassignment step is omitted?**
>
> **A3:**  As suggested, we have added more ablation study results in the revised appendix:
>
> - **Appendix G: Ablation Studies on Construction**
>     - G.1 Impact of the Dynamic Reassignment Step
>     - G.2 Impact of the BIC Criterion
>     - G.3 Does the Method Support a Single Principal Component?
> - **Appendix H: Anomaly Scoring Mechanisms**
>     - H.1 Impact of Hierarchical Aggregation (Hierarchy vs. Leaf-only)
>     - H.2 Impact of Weighting Schemes (Entropy vs. Uniform)
>     - H.3 Impact of Scoring Metric (Log-Likelihood vs. Distance)
>     - H.4 Impact of Score Normalization
>
> The table below illustrates the impact of the dynamic reassignment step you suggested. As can be seen, this design leads to consistent improvements in AUC and reductions in the false-positive rate (FPR). Notably, there is a +3.6% AUC improvement on Cardio, which demonstrates the effectiveness of this approach.
>
> | **Datasets** | **Reassign** | **AUC (%) ↑** | **FPR (%) ↓**  |
> | --- | --- | --- | --- |
> | Cardio | × | 80.41 | 1.74 |
> |  | ✓ | **84.04** | **1.63** |
> | Abalone | × | 70.46 | 1.78 |
> |  | ✓ | **72.10** | **1.72** |
> | SwedishLeaf | × | 98.10 | 2.10 |
> |  | ✓ | **98.66** | **1.81** |

---

> ### Author Response · Authors · 2025-12-03
> **Response to Reviewer 2hzn (Part 2)**
>
> **Q4:  How can this framework be extended to vision datasets?**
>
> **A4:**  By leveraging a pretrained visual encoder, our method can be easily applied to **vision datasets** and achieve competitive results.
>
> The table below reports the results on MVTec-AD, where PGBC and feature-based baselines (KNN, GBMOD, HGAD) were evaluated on 512-dimensional ViT features using the ADBench protocol. As shown in the table, PGBC consistently demonstrates exceptional performance across various benchmarks. For the MVTec-AD industrial dataset, PGBC leads with an AUC of 93.0%, surpassing competitive image-based approaches such as U-student (92.5%) and P-SVDD (92.1%).
>
> | **Method** | **KNN** | **GBMOD** | **HGAD** | **CutPaste** | **U-student** | **P-SVDD** | **Ours** |
> | --- | --- | --- | --- | --- | --- | --- | --- |
> | MVTec-AD | 87.1 | 81.9 | 91.2 | 90.9 | 92.5 | 92.1 | **93.0** |
>
> For full results, including additional datasets and metrics, please refer to Section 3.6 and Appendix F in the revised paper.
>
> **Q5: Is one-class data description applicable to other fields than anomaly detection?**
>
> **A5:** Yes, one-class data description is a versatile paradigm applicable beyond standard anomaly detection to fields like Open-Set Recognition (OSR), industrial quality control, fraud detection, and bioinformatics, effectively modeling "normal" behavior to identify deviations. We validated this potential in Section 3.3, where PGBC achieved state-of-the-art performance (99.6% AUC) on time-series OSR tasks.
>
> **Q6: Does the method support a single principal component, or the method can be extended?**
>
> **A6:** Yes. In Appendix G.3 and Figure 6 of the revised paper, we have added new experiments to address the case of a single principal component, as suggested.
>
> As shown, PGBC natively supports data dominated by a single principal component. It automatically fits a single elongated ellipsoidal granular-ball when the normal class exhibits a strongly anisotropic structure, demonstrating PGBC's ability to adapt to both simple and complex structures. As depicted in Algorithm 2 of the revised paper, when no splitting is performed, we can simply utilize a single Gaussian distribution for inference.

---

### Author Response · Authors · 2025-12-03
**Summary of Revisions**

We sincerely thank all the reviewers for their insightful reviews and positive perspectives, which are a great encouragement to us. Specifically, the reviewers commented:

- The method is **conceptually novel** and **statistically rigorous**, is a **principled approach** (Reviewer GYVs), and a **novel** framework (Reviewer ruXP). The motivation is **well-formulated** (Reviewers 2hzn) and **theoretically reasonable** (Reviewer 5XJV). The idea is **particularly interesting** (Reviewer ruXP). The method is also **robust and efficient** (Reviewer GYVs).
- The work is **well-written**, with clear organization, statements, and illustrations (Reviewers 5XJV, ruXP). It features **excellent presentation** and is **easy to follow with intuitive illustrations** (Reviewer 2hzn).
- The experiments are **comprehensive and extensive** (Reviewers GYVs, 2hzn), providing a **convincing evaluation** and **superior detection** performance (Reviewer 5XJV). The method consistently outperforms strong baselines and represents a **robust solution** (Reviewer ruXP).

The reviewers also raised insightful and constructive concerns. Here is the summary of the major revisions:

1. **Clarification of Root Node ($\mathcal{B}^{(0)}$)  (Reviewers 5XJV, 2hzn):**
    Reviewers are interested in the theoretical completeness of the algorithm in the degenerate case (where no splitting occurs, i.e., $ L = 0 $) and request clarification on the role of the root node ($ \mathcal{B}^{(0)} $) in the scoring mechanism.

    To address their concerns, we have improved Figure 2, Algorithm 2, and Section 2.2. Specifically, we introduced an explicit conditional check: if the granular-ball does not split (i.e., $ L = 0 $), the method gracefully reduces to fitting a single global Gaussian ($\mathcal{B}^{(0)} $). It is important to note that the global anomaly patterns captured by $ \mathcal{B}^{(0)} $ are often straightforward outliers. In a hierarchical setting ($ L \geq 1 $), these points remain distinct from the finer-grained distributions, making them easily detectable by the subsequent granular balls at levels $ l \geq 1 $. This is why our original version did not address the simplest case of $ L = 0 $.

2. **Extended to Visual Tasks & Baselines (Reviewers GYVs, 2hzn, ruXP, 5XJV):** We significantly expanded our evaluation to visual datasets (e.g., MVTec-AD) and incorporated diverse baselines across time-series (e.g., OCFlow) and vision (e.g., CutPaste). Results in Appendices E & F confirm that PGBC consistently outperforms these specialized methods across diverse domains.

3. **More Ablation Studies (Reviewers GYVs, 5XJV, ruXP, 2hzn):** To rigorously quantify the contribution of every key component in PGBC, we have significantly expanded our experimental analysis. In the revised paper, we added detailed ablation results organized by
    - **Appendix G: Ablation Studies on Construction**
        - **G.1:** Impact of the *Dynamic Reassignment Step*
        - **G.2:** Impact of the *BIC Criterion*
        - **G.3:** Does the Method Support a *Single Principal Component?*
    - **Appendix H: Anomaly Scoring Mechanisms**
        - **H.1:** Impact of *Hierarchical Aggregation* (Hierarchy vs. Leaf-only)
        - **H.2:** Impact of *Weighting Schemes* (Entropy vs. Uniform)
        - **H.3:** Impact of *Scoring Metric* (Log-Likelihood vs. Euclidean Distance)
        - **H.4:** Impact of *Score Normalization*
4. **Theoretical Justification (Reviewer GYVs):**
    - **Recursive Splitting:** In the response to Reviewer GYVs, we provided a theoretical analysis of the convergence mechanism, demonstrating that the splitting process is strictly bounded, as the diminishing likelihood gain eventually fails to overcome the persistent complexity penalty imposed by BIC. The empirical verification of this convergence is detailed in Appendix G.2.
    - **Entropy-based Weighting:** We clarified (in the response) that entropy quantifies the granularity of the data description. Our scheme assigns higher weights to levels with higher entropy (indicating a finer-grained partition), thereby prioritizing the layers that capture refined structural information over coarse global approximations. Supporting comparative analyses are provided in Appendix H.1 and Appendix H.2.
5. **Computational Efficiency (Reviewers 5XJV, GYVs):** To address scalability and complexity concerns, we provided empirical runtime comparisons in Appendix I.2. The results demonstrate that PGBC outperforms deep baselines (e.g., AutoEncoder) by avoiding computationally intensive gradient-based optimization. Notably, compared to hierarchical Gaussian mixture models HGAD, PGBC is nearly 2× faster due to its adaptive statistical splitting strategy.

All updates are highlighted in blue in the revised paper. These valuable suggestions from the reviewers are very helpful to us in revising the paper to a better shape. We hope you will appreciate our revised version.

---

### Meta-Review · Area_Chair_Xjtt · 2025-12-22

**Summary:**

The paper proposes PGBC, a hierarchical one-class data description method that models target data using recursively refined ellipsoidal granular-balls, improving representation of anisotropic, multi-scale structures compared with single- or fixed multi-sphere approaches. By aggregating layer-wise scores through a distribution-entropy scheme (interpretable as approximating a hierarchical Gaussian mixture), PGBC captures both global and local patterns, and experiments report competitive performance with low false positives.

Overall, the paper is clearly written and well-motivated, and it includes experiments on a broad set of datasets with promising results. Reviewers provided detailed, constructive feedback, with some common concerns including: (1) limited theoretical justification for the recursive splitting and entropy-based weighting; (2) missing evaluations on commonly used vision benchmarks and against recent deep one-class and diffusion-based anomaly detection baselines; (3) an incomplete ablation study that does not fully isolate key components; and (4) the absence of a formal time-complexity analysis for training and inference.

The authors’ rebuttal adds experiments and addresses some points to a degree. However, several major issues remain. For example, for concern (1) as mentioned above, the use of classical BIC may not be well supported since BIC relies on a Laplace approximation to the marginal likelihood and can be unreliable for mixture models. Likewise, while the rebuttal provides additional results, a more comprehensive and up-to-date evaluation is still needed to substantiate the claims. The authors are encouraged to incorporate the reviewers’ suggestions more thoroughly to strengthen the work for a future submission.

**Reviewer Concerns:**

The authors’ rebuttal provides empirical running-time results to clarify concerns about computational cost, and it adds further ablations to better isolate the contributions of individual components. However, key concerns remain insufficiently addressed, including the limited theoretical justification for the recursive splitting and entropy-based weighting, as well as the lack of evaluation on commonly used vision benchmarks and comparisons against recent relevant baselines.

**Reviewer Scores:**

Given the remaining major concerns, especially the limited theoretical grounding and the incomplete evaluation, the reviewers are likely to maintain their original scores.

---

### Decision · Program_Chairs · 2026-01-26

Reject